# Symbolic recording of signalling and *cis*-regulatory element activity to DNA

Wei Chen[1,2,11 ✉], Junhong Choi[1,3,4,5,11], Xiaoyi Li[1,5], Jenny F. Nathans[1,5,6], Beth Martin[1,5], Wei Yang[1,5], Nobuhiko Hamazaki[1,5,7,8,9], Chengxiang Qiu[1,5], Jean-Benoît Lalanne[1], Samuel Regalado[1,5], Haedong Kim[1,5], Vikram Agarwal[1], Eva Nichols[1], Anh Leith[1], Choli Lee[1,5] & Jay Shendure[1,3,5,9,10 ✉]

Measurements of gene expression or signal transduction activity are conventionally performed using methods that require either the destruction or live imaging of a biological sample within the timeframe of interest. Here we demonstrate an alternative paradigm in which such biological activities are stably recorded to the genome. Enhancer-driven genomic recording of transcriptional activity in multiplex (ENGRAM) is based on the signal-dependent production of prime editing guide RNAs that mediate the insertion of signal-specific barcodes (symbols) into a genomically encoded recording unit. We show how this strategy can be used for multiplex recording of the cell-type-specific activities of dozens to hundreds of *cis*-regulatory elements with high fidelity, sensitivity and reproducibility. Leveraging signal transduction pathway-responsive *cis*-regulatory elements, we also demonstrate time- and concentration-dependent genomic recording of WNT, NF-κB and Tet-On activities. By coupling ENGRAM to sequential genome editing via DNA Typewriter[1], we stably record information about the temporal dynamics of two orthogonal signalling pathways to genomic DNA. Finally we apply ENGRAM to integratively record the transient activity of nearly 100 transcription factor consensus motifs across daily windows spanning the differentiation of mouse embryonic stem cells into gastruloids, an in vitro model of early mammalian development. Although these are proof-of-concept experiments and much work remains to fully realize the possibilities, the symbolic recording of biological signals or states within cells, to the genome and over time, has broad potential to complement contemporary paradigms for how we make measurements in biological systems.

Conventional genomic, proteomic or imaging-based measurement paradigms are powerful yet limited in key ways. For example, with destructive methods such as RNA sequencing (RNA-seq) or mass spectrometry, individual samples provide only static snapshots of a system. Live imaging of fluorescent probes and reporters is better able to capture temporal dynamics but requires that the system be physically transparent, and is limited in terms of the number of analytes that can be concurrently monitored.

An alternative paradigm to endpoint or real-time measurement is to record information over time. DNA is the natural medium for biological information storage. Various enzymatic systems have been used to alter genomic DNA in a biologically conditional manner—for example, site-specific recombinases (SSRs)[2,3]. As conventionally used by developmental biologists, SSRs are expressed under the control of a cell-type-specific enhancer. In tissues in which that enhancer is active, SSR-mediated recombination at a target locus excises a stop sequence, unlocking the expression of a fluorescent reporter in that

cell and its descendants. Multiplex versions of SSR-based recorders leverage excision and flipping to combinatorially diversify fluorescent reporter expression[4,5] or DNA barcodes[6,7]. CRISPR genome editing has also been adapted to biologically conditional recording[8]. Some methods repurpose CRISPR–Cas spacer acquisition systems to 'log' events in prokaryotic systems—for example, DNA, RNA or metabolites[9–13]. Other methods, including CAMERA[14] and DOMINO[15], link specific small molecules or signalling pathways to CRISPR base editor activity.

However, each of these methods is sharply constrained with respect to multiplexability—that is, the number of independent signals that can be recorded simultaneously. For SSRs, multiplexing requires that each signal drive a distinct SSR. Also, SSR-based recording systems do not capture the strength, duration or order of signals. For biologically conditional CRISPR recorders[14,15], RNA polymerase II (Pol2) promoters can be used to drive single guide RNA (sgRNA) expression[16] such that, at least in principle, multiple signal-specific reporters could be deployed within the same cell—for example, by leveraging a separate

[1]Department of Genome Sciences, University of Washington, Seattle, WA, USA. [2]Molecular Engineering and Sciences Institute, University of Washington, Seattle, WA, USA. [3]Howard Hughes Medical Institute, Seattle, WA, USA. [4]Developmental Biology Program, Memorial Sloan Kettering Cancer Center, New York, NY, USA. [5]Seattle Hub for Synthetic Biology, Seattle, WA, USA. [6]Medical Scientist Training Program, University of Washington, Seattle, WA, USA. [7]Department of Obstetrics & Gynecology, University of Washington, Seattle, WA, USA. [8]Institute for Stem Cell & Regenerative Medicine, University of Washington, Seattle, WA, USA. [9]Brotman Baty Institute for Precision Medicine, Seattle, WA, USA. [10]Allen Discovery Center for Cell Lineage Tracing, Seattle, WA, USA. [11]These authors contributed equally: Wei Chen, Junhong Choi. ✉e-mail: wchen108@uw.edu; shendure@uw.edu

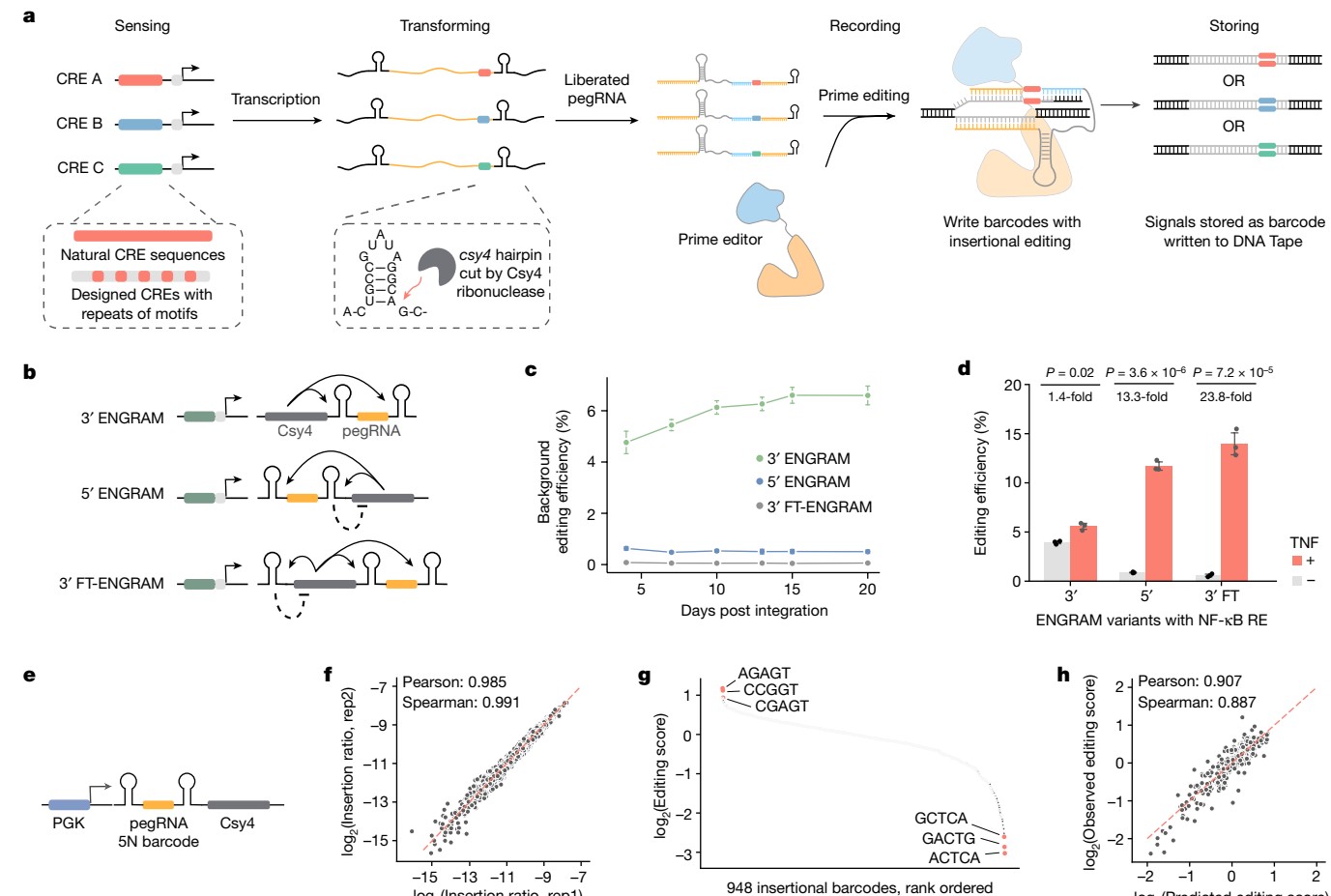

**Fig. 1 | ENGRAM. a**, Schematic of ENGRAM. Endogenous or designed CREs drive signal-dependent, Pol2-mediated production of a Csy4 transcript bearing an embedded pegRNA. Csy4 cleaves two 17 bp *csy4* hairpins from its own transcript, liberating the pegRNA to write a CRE-specific insertional barcode to DNA Tape. **b**, Three ENGRAM architectures were tested. Solid lines correspond to Csy4 targeting *csy4* hairpins; dashed lines correspond to potential for cleavage events to mediate autoregulatory negative feedback on Csy4 levels. **c**, ENGRAM recorders, driven only by minP and encoding a degenerate 5-mer insertion to the *HEK3* locus, were integrated to PE2(+) HEK293T cells. Background accumulation at *HEK3* was monitored for 20 days. **d**, NF-κB recorders were integrated to PE2(+) HEK293T cells. Recording at *HEK3* was measured in the presence versus absence of 10 ng ml$^{-1}$ TNF. *P* values derived from two-tailed *t*-test. Data in **c** and **d** are mean and s.d. from *n* = 3 integration replicates. **e**–**h**, Insertional barcodes predictably bias recording efficiency. **e**, A 5′ ENGRAM recorder library with constitutive (PGK-driven) production of pegRNAs encoding a degenerate 5-mer insertion into *HEK3* was integrated to PE2(+) HEK293T cells. **f**, The log-scaled abundances of individual 5-mer insertions at *HEK3* were highly correlated between transfection replicates (rep1 and rep2). **g**, Editing scores were calculated as (genomic reads with insertion/total edited *HEK3* reads)/(plasmid reads with insertion/total plasmid reads) and are plotted here for 948 5-mers. **h**, Predicted versus observed editing scores for 5-mer insertions. A linear lasso regression model was trained on one-hot encoded single and dinucleotide content of the 5-mer, together with the MFE of the predicted secondary structure. The model was trained with tenfold cross-validation on a 680 barcode training set and then applied to predict editing scores on a held-out 268 barcode test set.

trio of components (*cis*-regulatory elements (CREs), sgRNA and target site) per signal[17,18]. However, in practice, such a system would be limited by the fact that the information is effectively captured by the location(s) of the edited site(s) rather than by the edit itself. Even with homing or self-targeting sgRNAs[19,20], the 'write' events corresponding to each signal would occur at different locations in the genome. This is particularly limiting if one is aiming to temporally order multiple signals, because each potential order (the number of which grows exponentially with the number of signals) must be 'precoded' as an editable template[14,15] (Supplementary Table 1).

Overall there remains a need for a biologically conditional recording system that is at once quantitative, reproducible, temporally resolved, applicable to opaque systems and expansible to the concurrent measurement of thousands of biological signals. Here we describe a new framework, enhancer-driven genomic recording of transcriptional activity in multiplex (ENGRAM), that aims to meet these criteria. We reasoned that a signal-responsive CRE positioned

upstream of a minimal promoter (minP)[21] could drive the production of a 'writing unit', in the form of a prime editing[22] guide RNA (pegRNA) that programmes the insertion of a CRE-specific insertion to a DNA Tape (Fig. 1a). To facilitate multiplexing, many ENGRAM recorders in the same system may share a common spacer while encoding different insertions, such that signal-specific symbols will be written to a shared location. The DNA Tape can be either an endogenous locus (two copies, 'endogenous DNA Tape') or a synthetic sequence (for example, dozens of copies of target introduced via piggyBac transposition, 'synthetic DNA Tape'). As we show here, ENGRAM is compatible with DNA Typewriter[1], a method enabling sequential genome editing, such that all possible orders of a vocabulary of biologically conditional symbols can potentially accrue to a common DNA Typewriter Tape (DTT).

The acronym ENGRAM is inspired by the use of 'engram' in neuroscience to refer to the physical manifestation of a unit of memory. We use 'ENGRAM recorder' to refer to the pegRNA expression cassette, and

'ENGRAM recorder system' to refer to the combination of ENGRAM recorder, prime editor and DNA Tape. In evaluation of reproducibility we relied on 'transfection replicates' and 'integration replicates'. Detailed explanations of the nature of replicates used in each experiment are provided in Supplementary Table 7.

## Development and evaluation of ENGRAM

Transcripts for translated genes, including CRE-minP-driven reporter transcripts, are made by Pol2 whereas small untranslated RNAs are made by RNA polymerase III (Pol3). When CRISPR sgRNAs are used in mammalian cells they are routinely driven by a Pol3 promoter. To facilitate CRE-dependent, Pol2-mediated production of pegRNAs we leveraged the CRISPR endoribonuclease Csy4 (Cas6f), which cleaves at the 3′ end of a 17-base-pair (bp) RNA hairpin (*csy4*)[23–26]. In this scheme, CRE activity drives expression of a Pol2 transcript that includes *csy4*-pegRNA-*csy4*. Csy4 cleaves both *csy4* hairpins, liberating a functional pegRNA (Fig. 1a). In nearly all experiments reported in this paper, pegRNAs were designed to write insertions to endogenous or synthetic *HEK3* target sites[22] (Extended Data Fig. 1a).

Following early tests (Supplementary Note 1), we settled on an architecture for ENGRAM in which the *csy4*-pegRNA-*csy4* cassette is embedded within an untranslated region (UTR) of a CRE-minP-driven Csy4 transcript (Extended Data Fig. 1b). We evaluated three designs: two positioning *csy4*-pegRNA-*csy4* in the 3′ UTR (3′ ENGRAM) or 5′ UTR (5′ ENGRAM) and a variant of 3′ ENGRAM that bore an additional *csy4* hairpin in the 5′ UTR (3′ FT ENGRAM) (Fig. 1b). The 5′ and 3′ FT ENGRAM designs, which share the potential for autoregulatory negative feedback on Csy4 levels, exhibited 12- and 110-fold lower background than 3′ ENGRAM, respectively (Fig. 1b,c). For all three designs, background editing plateaued after several days (Fig. 1c).

To further compare these three designs, we positioned an NF-κB response element[27] upstream of minP and separately integrated each design into the genomes of PE2(+) HEK293T cells via piggyBac transposition. We then measured recording at the endogenous *HEK3* locus in the absence versus presence of TNF, an NF-κB agonist. We observed 1.4-, 13.3- and 23.8-fold activation for 3′, 5′ and 3′ FT ENGRAM recorders, respectively (Fig. 1d). Although 3′ FT ENGRAM exhibited the highest signal-to-noise ratio (13.9 versus 0.58% editing with versus without TNF), all subsequent experiments were performed with 5′ ENGRAM because its architecture facilitates straightforward pairing of CREs and pegRNA-encoded insertions during cloning.

Additional optimizations included evaluation of alternative guide sequences (engineered pegRNAs (epegRNAs) versus pegRNAs)[28], prime editors (PEmax versus PE2)[29] or guide release systems (*csy4* versus transfer RNA (tRNA))[16] (Supplementary Note 1). We found that ENGRAM benefits from more active prime editors such as PEmax (used in some experiments below), but not from switching to epegRNAs nor from tRNA-based guide release. We confirmed that the ENGRAM recorder system does not substantially alter the cellular transcriptome (Supplementary Note 1).

To evaluate whether the insertion barcode biases editing efficiency, we leveraged a constitutively active ENGRAM recorder library bearing pegRNAs encoding a degenerate 5-mer insertion to be written to endogenous *HEK3* (Fig. 1e). Measuring recording efficiencies 3 days following transient transfection of this library into PE2(+) HEK293T cells in triplicate, we observed nearly all 5-mer insertions at highly reproducible frequencies (1,023 out of 1,024; Fig. 1f and Extended Data Fig. 2a–c). Of the 948 5-mers observed at least once in each transfection replicate, 91% exhibited efficiencies within a fourfold range (Fig. 1g). The means by which insertion barcodes were chosen for ENGRAM recorders used in experiments throughout this paper are summarized in Supplementary Table 6.

We suspected that heterogeneity in insertion efficiencies might be a consequence of the influence of the 5-mer sequence on pegRNA

secondary structure. Consistent with this, the least efficient 5-mer is predicted to pair with the spacer sequence to form a more stable secondary structure whereas the most efficient 5-mer insertion is not (Extended Data Fig. 2d,e). To ask whether we could predict insertional bias, we performed linear lasso regression with 84 binary sequence features and one secondary structural feature (minimum free energy, MFE). The resulting model was reasonably accurate, with MFE emerging as the most predictive feature (Fig. 1h and Extended Data Fig. 2f,g).

## Recording of enhancer activities

To assess whether ENGRAM could record CRE activity, we first evaluated a pair of 170 bp CRE fragments previously exhibiting high versus low enhancing activity in a massively parallel reporter assay (MPRA)[30] conducted in K562 cells, together with minP-only and no-promoter controls; each of these four constructs was linked to two distinct 5 bp insertions (Extended Data Fig. 3a). An equimolar mixture of these eight recorders was introduced into PE2(+) K562 cells via piggyBac in triplicate. At 5 days post transfection, only 2.8% of endogenous *HEK3* sites were edited but 88% of 5 bp insertions there were associated with the high-activity CRE fragment (Extended Data Fig. 3b). Of note, the 15.1-fold difference in DNA-based recording between high- versus low-activity CRE fragments matched a 15-fold difference as measured by MPRA[30].

Next we cloned 300 CRE fragments[30] to the ENGRAM construct, each driving a pegRNA encoding a unique 6 bp insertion and targeting *HEK3* (Fig. 2a,b and Supplementary Table 2). We introduced these recorders to PE2(+), synthetic *HEK3*(+) K562 cells via piggyBac in triplicate. At 5 days post transfection we separately amplified and sequenced barcodes from endogenous *HEK3* (DNA, two or three copies), synthetic *HEK3* (DNA, around 20 copies) or pegRNA transcripts (RNA). Although we observed a modest difference in efficiency in writing to endogenous versus synthetic DNA Tape, probably secondary to chromatin environment[31], the relative activities of individual CREs were highly correlated between these site classes (Extended Data Fig. 3c,d).

Both RNA- and DNA-based measurements were highly reproducible between integration replicates (Extended Data Fig. 3e,f). Furthermore, we observed a strong correlation between directly measured (MPRA, RNA) versus recorded (ENGRAM, DNA) activities (Fig. 2c and Extended Data Fig. 3g). This relationship was broadly maintained at the level of relative rank, overall, as well as across quartiles of activity (Fig. 2d and Extended Data Fig. 4a). Even within activity quartiles (including the lowest bin), the rank orders of randomly selected subsets of CREs exhibited reasonable correlations for MPRA versus ENGRAM (Fig. 2e).

To assess robustness we recorded CRE activity while varying the number of input cells. With synthetic DNA Tape (roughly 20 copies per cell) we could reproducibly record the relative activities of CREs from as few as 12,000 cells (291 out of 300 detected; mean Pearson's correlation of 0.87 between integration replicates; Extended Data Fig. 4b,c). By downsampling the number of reads used in association with the 96,000 cell input condition, we found that 250,000 reads were sufficient (278 of 300 detected; mean Pearson's correlation of 0.87 between integration replicates; Extended Data Fig. 4d,e).

*Cis*-regulatory activity is mediated by transcription factors binding to their cognate motifs. We next sought to design synthetic CREs that capture the differential propensity of individual transcription factor motifs to enhance transcriptional activation across mammalian cell types. We designed and synthesized a library of synthetic CREs, each bearing a homotypic array of a known transcription factor motif. To minimize redundancy we clustered 841 vertebrate motifs by similarity[32,33] and then manually curated these to a set of 98 mammalian motifs (6–20 bp), each representing one transcription factor or transcription factor family (Supplementary Table 3). We then obtained synthetic CREs, each composed of a homotypic array of one of these motifs[30]. The synthetic CREs were cloned into the ENGRAM construct, each upstream of minP, followed by a *HEK3*-targeting pegRNA encoding a CRE-specific 5-mer

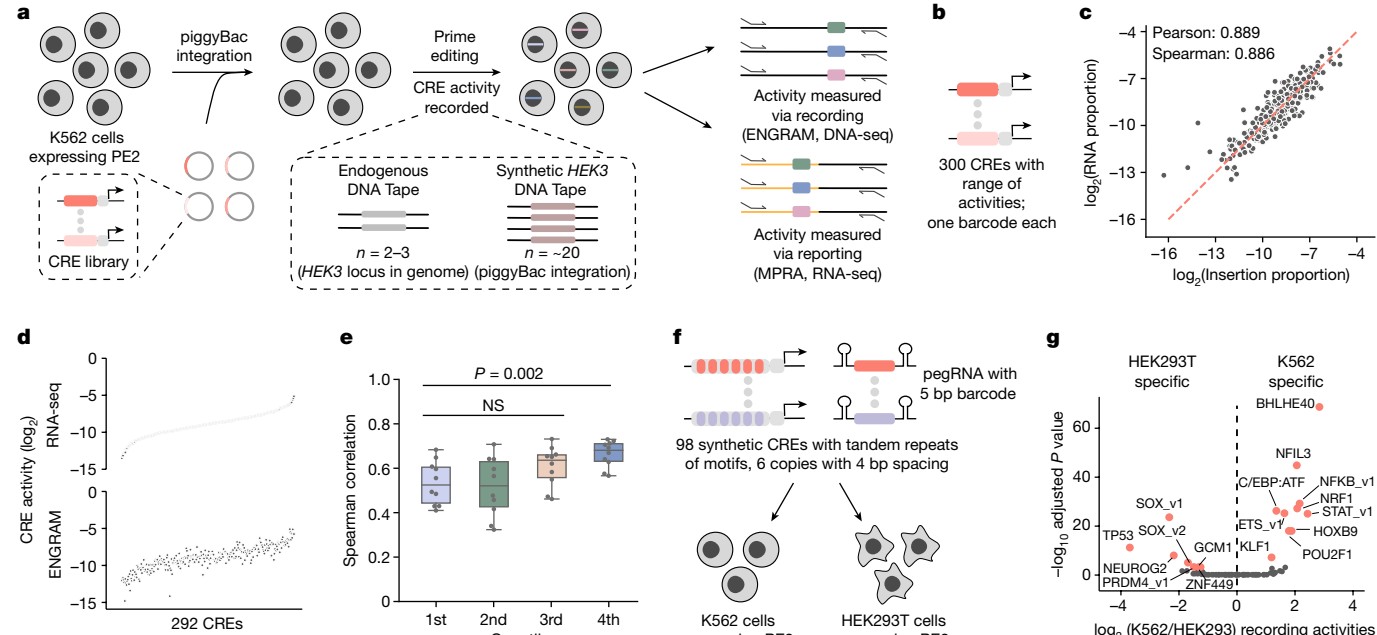

**Fig. 2 | Multiplex recording of CRE activities with ENGRAM. a**, A library of ENGRAM reporters bearing various CREs was constructed and integrated into PE2(+) K562 cells. CRE activity was recorded at an endogenous or synthetic (piggyBac) DNA Tape. For benchmarking, relative activities of CREs were measured via either recording (ENGRAM) or reporting (MPRA). **b**, Each of 300 CREs was linked to a distinct pegRNA-encoded 6-mer insertional barcode. **c**, ENGRAM-recorded barcode proportions were highly correlated with MPRA-reported barcode proportions. Correction of ENGRAM-recorded proportions by MFE of corresponding pegRNAs did not markedly alter correlation ($r$ = 0.860 versus 0.889 with versus without MFE correction). **d**, ENGRAM preserves overall rank order of CRE activity reasonably well. Top, CREs ranked by MPRA-reported activity; bottom, ENGRAM-recorded activities plotted in the same order. **e**, Boxplot of Spearman correlations within each quartile of CRE activity. CREs were split into four quartiles based on MPRA-reported activities. Within each quartile, 20 CREs were randomly sampled and their rank order compared for MPRA versus ENGRAM. Points represent sampling iterations ($n$ = 10), boxes represent 25th, 50th and 75th percentiles, whiskers represent 1.5× interquartile range. $P$ values derived from two-tailed $t$-test. **f,g**, ENGRAM recording of cell-type-specific activities of 98 synthetic CREs. **f**, Design of synthetic CREs. Each synthetic CRE is homotypic, bearing tandem copies of one transcription factor binding site motif, and is linked to a pegRNA encoding a 5-mer insertional barcode. The recorder library was transiently transfected to PE2(+) K562 or HEK293T cells in triplicate. Genomic DNA was harvested 48 h later, followed by PCR and sequencing. **g**, Volcano plot of differentially recorded activity in K562 versus HEK293T cells. Red points indicate significant and substantial differences (Wald test with Benjamini–Hochberg correction, $P$ < 0.001 for fold difference above 2). Labels correspond to names of transcription factor representatives for synthetic CRE motifs (Supplementary Table 3). NS, not significant.

insertion (Fig. 2f). The recorder library was transiently transfected into two PE2(+) cell lines (HEK293T and K562) in triplicate, and cells obtained 2 days after transfection. Following sequencing of the endogenous *HEK3* site, we observed 12.6 and 1.0% editing of *HEK3* in HEK293T and K562 cells, respectively (Extended Data Fig. 5a). Recording levels for individual CREs were highly reproducible (Extended Data Fig. 5b,c).

Of the 98 synthetic CREs, 15 exhibited significant and substantial differences in recorded activity between the two cell lines (Wald test with Benjamini–Hochberg correction, $P$ < 0.001 for a fold difference above 2; Fig. 2g). Most differential recording was directionally concordant with the summed expression of transcription factors assigned by the JASPAR database to the transcription factor motif of a given synthetic CRE recorder (13 out of 17, $P$ = 0.02, binomial test; Extended Data Fig. 5d,e). However, these coincidences should be interpreted with caution because we have not confirmed that these transcription factors are binding to the corresponding synthetic CREs in these cell lines. A further caution is that these differences between cell lines in recorded activity are quantified relative to the other synthetic CRE recorders, rather than in terms of absolute activity.

## Recording of signalling activities

We next sought to apply ENGRAM to record the intensity or duration of signalling pathway activation. We selected several signal-responsive regulatory elements: the tetracycline (Tet) response element (TRE, activated by doxycycline—that is, Tet-On)[34], an NF-κB response element

(activated by TNF)[27] and a TCF-LEF response element (WNT signalling, activated by CHIR99021)[35] (Supplementary Table 4). These CREs were cloned to the ENGRAM construct, each driving expression of a pegRNA encoding one or two signal-specific insertions to endogenous *HEK3* (Fig. 3a). These recorders were separately integrated into PE2(+) HEK293T cells via piggyBac (for the doxycycline recorder, a constitutively expressed reverse tetracycline-controlled transactivator was integrated separately). All cells were cultured for at least 1 week before agonist exposure. A twofold dilution series of either doxycycline, TNF or CHIR99021 was added to the medium of the corresponding recorder cells in triplicate, and gDNA harvested 48 h later. For CHIR99021 we tested additional concentrations in the range 1–4 μM.

For all three recorders, editing rates at *HEK3* exhibited a sigmoidal dependence on the log-scaled concentration of the corresponding agonist (Fig. 3b–d). The WNT recorder showed almost switch-like behaviour across a fourfold range of CHIR99021 concentration (Fig. 3d). As previously, we observed a low level of non-accumulating basal recording even in the absence of agonist (0.1–0.2%; Extended Data Fig. 6a). Nonetheless, the dynamic range in editing efficiency between background versus maximal stimulation was 11.5-, 19.0- and 22.6-fold for the Tet-On, NF-κB and WNT recorders, respectively (Fig. 3e).

To explore the dependence of recording levels not only on signal intensity but also on duration, we performed a matrix experiment on the NF-κB and WNT recorders, varying both agonist concentration and duration of exposure (two recorders × eight concentrations × eight durations (6–48 h) × three integration replicates = 384 conditions;

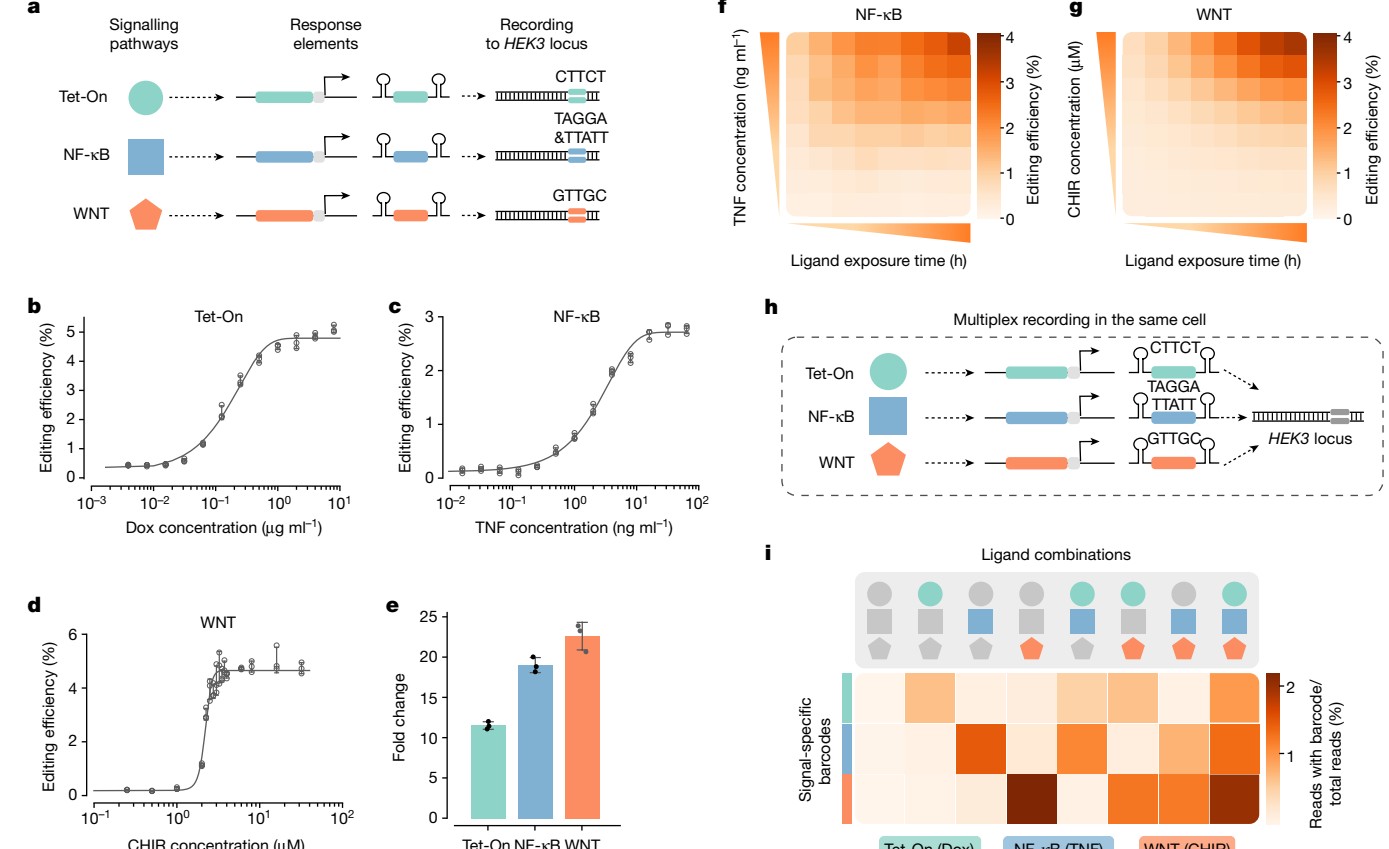

**Fig. 3 | Multiplex recording of intensity and duration of signalling pathway activity. a**, ENGRAM recorders driven by signal-responsive CREs for doxycycline (Tet-On, TRE), TNF (NF-κB response element) and CHIR99021 (TCF-LEF response element, WNT signalling) were constructed. Each recorder was linked to one or two unique barcodes. **b–d**, Recording levels are dependent on agonist concentration. Recorders were integrated to PE2(+) HEK293T cells, which were exposed to a serial twofold dilution series of doxycycline (**b**), TNF (**c**) or CHIR99021 (**d**), with starting concentrations of 8 μg ml⁻¹, 64 ng ml⁻¹ and 32 μM, respectively, for 48 h in triplicate. For CHIR99021, additional concentrations were sampled between 1 and 4 μM. The half-maximal effective concentrations for doxycycline, TNF and CHIR99021 are 0.17 μg ml⁻¹, 2.5 ng ml⁻¹ and 2.2 μM, respectively. Data were fitted to sigmoid curves using nonlinear regression. **e**, Fold difference in editing levels observed for the three signalling pathway

recorders with versus without the maximum dose of the corresponding agonist. Data in **b–e** are mean and s.d. from $n = 3$ integration replicates. **f,g**, Heatmap showing editing efficiencies observed in matrix experiments on NF-κB (**f**) and WNT (**g**) recorders in which both agonist concentration and exposure duration were varied. **h**, Schematic of multiplex recording of signalling pathway activities. The three recorders shown in **a** were mixed at an equimolar ratio and integrated to PE2(+) HEK293T cells. The recorders write different barcodes to the same DNA Tape (endogenous *HEK3*). **i**, These cells were exposed to all possible combinations of three agonists for 48 h, followed by sequencing-based measurement of recording levels based on signal-specific barcodes written to *HEK3*. Coloured shapes as in **a**. Concentrations used were 500 ng ml⁻¹, 10 ng ml⁻¹ and 3 μM for doxycycline, TNF and CHIR99021, respectively.

Fig. 3f,g). In this experiment, each batch of cells was harvested 24 h following the removal of agonists from media. In sequencing DNA Tape we observed recording levels to be a function of both intensity and duration (Fig. 3f,g). For both recorders, 6 h of stimulation was sufficient to observe signal in excess of background. However, the NF-κB recorder appeared to exhibit faster kinetics than the WNT recorder (Extended Data Fig. 6b,c).

We also tested whether these recorders could be concurrently deployed to record multiple signals to a shared DNA Tape (Fig. 3h). The Tet-On, NF-κB and WNT recorders were mixed at an equimolar ratio and integrated to PE2(+) HEK293T cells. Although we did not construct a monoclonal line or explicitly confirm that individual cells contained copies of all three recorders, the conditions were estimated to yield 15–20 integrations per cell such that the vast majority of cells should contain at least one copy of each recorder (Extended Data Fig. 6d,e). Cells were exposed to a high concentration of all possible combinations of between zero and three agonists ($2^3$, or eight, combinations × three integration replicates = 24 conditions). After 48 h of stimulation and sequencing endogenous *HEK3*, we found abundances of signal-specific barcodes to be highly dependent on the combination of stimuli applied

(Fig. 3i). Put another way, we observed minimal cross-talk, consistent with the notion that these signalling pathways are orthogonal to one another (Extended Data Fig. 6f).

To further evaluate multiplex signal recording, we performed a separate experiment in which subsamples of a population of cells bearing all three recorders were exposed to all possible combinations of low, medium or high concentrations of each agonist (three agonists^(three concentrations) × three integration replicates = 81 conditions). Again we found that signal-specific barcode abundances were informative with respect to the strength of the corresponding stimuli (Extended Data Fig. 6g,h), lending additional support to the conclusion that these recorders are able to capture quantitative information in multiple channels while writing to a shared DNA Tape.

## Combining ENGRAM and DNA Typewriter

Thus far we have shown that ENGRAM enables multiplex, quantitative recording of the activities of signal-responsive CREs to a shared DNA Tape. However, the limitations of ENGRAM on its own include an inability to distinguish between intensity versus duration, and an inability to

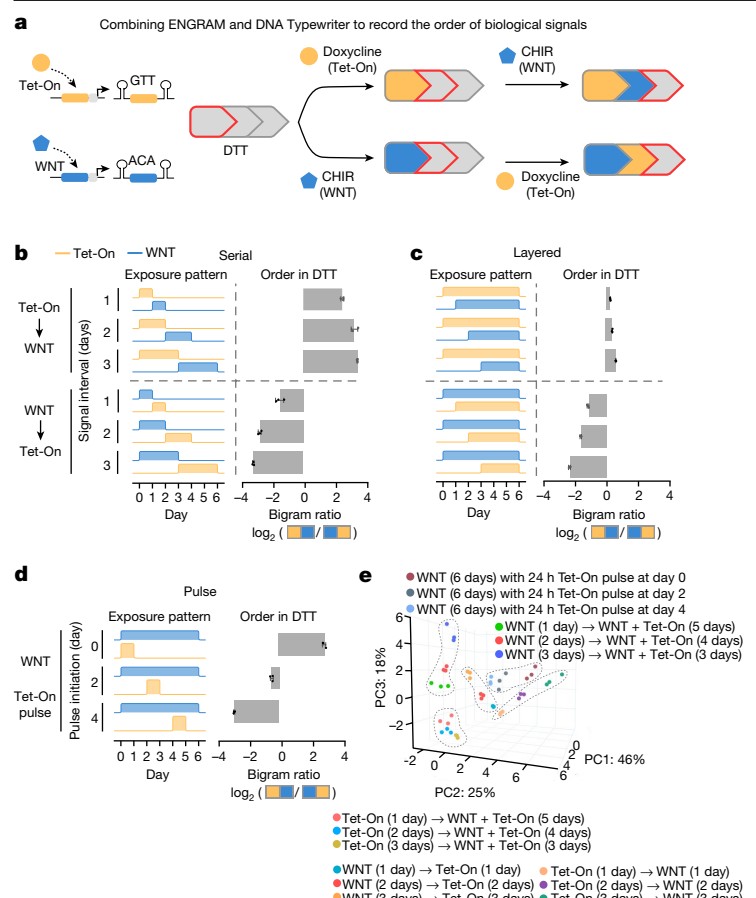

**a**  Combining ENGRAM and DNA Typewriter to record the order of biological signals

**Fig. 4 | Combining ENGRAM and DNA Typewriter. a**, Tet-On (orange) and WNT (blue) ENGRAM recorders were modified to drive the expression of pegRNAs that write to DTT. With DNA Typewriter, insertional edits include a barcode but also a key that shifts the type guide position to the next unit of the DTT. Temporal dynamics (for example, the order of two serially applied agonists) should be captured by the order in which the corresponding symbols appear in DTT. **b,c**, Modified ENGRAM recorders and five-unit DTT were sequentially integrated to PEmax⁺ HEK293T cells. We designed and tested serial (**b**) and layered (**c**) programmes in which cells were exposed to different patterns of either 100 ng ml⁻¹ doxycycline or 3 μM CHIR99021 (left-hand columns) across (two patterns × three intervals × two possible orders × three integration replicates = 36 cell populations). Using sequencing DTT after 6 days, we calculated the log ratio of (Tet-On → WNT) versus (WNT → Tet-On) bigrams at sites 1 and 2, predicting and observing positive values when Tet-On activation preceded WNT activation, and negative values when WNT activation preceded Tet-On activation (right-hand columns). **d**, As in **b,c**, but for pulse programmes (three pulse timings × three integration replicates) in which these cells were exposed to 500 ng ml⁻¹ doxycycline for 24 h against a background of continuous 3 μM CHIR99021 stimulation. Data in **b–d** are mean and s.d. from $n = 3$ integration replicates. **e**, PCA on proportions of unigrams and bigrams observed at each of five DTT positions and four DTT position-pairs, respectively, across 45 cell populations subjected to various patterns of exposure to doxycycline and CHIR99021 (15 programmes, executed in triplicate). Circled subsets correspond to serial and layered programmes in either order, or to pulse programmes. The top three PCs are plotted, collectively explaining 90% of variance.

explicitly capture the temporal order in which multiple signals occur. In other words, we are successfully quantifying the 'integral' of biological signal(s) but failing to capture their temporal dynamics.

To address this we sought to combine ENGRAM with DNA Typewriter, a related method that we recently developed for sequential genome editing and cell lineage tracing[1]. In DNA Typewriter, genomic tape (DTT) consists of a tandem array of partial CRISPR–Cas9 target sites with all but the first truncated at their 5′ ends and therefore inactive. As with ENGRAM, short insertional edits serve as 'symbols' that record the identity of the pegRNAs mediating the edit. However, DNA Typewriter edits contain an additional 'key' that completes the subsequent target site, effectively shifting the position of the editable 'type guide' to the next unit along the DTT. Because both are based on insertional prime editing, combining ENGRAM and DNA Typewriter requires only that some or all of the symbols of a DNA Typewriter be driven by ENGRAM recorders (Fig. 4a).

As a proof of concept, we sought to record the temporal dynamics of two orthogonal signal transduction pathways by combining ENGRAM and DNA Typewriter. Specifically, we designed and cloned Tet-On and WNT recorders encoding pegRNAs targeting DTT (Extended Data Fig. 7a). To minimize background, the recorders and five-unit DTT were sequentially integrated to PEmax⁺ HEK293T cells via piggyBac.

To these cells we applied a set of agonist exposure programmes. For 'serial patterns', agonists were applied sequentially, each for 1, 2 or 3 days (Fig. 4b, left-hand column) whereas for 'layered patterns' the second agonist was introduced 1, 2 or 3 days after the first agonist, which was continued (Fig. 4c, left-hand column). In total we tested 12 programmes (two patterns × three intervals × two possible orders) in triplicate, changing media each day and passaging cells every 2 days during agonist exposure. All samples were harvested after 6 days, gDNA isolated and DTT amplified and sequenced. For DTT in which symbols corresponding to both signal transduction pathways were observed,

we predicted that the order of symbols would inform which agonist was applied first (Fig. 4a).

Most recording occurred at the first two DTT sites, indicating that we had yet to saturate recording capacity (Extended Data Fig. 7b,c). The higher levels of editing in comparison with those from earlier experiments (for example, greater than 60% at the first DTT site in layered programmes) may be due to several factors, including the switch to PEmax (Supplementary Note 1), sorting of this PEmax line for high levels of coexpressed mCherry and the switch from writing to endogenous *HEK3* to writing to DTT embedded in a highly expressed transcript.

To distinguish programmes in which Tet-On preceded WNT activation or vice versa, we calculated the log ratio of read counts bearing (Tet-On → WNT) versus (WNT → Tet-On) bigrams at adjacent sites. For all 12 programmes this bigram ratio correctly showed which agonist was applied first, with a high degree of reproducibility across integration replicates (Fig. 4b,c, right-hand columns, and Extended Data Fig. 7e–g). The data were clearer for serial patterns and longer intervals, but detectable even in layered patterns in which the first agonist was present for only the first of 6 days. Longer signal durations were associated with longer 'homopolymeric' runs of the corresponding symbol (Extended Data Fig. 7i).

Can we discern the timing of a strong burst of activity from one signal transduction pathway superimposed on the continuous activity of a second pathway? As a third class of programme we implemented 'pulse patterns', which used the same PEmax⁺ HEK293T cells bearing five-unit DTT and ENGRAM recorders, introducing a strong 24 h pulse of doxycycline at days 0, 2 or 4, against a backdrop of continuous WNT stimulation (Fig. 4d, left-hand column, and Extended Data Fig. 7d). Because the 'integral' of exposure of each programme to each agonist was identical, we predicted and observed the corresponding symbols occurring at roughly similar rates (Extended Data Fig. 7e). However, the (Tet-On → WNT) versus (WNT → Tet-On) bigram ratios

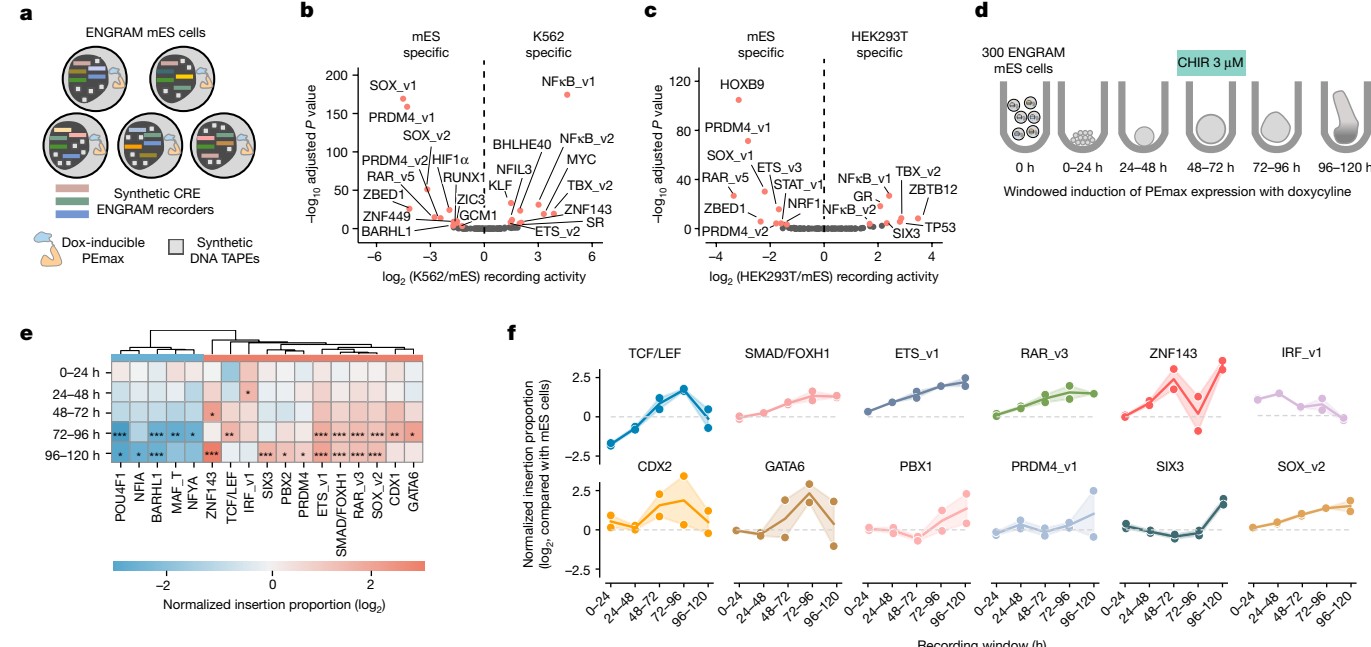

**Fig. 5 | Biologically conditional recording in mES cells and gastruloids.**
**a**, Schematic of polyclonal ENGRAM mES cells, with each cell bearing multiple copies of doxycycline-inducible PEmax, ENGRAM recorders and synthetic DNA Tape (*HEK3*) (Fig. 5a). **b**,**c**, Volcano plot of differential activity of ENGRAM recorders in cultured mES cells versus K562 cells (**b**) and in cultured mES cells versus HEK293T cells (**c**). Red points indicate significant and substantial differences (Wald test with Benjamini–Hochberg correction, *P* < 0.001 for fold difference above 2). Labels correspond to names of transcription factor representatives for synthetic CRE motifs (Supplementary Table 3). mES cell data were corrected for relative abundance of recorders in the polyclonal mES cell line versus the plasmid pool for transient transfection of K562 and HEK293T cells (Extended Data Fig. 9b). **d**, Polyclonal ENGRAM mES cells were differentiated to gastruloids. For each of the five 24-h windows, PEmax was activated by the addition of 50 ng ml⁻¹ doxycycline; gastruloids were harvested 24 h later.

Each recording window was tested in duplicate. **e**, Hierarchically clustered heatmap showing recorded activities across each 24 h interval (rows) for 17 of the 98 ENGRAM recorders exhibiting significant and substantial differences (Wald test with Benjamini–Hochberg correction, *P* < 0.1 for fold difference above 2) in one or more of the five windows (columns) relative to cultured mES cells. Values are log-scaled barcode proportion ratios. *\*P* < 0.10, *\*\*P* < 0.01 and *\*\*\*P* < 0.001 **f**, Dynamics of selected ENGRAM recorders during gastruloid induction. Labels are representative of transcription factor(s) thought to bind each motif, and it remains uncertain which are driving the activity of each synthetic CRE recorder (Supplementary Table 3). Plotted on the *y* axis are the log₂-scaled barcode proportion ratios for gastruloids with windowed recording in a particular 24 h interval (*x* axis) versus cultured mES cells. Dots and line shadow represent two integration replicates and 95% confidence interval, respectively.

grossly differed between the programmes in which the 24 h doxycycline pulse was initiated at day 0, 2 or 4, with strong reproducibility across integration replicates (Fig. 4d, right-hand column, and Extended Data Fig. 7e,h).

In summary, by combining ENGRAM and DNA Typewriter we successfully recorded information about the temporal dynamics of two orthogonal signal transduction pathways to gDNA. In total we tested 15 unique temporal signalling programmes (Fig. 4b–d, left-hand columns). Within each programme class we could distinguish not only different orders but also different timings, solely based on the ratios of bigrams observed in DTT (Fig. 4b–d, right-hand columns). On the other hand, these bigram ratios were not sufficient to distinguish patterns across classes—for example, serial versus layered versus pulse patterns. To assess whether all 15 programmes were distinguishable from one another, we performed principal components analysis (PCA) on variables observed in each of 45 treatments (15 programmes × three integration replicates), evaluating three strategies for encoding the ensemble of patterns observed (Extended Data Fig. 8a,b). For the optimal strategy, the 26 variables included the proportions of symbols at each DTT site (two symbols × five positions) and the proportions of each possible bigram at pairs of adjacent sites (four bigrams × four position-pairs). The top three PCs together explained 90% of the variance, and integration replicates tightly clustered (Fig. 4e). This suggests that the 15 temporal signalling programmes give rise to reproducibly distinct ensembles of symbol patterns recorded to five-unit DTT. Consistent with this, a random forest classifier was able to predict from which of

the 15 programmes an unseen set of sequenced DTT was derived with a mean accuracy of 0.91 (Extended Data Fig. 8c).

## Recording in stem cells and gastruloids

The implementations of ENGRAM described thus far were performed in workhorse cancer cell lines (HEK293T and K562 cells). Because our ultimate goal is biologically conditional in vivo recording, we next sought to validate ENGRAM in mouse embryonic stem (mES) cells, which can readily be differentiated into a diversity of organoid models or used to make mice.

We constructed a polyclonal mES cell line harbouring doxycycline-inducible PEmax, an ENGRAM recorder library driven by the previously described 98 synthetic transcription factor-motif CREs (Fig. 2f and Supplementary Table 3) and synthetic DNA Tape (*HEK3*) (Fig. 5a). To minimize background, PEmax and ENGRAM recorders were integrated first, followed by synthetic DNA Tape in a separate round, all via piggyBac at high multiplicity of infection (MOI). We estimated that individual cells in the polyclonal mES cell line would bear an average of five copies of doxycycline-inducible PEmax, 10 of the 98 ENGRAM recorders and ten of synthetic DNA Tape.

We cultured mES cells over 2 days then compared the resulting measurements with those made from the same recorders in K562 and HEK293T cells (Fig. 2g). The mES cell-recorded activities were highly reproducible (Extended Data Fig. 9a). Of the 98 recorders, 22 (mES versus K562 cells) and 16 (mES versus HEK293T cells) exhibited significant

and substantial differences between pairs of cell lines (Wald test with Benjamini–Hochberg correction, $P < 0.001$ for a fold difference above 2; Fig. 5b,c and Extended Data Fig. 9b). These included five recorders consistently more active in mES cells, among which were those for SOX, PRDM4, retinoic acid response element (RARE) and ZBED1 motifs, and two recorders consistently less active in mES cells, specifically those for TBX and NF-κB motifs. Cell-type-specific differences were consistent across independent pairs of integration/transfection replicates (Extended Data Fig. 9c–e), and recording events were contributed by nearly all recorders in all samples tested (Extended Data Fig. 9f). However, we reiterate that, although the motifs homotypically embedded in the synthetic CRE recorders are associated with these transcription factors or transcription factor families in the literature, it remains uncertain which specific transcription factor(s) are driving their activity in these cell lines.

To introduce dynamics, we differentiated these mES cells into gastruloids. Gastruloids are a stem cell-derived organoid model that mimics aspects of early mammalian development including germ layer specification, symmetry breaking and axial organization[36–38]. A conventional protocol for mouse gastruloid induction begins with the aggregation of around 300 mES cells, which are then subjected to 24 h of WNT stimulation at 48–72 h post aggregation. At 120 h post aggregation, most mouse gastruloids are elongated and include cell types derived from all germ layers.

In a series of experiments in which ENGRAM-bearing mES cells were differentiated to gastruloids, we varied the recording window by the addition of doxycycline for a specific 24 h window of a 5 day time-course (Fig. 5d and Extended Data Fig. 10a). Approximately 20–50 aggregates/gastruloids per time window were processed per integration replicate; 8–15% of sequenced DNA Tapes were edited, showing that PEmax and ENGRAM are active in differentiating gastruloids, albeit trending downwards with time (Extended Data Fig. 10b). As with mES cells, recording levels in differentiating gastruloids were highly reproducible (Extended Data Fig. 10c).

To evaluate dynamics we compared recording activity for each synthetic CRE, as integrated across each 24 h post-induction window, against recorded activity in mES cells (Fig. 5d). Seventeen of 98 recorders exhibited significant and substantial differences during one or more of the 24 h differentiation windows as compared with cultured mES cells (Fig. 5e; Wald test with Benjamini–Hochberg correction, $P < 0.1$ for a fold difference over 2; Extended Data Fig. 10d,e and Supplementary Table 3). Of these, 12 showed increased activity (IRF_v1, ZNF143, RAR_v3, ETS_v1, FOXH1/SMAD, SOX_v2, TCF/LEF, CDX1, GATA6, SIX3, PBX2 and PRDM4_v1) and five decreased activity (BARHL1, POU4F1, MAF_T, NFYA and NFIA).

Of note, the dynamics of four major developmental signalling pathways—WNT, Nodal, FGF and retinoic acid—are potentially probed by subsets of the 98 recorders bearing motifs for their effector transcription factors (Supplementary Table 3). Recorders driven by arrays of TCF/LEF (WNT), SMAD/FOXH1 (Nodal), ETS (one of three versions) (FGF) and RARE (one of five versions) (retinoic acid) consensus motifs were significantly dynamic, rising monotonically with each successive interval, but with activity of the TCF/LEF recorder dropping and that of the SMAD/FOXH1 recorder plateauing in the final 24 h (Fig. 5f). Other recorders also exhibited dynamic behaviour (Fig. 5f and Supplementary Table 3). Although some of these transcription factor motifs are associated with core developmental transcription factors or transcription factor families, we reiterate that it remains ambiguous which specific transcription factor(s) are driving the dynamic activity of these recorders in differentiation of gastruloids. A further caution is that the observed dynamics are relative to all other recorders in this panel, rather than in terms of absolute activity.

As expected, the number of recorders capturing differential activity increased over time, with nearly all significant differences from cultured mES cells accruing in either the 72–96- or 96–120-h interval,

coincident with cell type diversification[39] (Fig. 5e). Because we are assaying the DNA Tape from bulk gDNA, we are accessing only the average recorded activity across all cells and cell types in differentiated gastruloids, integrated across a particular 24 h window. However, we presume that the many of these differences would be even more pronounced if we could distinguish the 'cell type of origin' of each sequenced DNA Tape. Looking forward, we anticipate this will be possible by coassaying transcriptionally expressed DNA Tape with single-cell RNA-seq, as we have recently done with DNA Typewriter[1].

## Discussion

Here we describe ENGRAM, a multiplex strategy for biologically conditional genomic recording in which signal-specific CREs drive the insertion of signal-specific barcodes to a common DNA Tape. Because gDNA is stable and is also passed to daughter cells, signals recorded to DNA Tape can be read out at a subsequent point in time, in ths case by gDNA sequencing but potentially also by single-cell RNA-seq[1] or DNA/RNA FISH[40,41].

Although both are driven by CREs, ENGRAM is a recorder assay in which measurements are written to DNA, and an MPRA is a reporter assay in which measurements are made from RNA. A first corollary is that ENGRAM can be used to measure the past state of cells whereas MPRAs can be used only to measure their present state. For example, with ENGRAM but not MPRAs, one could ask how the endpoint phenotypes of individual gastruloids correlate with the signalling histories of their constituent cells. A second corollary is that ENGRAM can be used to integrate activity over time whereas MPRAs measure activity only at the endpoint. This aspect of ENGRAM may be particularly useful for capturing transient aspects of *cis*-regulation[42,43].

ENGRAM's capacity for multiplexing follows from the use of short insertions to represent each signal. With the 5–6 bp insertions used here, between 1,024 and 4,096 unique biological signals could theoretically be recorded within the same cell, all competing to write to a shared DNA Tape. The advantage of a shared recording medium for all signals of interest is particularly manifest in the combination of ENGRAM and DNA Typewriter[1]. As with written language, a linear increase in the number of signals/symbols results in an exponential increase in the number of potential signal/symbol orders. However, in contrast to other CRISPR-based signal recording systems[14,15], the combination of ENGRAM and DNA Typewriter does not require each possible signal order to be precoded as a distinct template; instead, all possible orders can be written to the shared DTT.

Of note, our 98 synthetic CREs were unoptimized designs consisting of homotypic arrays of representative transcription factor motifs. Although 46 of 98 exhibited reproducible, substantial and significantly differential recording activity in at least one comparison, we cannot definitively assign these differences to specific transcription factors. However, we predict that as efforts to devise and optimize synthetic CREs for specific transcription factors, signalling pathways and cell types advance[44,45], such elements can be leveraged by ENGRAM to record their activities both in vitro and in vivo.

Several challenges remain. First, like MPRAs, because ENGRAM relies on CRE-mediated enhancement of Pol2 transcription, it is not well suited to biological signals or states that are not readily coupled to CREs, nor to recording at fast timescales. These challenges could be addressed in part by heterologous signal conversion (for example, Tet-On) or using entirely different strategies for biologically conditional prime editing[46]. Second, although in principle thousands of ENGRAM recorders could be stably deployed within a single cell or organism, this is challenging to achieve with random integration. However, as larger numbers of biologically conditional recorders are validated, these could potentially be consolidated to a single 'recorder locus', which could then serve as a common reagent for multiplex recording. Third, the deconvolution of ENGRAM signals, particularly

when coupled to DNA Typewriter, will pose new algorithmic challenges (Fig. 4e).

One can imagine variants of ENGRAM—for example, integration of a minimal *csy4*-pegRNA-*csy4* cassette to endogenous gene bodies—with the goal of recording endogenous gene expression levels to DNA Tape. Our initial attempts at achieving exactly this leveraged random integration of a T7-mappable version[31] of a minimal ENGRAM cassette, but failed in that the barcode proportions observed in DNA Tape did not correlate with the expression levels of genes in which pegRNAs encoding those barcodes resided (Supplementary Note 2). Possible explanations for this negative result include (1) the dominance of barcodes derived from pegRNAs that happened to integrate within highly transcribed ribosomal gDNA regions (confounding dynamic range), (2) the short life and nuclear location of the pre-mRNA (confounding intronic integrations) and/or (3) cryptic splicing sites within the minimal ENGRAM cassette (confounding exonic integrations). Further work is required to adapt ENGRAM to quantitatively record endogenous gene expression levels.

In summary, ENGRAM enables quantitative, multichannel, DNA-based recording of biological signals. In an ideal future we envision that hundreds to thousands of biological signals could be coupled to the ordered writing of signal-specific insertions ('symbols') to DTT(s), either by ENGRAM or other modes of biologically conditional editing. A further set of non-specific symbols, stochastically written to the same Tape(s), would facilitate the capture of cell lineage[1]. All components would be genomically encoded by a recorder locus within the millions to billions of cells of a model organism, capturing biology as it unfolds over time, and collectively read out at a single endpoint. Provided that such a system behaved reproducibly, it would facilitate the comparison of signalling and lineage histories across cells, tissues and individuals, in correlation with genetic background and/or environmental history[40]. Furthermore, analogous to a flight recorder, the recovery of past biological states and their correlation to future outcomes within the very same cells might facilitate causal inference.

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

## Methods

### Molecular cloning

Sequences of the 300 native CREs, 98 synthetic CREs (motifs), three signal-responsive elements and primers/oligos used in this paper are listed in Supplementary Tables 2, 3, 4 and 5, respectively. Three hundred native CREs were picked from ref. 30 with a wide range of activities. The set of 98 synthetic CREs were generated in two steps; first, 841 vertebrate motifs in the JASPAR database were clustered by similarity[32,33] and then manually curated to a set of 98 mammalian motifs (6–20 bp), each representing a single transcription factor or transcription factor family. Second, to design the 98 synthetic CREs, six homotypical motif repeats separated by 4 bp spacers were embedded in an inactive DNA sequence[30]. The TRE, consisting of seven modified tet operator sequences (tetO, 19 bp), was obtained from the Tet-On system (Takara). The sequence of the NF-κB response element was obtained by embedding six homotypic NF-κB motifs separated by 2 bp spacers into an inactive DNA sequence[30]. The TCF-LEF response element was obtained from the TCF-LEF reporter (Promega)[35].

All PCR and digestion purifications were performed with AMPure XP beads (0.6× for plasmids and 1.2× for fragments of base pair size 200–300) using the manufacturer's protocol unless otherwise specified. All ligation reactions used Quick ligase (NEB) with a vector:insert molar ratio of 1:6 unless otherwise specified. All Gibson reactions used NEBuilder (NEB) with a vector:insert molar ratio of 1:6 unless otherwise specified. All plasmid DNA for mammalian cell experiments was prepared using the ZymoPURE II Plasmid Kit.

The pegRNA-5N recorders were cloned in two steps. First, a gene fragment containing CTT pegRNA (Addgene, 132778) was PCR amplified using primer sets, adding a 5 bp degenerate barcode and flanking BsmBI site for use in downstream cloning steps. A carrier plasmid containing two BsmBI sites and two csy4 hairpins was ordered from Twist Bioscience. The carrier plasmid and PCR product from the previous step were digested with BsmBI (NEB, buffer 3.1) at 55 °C for 1 h and then purified for ligation. The complete pegRNA with 5N degenerate barcode and csy4 hairpins was PCR amplified from the ligation product. The ENGRAM plasmid and PCR product from the previous step were digested with BsmBI (NEB) at 55 °C for 1 h. Ligation products were purified and resuspended with 5 μl of water for electroporation, which was performed using NEB 10-beta Electrocompetent E. coli (C3020) with the manufacturer's protocol. Transformed cells were cultured at 30 °C overnight.

The libraries of 300 CREs, 98 synthetic CREs and plasmids bearing signal-responsive elements were cloned in two steps. First, a library of DNA oligonucleotides containing CREs, two BsmBI restriction sites, DNA insertion barcode, the 3′ end of pegRNA and the csy4 hairpin were ordered as oPools from IDT. The 5′ ENGRAM recorder was digested with XbaI and NcoI (NEB) at 37 °C for 1 h and purified. DNA oligonucleotides were first amplified with primers to add Gibson overhangs and then cloned into the 5′ ENGRAM recorder using Gibson assembly. Second, a gene fragment containing minP, csy4 hairpin, HEK3 spacer sequence and pegRNA backbone flanked with two BsmBI sites was ordered as a gBlock from IDT. The gBlock and plasmid constructed from the first step containing BsmBI restriction sites were digested with BsmBI (NEB, buffer 3.1) at 55 °C for 1 h to generate compatible sticky ends, and were purified for ligation. Ligation products were transformed into Stable Competent E. coli (NEB, C3040). Transformed cells were cultured at 30 °C overnight.

The synHEK3-TAPE construct was cloned in two steps. First, piggyBac-CMV-MCS-EF1α-Puro plasmid was digested with BsiWI and SphI to remove core insulators and selection markers from piggyBac transposon long terminal repeats. A gBlock (IDT), consisting of a flanking sequence (part of green fluorescent protein (GFP)) and two divergent BsmBI restriction sites, was cloned to the BsiWI- and SphI-digested piggyBac plasmid using Gibson assembly (NEBuilder, NEB) to create

a shuttle vector. Second, a 87 bp region around the HEK3 locus was synthesized (IDT) and amplified with a pair of primers to introduce the T7 promoter and a 16 bp barcode to the 5′ and 3′ end, respectively. The resulting PCR product was purified and cloned into the construct from step 1 (digested with BsmBI) using Gibson assembly. Assembled products were purified and resuspended in 5 μl of water for electroporation, which was performed using NEB 10-beta Electrocompetent E. coli (C3020) following the manufacturer's protocol. Transformed cells were cultured at 30 °C overnight.

The DTT, consisting of five recording units, was previously cloned[1]. Signal-responsive ENGRAM recorders targeting DTT were generated by replacing HEK3-targeting pegRNAs with DTT-targeting pegRNAs. In brief, signal-responsive ENGRAM recorders were subjected to digestion by NcoI and AgeII to remove HEK3-targeting pegRNAs. DTT-targeting pegRNAs with Gibson overhangs were ordered as gBlocks (IDT). Assembled products were transformed into Stable Competent E. coli (NEB, C3040). Transformed cells were cultured at 30 °C overnight.

### Cell culture, transient transfections, nucleofection and piggyBac integrations

HEK293T (CRL-11268) and K562 cells (CCL-243) were purchased from ATCC. CF-1 MEF (ASF-1216) feeder cells were purchased from Applied StemCell. Mouse ES cells (E14TG2a) were a gift from C. Schröter. HEK293T cells and MEFs were cultured in DMEM, high glucose (Gibco). K562 cells were cultured in RPMI 1640 medium (Gibco). All media were supplemented with 10% fetal bovine serum (Hyclone) and 1% penicillin/streptomycin (Gibco). MEF medium was supplemented with additional 1× GlutMAX (Gibco). Normal mES cells were cultured in Ndiff 227 medium (Takara) supplemented with 3 μM CHIR99021 (Selleck, S2924), 1 μM PD0325901 (Selleck, S1036), 1,000 units of ESGRO recombinant mouse LIF protein (Sigma-Aldrich, ESG1107) and 1% penicillin/streptomycin (2i + LIF medium). For culture of both MEFs and mES cells, wells in the culture plates were coated with 0.1% gelatin (Sigma, G1393) in an incubator at 37 °C for 60 min. Cells were grown with 5% $CO_2$ at 37 °C.

Transfection of HEK293T, K562 and mES cells was performed using Lipofectamine 3000 (ThermoFisher, L3000015), a Lonza 4D-Nucleofector and Lipofectamine 2000 (ThermoFisher, 11668019), respectively, following the manufacturer's protocol.

For transfection of HEK293T cells, $1 × 10^5$ cells were seeded on a 24-well plate 1 day before transfection; 500 ng of plasmid (prime editor plasmid, pegRNA plasmid or a mixture of both, with a mass ratio of 1:4) was used for transient transfections; 500 ng of cargo plasmid (prime editor plasmid, ENGRAM pegRNA plasmid or DTT) and 100 ng of Super piggyBac transposase expression vector (SBI) were used for piggyBac integrations. PE2(+) HEK293T cells were picked by sorting single cells to a 96-well plate, followed by selection with 1 μg ml⁻¹ puromycin dihydrochloride (Gibco) and prime editing efficiency verification. Single-cell-derived PEmax(+) HEK293T cells were obtained using the same approach and were then used in recording with DNA Typewriter, whereas PE2(+) cells were used in all other recording experiments relying on HEK293T cells. For nucleofection of K562 cells, $4 × 10^5$ cells were transfected with either 2 μg of plasmid (prime editor plasmid, pegRNA plasmid or a mixture of both, with a mass ratio of 1:4) for transient transfection or 2 μg of cargo plasmid (prime editor plasmid, synthetic DNA Tape, 300 CRE library or 98 synthetic CRE library) + 400 ng of transposase expression vector for piggyBac integration. All transfections were performed in 16-well strips (20 μl) with programme code FF-120. Single-cell-derived PE2(+) K562 cells were picked by the methods described above.

For transfection in mES cells and construction of the ENGRAM mES cell line, three recording components—Dox-inducible PEmax (TRE-PEmax-mCherry-BlastR), a library of ENGRAM recorders (including all 98 synthetic CREs, driving expression of uniquely barcoded pegRNA) and DNA TAPE bearing the synthetic HEK3 target sequence (synHEK3-TAPE)—were integrated in two steps to minimize background

recording activity. First, 600 ng of TRE-PEmax-mCherry-BlastR plasmid, 3 µg of ENGRAM recorder plasmid and 400 ng transposase expression vector were mixed and transfected into $1 \times 10^6$ mES cells using Lipofectamine 2000. At 24 h post transfection, 8 µg ml$^{-1}$ Blasticidin S HCl (Gibco) was added to the medium for selection of cells with the TRE-PEmax-mCherry-Blast plasmid. Of note, massive cell death was observed about 6 days post transfection, possibly due to the low integration efficiency of the large (over 10 kb) TRE-PEmax-mCherry-Blast plasmid. Polyclonal mES cells bearing Dox-inducible PEmax and ENGRAM recorders were cultured in 2i + LIF Ndiff 227 medium. Second, 600 ng of plasmid encoding the puromycin resistance gene (*PuroR*), 3 µg of plasmid bearing synHEK3-TAPE and 400 ng of transposase expression vector were mixed and transfected into $1 \times 10^6$ mES cells using Lipofectamine 2000. At 24 h post transfection, 800 ng ml$^{-1}$ puromycin dihydrochloride was added to 2i + LIF Ndiff 227 medium for selection of cells with PuroR plasmid.

### Signal recording with ligands
Doxycycline hyclate (Dox; Sigma, D9891) was reconstituted in PBS to a final concentration of 10 mg ml$^{-1}$. TNF (R&D Systems, 210-TA-020/CF) was reconstituted in 1 ml of PBS to make 20 µg ml$^{-1}$ stock. CHIR99021 (Selleck, S2924) was purchased as 10 mM stock (1 ml in DMSO). All ligands were stored at −20 °C, thawed immediately before use and diluted with the appropriate culture medium. Concentrations tested here fall within the range in which these agonists are typically used[36–38].

For ligand-recording experiments, $1 \times 10^5$ PE2(+) HEK293T cells were seeded on a 48-well plate 6 h before treatment then 1 ml of medium with ligand or negative control was added to each well. For the time-series experiment, cells were washed with warm medium and harvested 24 h following ligand removal. The same volume of DMSO or PBS was added to the medium as a negative control. Cells were split in a 1:5 ratio every 2 days and medium was changed every day.

For sequential editing with DNA Typewriter, $1 \times 10^5$ PEmax(+) HEK293T cells were seeded on a 48-well plate 6 h before treatment then 1 ml of medium with 100 ng ml$^{-1}$ doxycycline or 3 µM CHIR99021 was added to each well. Cells were split in a 1:5 ratio every 2 days and medium was changed every day. Cells were harvested on day 6 of the experiment.

### Gastruloid induction and recording
Mouse gastruloids were induced using a published protocol[36,37]. In brief, 100,000 ENGRAM mES cells were seeded on a gelatin-coated, six-well plate and cultured in 2i + LIF Ndiff medium for 2 days, which produced a more homogenous starting population for gastruloid induction. To start induction, cells were dissociated with TrypLE Express Enzyme (Gibco) at 37 °C for 4 min to create a single-cell suspension. Cells were counted and diluted in Ndiff medium to a concentration of 6,000–7,000 ml$^{-1}$, and 300–350 then seeded to a 96-well, U-shaped-bottom microplate (Nunclon Sphera, treated, Thermo, 174929) with 50 µl of Ndiff medium. The medium was changed every day, and 3 µM CHIR99021 was added briefly from 48–72 h following aggregation. Windowed recording was activated by the addition of 50 ng ml$^{-1}$ doxycycline for 24 h. Gastruloids were harvested for sequencing 24 h post activation.

### Recovery of recorded information from DNA Tape and DTT
Genomic DNA was extracted using a previously described protocol[22]. In brief, cells were washed once with PBS and lysed with freshly prepared lysis buffer (10 mM Tris-HCl pH 7.5, 0.05% SDS and 25 µg ml$^{-1}$ proteinase K (ThermoFisher, EO0492)) to a final concentration of 5,000 cells µl$^{-1}$. The lysate was incubated at 50 °C for 1 h, followed by an 80 °C enzyme inactivation step for 30 min.

For retrieval of information recorded to various kinds of DNA Tape (including the endogenous *HEK3* locus, the synthetic *HEK3* locus integrated into the genome and the DTT integrated into the genome), the target region in gDNA was amplified with two-step PCR (KAPA2G Robust HotStart ReadyMix) and sequenced on an Illumina sequencing platform. The first PCR reaction included 2 µl of cell lysate and 0.5 µM forward and reverse primer with a final reaction volume of 50 µl. The number of PCR reactions required for each sample depends on the complexity of the recorded signal, because more complex recording patterns would require more reactions to capture the full diversity of edits. We typically aimed to PCR amplify at least 2,000 DNA Tape-containing amplicon molecules per signal, which is equivalent to 1,000 cells per signal for the endogenous *HEK3* locus or 100 cells for synthetic DNA Tapes such as synHEK3-TAPE or DTT, assuming 20 integrations per cell. PCR reactions were performed as follows: 95 °C for 3 min and 22 cycles of 98 °C for 20 s, 65 °C for 15 s and 72 °C for 40 s. The resulting PCR product was then size selected using a dual-size-selection clean-up of 0.5× and 1.0× AMPure XP beads (Beckman Coulter) to remove gDNA and small fragments (below 200 bp), respectively. The second PCR reaction included 1 ng of the size-selected product and 0.2 µM forward and reverse primers containing a flow-cell adaptor and sample index, with a final reaction volume of 25 µl. PCR reactions were performed as follows: 95 °C for 3 min and five cycles of 98 °C for 20 s, 65 °C for 15 s and 72 °C for 40 s). The final PCR product was pooled and cleaned with 0.9× AMPure XP beads (Beckman Coulter). The library was sequenced as a single-end read with either a 150 cycle kit on MiSeq or NextSeq 500/550, or a 100 cycle P1/P2 kit on NextSeq 2000. FASTQ files were demultiplexed with bcl2fastq (v.2.20, Illumina). Primers used for PCR are provided in Supplementary Table 5.

### Analysis of recording data
The barcodes used in this paper include CTT insertion, pentamer (5 bp degenerate or specific barcodes) and hexamer (300 specific barcodes for 300 unique CREs) on the HEK3 DNA Tape, and the hexamer (NNNGGA, two unique barcodes for two signals) on DTT. To ensure distinctiveness for CRE and signal recordings, hexamer and pentamer barcodes were selected with a Hamming distance greater than two from other members within the same set. For some but not all experiments, barcodes were picked to have a balanced editing score to minimize recording efficiency bias across different insertion sequences. The criteria by which insertional barcodes were chosen for ENGRAM recorders used in experiments throughout this paper is summarized in Supplementary Table 6.

For extraction of barcode information from sequencing reads, custom commands and python code were used. For barcodes recorded in the *HEK3* locus, a custom pattern-matching function was used followed by analysis with custom python code. For CRE- and signal-specific barcodes, unexpected barcodes within one Hamming distance from the expected sequences were corrected for insertion counts whereas raw counts were used in 5N degenerate barcode recording. Barcodes with fewer than five reads were excluded from downstream analysis. The editing score was calculated as (genomic reads with specific insertion/total edited HEK3 reads)/(plasmid reads with specific insertion/total plasmid reads). Two-tailed Student's *t*-tests were performed for comparison of differences between two recording conditions. Differential activity analysis for 98 synthetic ENGRAM recorders between different cells was performed using DESeq2 (ref. 47), with raw barcode counts as input. Barcodes accounting for less than 0.01% of total barcode reads were removed from the analysis. Differential active recorders were called with thresholds of adjusted $P < 0.001$ (mES versus K562 cells, and mES versus HEK293T cells) or $P < 0.1$ (mES cells versus gastruloid) for a fold difference greater than two.

For hexamers recorded to DTT (NNNGGA), sequencing reads were first aligned to the five-unit DTT reference using bwa (v.0.7.1)[48] with default settings. The aligned reads were then processed with custom python code to extract the positional insertion and bigram proportions at adjacent positions on the five-unit DTT. The order of signals was

inferred by calculation of the bigram ratio ($\log_2$-transformed ratio of the (Tet-On → WNT) versus (WNT → Tet-On) bigrams at adjacent positions).

## Bulk RNA-seq and data analysis

For bulk RNA-seq experiments, HEK293T cells, single-cell-derived PE2(+) HEK293T cells and PE2(+) ENGRAM-NF-κB-recorder(+) HEK293T cells (treated with 10 ng ml$^{-1}$ TNF or PBS for 48 h) were collected in triplicate. RNA from cells collected was purified using the RNeasy Mini Kit (Qiagen, 74104) with on-column DNase treatment using the RNase-Free DNase Set (Qiagen, 79254). A complementary DNA sequencing library was generated using TruSeq RNA Library Prep Kit v.2 (Illumina), following the manufacturer's protocol, and sequenced with a paired-end, 100 cycle P2 kit on NextSeq 2000. Fastq files were demultiplexed with bcl2fastq (v.2.20, Illumina). Sequencing reads were trimmed using Cutadapt[49] and aligned to the human reference genome (hg38) using STAR (v.2.7.3)[50], both with default settings. Differential expression analysis was performed using DESeq2 (ref. 47). Differentially expressed genes were called with thresholds of adjusted $P < 0.05$ for a change of over 50% ($\log_2$-transformed fold change above 0.58).

## Prediction of RNA structure and editing score

Both RNA structure and minimal free energy prediction were performed using the NUPACK python package[51] with default settings. A linear lasso regression model to predict editing score of 5 bp barcodes was trained using the python package scikit-learn. We defined 85 features to characterize the 5 bp sequence for which insertional efficiency is predicted: (1) sequence features or 84 binary features corresponding to one-hot encoded sequence, including 20 for single-nucleotide content (four nucleotides × five positions) and 64 for dinucleotide content (16 dinucleotides × four positions); and (2) structure feature or rescaled minimum free energy within the range (0,1). Samples were split with 724 barcodes in a training set and 300 in a test set. The model was trained with tenfold cross-validation on the training set and then used to predict the test set.

## MOI estimation using PCR and qPCR

The MOI of various constructs was determined with one of two methods: quantitative PCR (qPCR) and PCR followed by DNA quantification with TapeStation High Sensitivity reagents (Agilent).

To assess the overall MOI of piggyBac integration, K562 cells were transfected with GFP cargo plasmid with or without piggyBac transposase plasmid. Genomic DNA was purified every 2–3 days for 15 days using the DNeasy Blood & Tissue Kit (Qiagen, 69504). Either qPCR on gDNA was performed using TaqPath qPCR Master Mix (ThermoFisher, A15297) with primers designed for GFP and RPPH1 as internal control. MOI was estimated by normalization of $GFP\ C_t$ values to $RPPH1\ C_t$ values, assuming two copies in the genome.

For assessment of the MOI of specific recording components (PEmax, ENGRAM recorder and synHEK3-Tape), gDNA from the cell lysate was amplified with specific primers and quantified relative to PCR using RPPH1 primers. In brief, 1 million cells were counted and lysed in 200 μl of lysis buffer. PCR included 2 μl of cell lysate (equivalent to an input of 10,000 cells) and 0.5 μM forward and reverse primers targeting a specific region, with a final reaction volume of 25 μl. PCR reactions were performed as follows: 95 °C for 3 min and 22 cycles of 98 °C for 20 s, 65 °C for 15 s and 72 °C for 40 s. PCR products were quantified using Tapestation and MOI was estimated by normalization of the target

DNA concentration to RPPH1 DNA concentration. The sequences of primers used for qPCR and PCR are provided in Supplementary Table 5.

## Reporting summary

Further information on research design is available in the Nature Portfolio Reporting Summary linked to this article.

## Data availability

Raw sequencing data have been uploaded on Sequencing Read Archive (SRA) with associated BioProject ID PRJNA780310. Processed data are available at GitHub (https://github.com/shendurelab/ENGRAM). With the Jupyter notebook provided, all results and figures in the manuscript are fully reproducible. Plasmids for ENGRAM recorders (piggyBac-5′-ENGRAM and piggyBac-3′-FT-ENGRAM) have been deposited to Addgene (ID 179157 and 179158).

## Code availability

Custom analysis code used for this project is available at GitHub (https://github.com/shendurelab/ENGRAM).

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

**Acknowledgements** We thank members of the Shendure Laboratory, in particular S. Domcke, A. McKenna, A. Minkina, F. Chardon and T. McDiarmid, as well as members of the Allen Discovery Center for Cell Lineage Tracing, in particular M. Elowitz and A. Schier, for helpful discussions. We thank D. Liu's laboratory for sharing prime editing plasmids. We thank C. Schröter for sharing mESC cells. This work was supported by a grant from the Paul G. Allen Frontiers Group (Allen Discovery Center for Cell Lineage Tracing, to J.S.), Alex's Lemonade Stand Foundation (to J.S.) and the National Human Genome Research Institute (UM1HG011586 to J.S., K99HG012973 to J.C. and F31HG011576 to S.R.). J.C. was a Howard Hughes Medical Institute Fellow of the Damon Runyon Cancer Research Foundation (DRG-2403-20). J.-B.L. is a fellow of the Damon Runyon Cancer Research Foundation (DRG-2435-21). J.F.N. is supported by the UW ISCRM Fellows Program. J.S. is an Investigator of the Howard Hughes Medical Institute.

**Author contributions** W.C., J.C. and J.S. conceptualized the project. W.C. and J.S. designed the experiments with input from J.C. W.C. performed the experiments with assistance from J.C., X.L., J.F.N., B.M., W.Y., E.N., A.L. and C.L. N.H. and S.R. helped with mES cell and gastruloid culture. X.L. cloned the synHEK3 reporter construct. H.K. cloned the TRE-PEmax construct. W.C. analysed the data with input from J.C., J.-B.L., C.Q. and V.A. W.C. and J.S. wrote the manuscript with input from all other authors. J.S. supervised the project.

**Competing interests** The University of Washington has filed a patent application partially based on this work, in which J.C., W.C. and J.S. are listed as inventors. J.S. is on the scientific advisory board, a consultant and/or a cofounder of Adaptive Biotechnologies, Cajal Neuroscience, Camp4 Therapeutics, Guardant Health, Maze Therapeutics, Pacific Biosciences, Phase Genomics, Prime Medicine, Scale Biosciences, Somite Therapeutics and Sixth Street Capital. The remaining authors declare no competing interests.

**Additional information**
**Correspondence and requests for materials** should be addressed to Wei Chen or Jay Shendure.

**a**

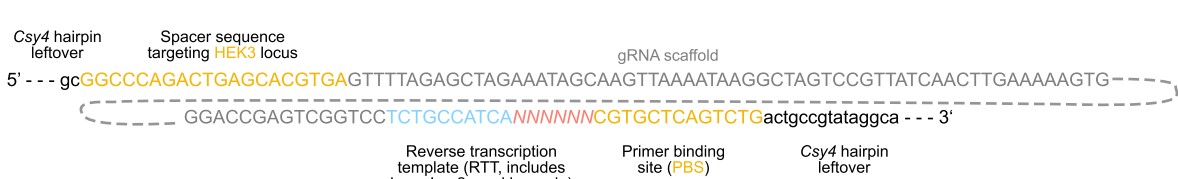

Csy4 hairpin leftover　　Spacer sequence targeting HEK3 locus　　　　　　gRNA scaffold

5' - - - gcGGCCCAGACTGAGCACGTGAGTTTTAGAGCTAGAAATAGCAAGTTAAAATAAGGCTAGTCCGTTATCAACTTGAAAAAGTG- - - -

- - - - GGACCGAGTCGGTCCTCTGCCATCA*NNNNNN*CGTGCTCAGTCTGactgccgtataggca - - - 3'

Reverse transcription template (RTT, includes homoLog2y and barcode)　　Primer binding site (PBS)　　Csy4 hairpin leftover

**b**

*ENGRAM*

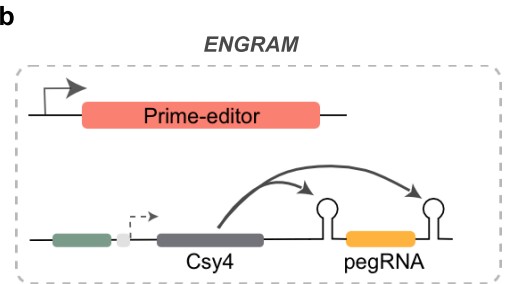

Prime-editor

Csy4　　pegRNA

**Extended Data Fig. 1 | ENGRAM architecture. (a)** Sequence of the pegRNA predicted to be liberated from a Pol-2 transcript by Csy4. The *csy4*-hairpin residuals on both ends are shown in lower case. The spacer and primer binding sequence (PBS) are highlighted in orange. The reverse transcription template (RTT) consists of a homology arm (blue) and barcode (red). **(b)** Schematic of ENGRAM recorder. A pegRNA writing unit is flanked by *csy4* hairpins and embedded within the 3′ or 5′ UTR of a Pol-2-driven Csy4-encoding mRNA. PE2 (or PEmax) is constitutively expressed from a separate locus. When the ENGRAM recorder is active, Csy4 is produced, cleaves at the *csy4* hairpins and releases the active pegRNA.

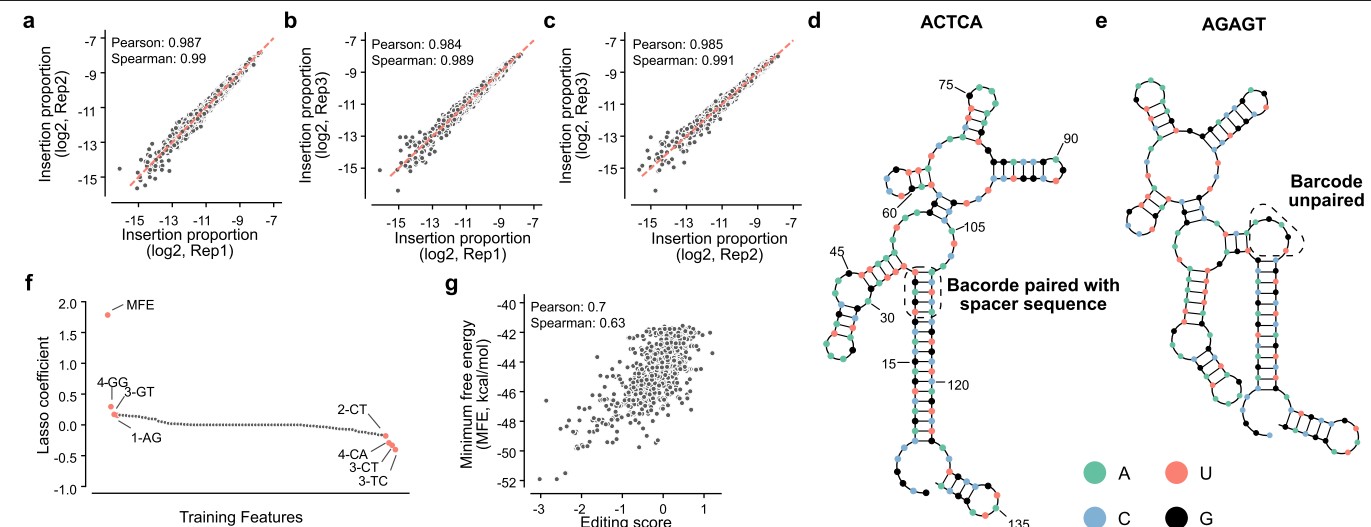

**Extended Data Fig. 2 | ENGRAM installs insertional barcodes with reproducible, predictable efficiencies. (a-c)** The relative proportions of 1023 5 N barcodes installed by ENGRAM driven by the constitutive Pol-2 PGK promoter were measured in triplicate. Log-scaled insertion proportions (calculated as the proportion of edited *HEK3* sites with a given insertion) were strongly correlated between pairs of transfection replicates. **(d-e)** Predicted secondary structures for pegRNAs with the lowest (left) and highest (right) insertional efficiencies. Sequences shown above are those observed in DNA Tape, which are the reverse complement of sequences in pegRNAs. **(f)** The rank-ordered coefficients of the linear lasso regression. Positional information of single nucleotides and dinucleotides and minimum free energy (MFE) of secondary structure were used as input features for training. In addition to MFE, which received the highest coefficient, the top 4 and bottom 4 coefficients for sequence features are annotated (*e.g.* 3-TC means TC dinucleotide starting at position 3). **(g)** MFE alone can explain 70% of the variance in editing scores observed for different insertional barcodes.

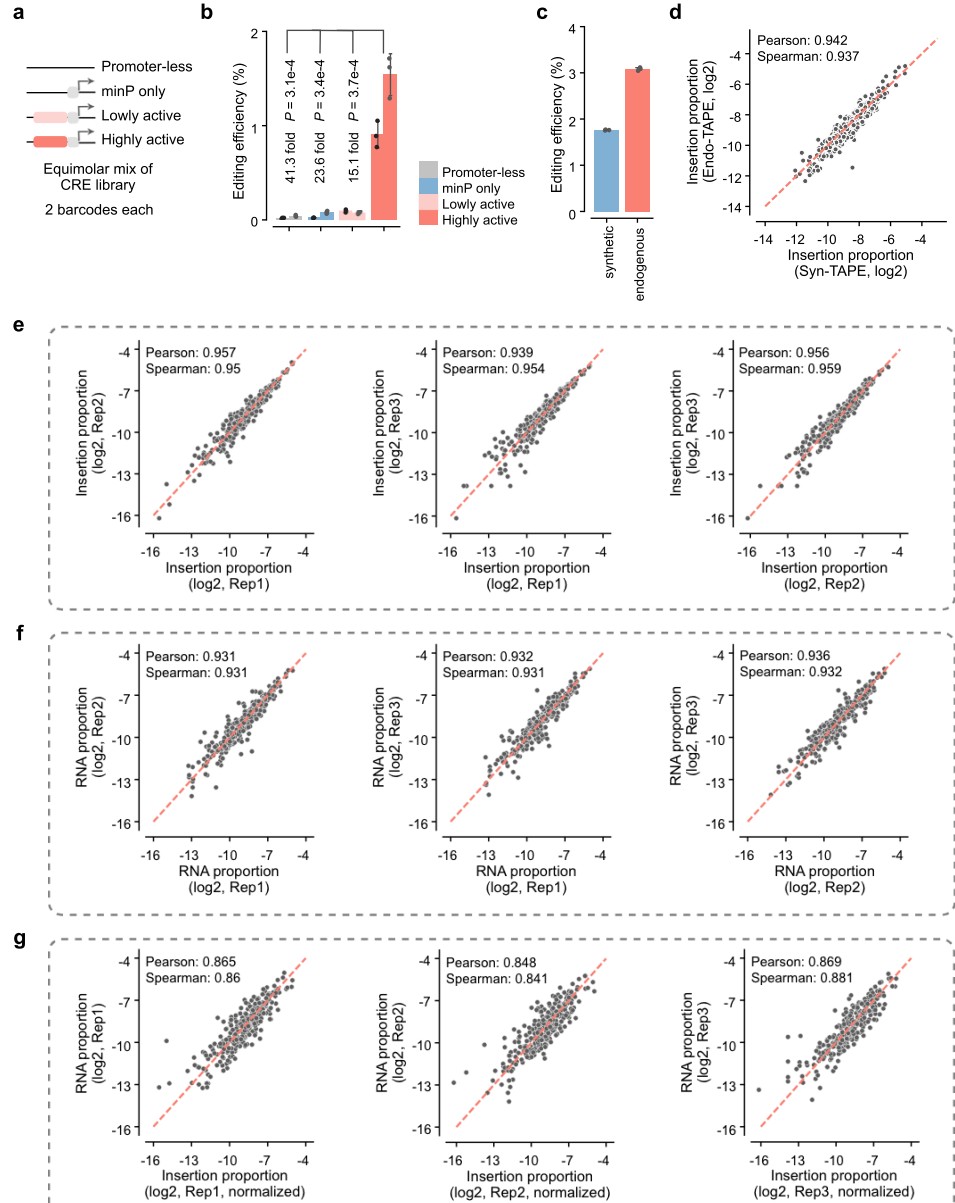

**Extended Data Fig. 3 | Benchmarking of ENGRAM against reporter assays.**
**(a)** ENGRAM recorders with highly vs. lowly active CRE fragments (as previously measured via MPRA) upstream of a minP, together with minP-only and promoter-less constructs, were cloned, each driving expression of two distinct pegRNA-encoded barcodes. **(b)** Barplot showing the editing efficiency of individual barcodes associated with each of the eight members of the CRE library (4 architectures x 2 barcodes each). Fold differences were calculated by first summing the counts for the pair of barcodes associated with each architecture, and then calculating the ratio between pairs of architectures. Barcodes corresponding to the highly active CRE were 41.3-fold, 23.6-fold, and 15.1-fold more abundant than barcodes corresponding to promoter-less, minP-only or lowly active CRE controls, respectively. P-values were from two-tailed t-test. **(c)** Insertion efficiency of various barcodes at synthetic (1.8%) vs. endogenous (3.1%) *HEK3* loci are highly correlated. Of note, for synthetic

*HEK3* sites, the observed efficiencies reflect an average across many genomic contexts. The center and error bars in **b-c** correspond to mean and standard deviations, from n = 3 integration replicates. **(d)** Log-scaled insertion proportions for 300 6-mer barcodes were highly correlated between DNA Tape sites located at synthetic vs. endogenous *HEK3* loci. **(e)** Log-scaled insertion proportions for 300 6-mer barcodes were highly reproducible between integration replicates. Each value corresponds to the proportion of barcodes read out from synthetic DNA Tape. **(f)** Log-scaled RNA proportions for 300 6-mer barcodes were highly reproducible across integration replicates. Each value corresponds to the proportion of barcodes read out at the RNA level from transcribed pegRNAs. **(g)** The log-scaled proportions of ENGRAM events recorded to DNA were highly correlated with log-scaled proportions of barcodes measured directly from RNA.

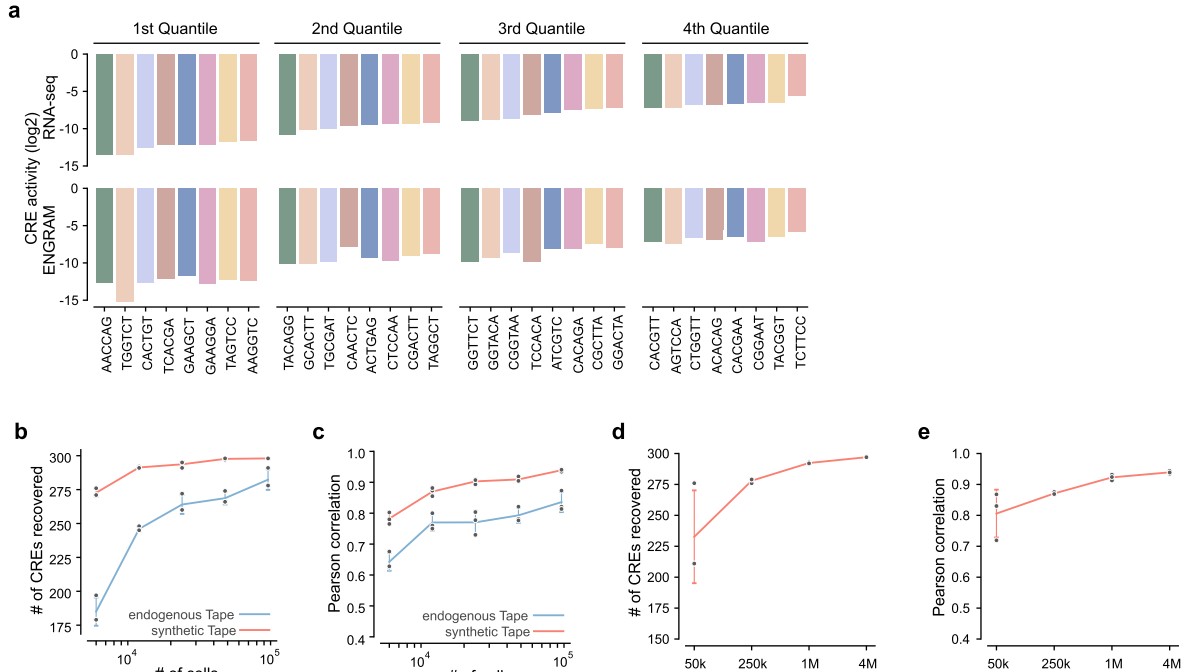

**Extended Data Fig. 4 | Further benchmarking of ENGRAM. (a)** Comparison of CRE ranks for ENGRAM vs. MPRA across quartiles. Eight CREs were randomly sampled from each of four quartiles based on the RNA-based activity measurement (*i.e.* MPRA). The relative activity based on reporters (MPRA) for each set of eight is shown at the top, and the activity for the same CREs based on recorders (ENGRAM) is shown at the bottom. Overall, ENGRAM reasonably preserved the rank of CREs when comparing the quartiles to one another. **(b-c)** Different cell numbers were sampled (6,000, 12,000, 24,000, 48,000, 96,000 cells) prior to measuring ENGRAM recorded activity of 300 CREs, either from endogenous and synthetic DNA Tape, and then recovery **(b)** and reproducibility **(c)** were assessed. **(d-e)** Sequencing data from synthetic DNA Tape and 96,000 cell input condition was downsampled, and then recovery **(d)** and reproducibility **(e)** were assessed. The error bars in **b-e** correspond to standard deviations, from n = 3 integration replicates.

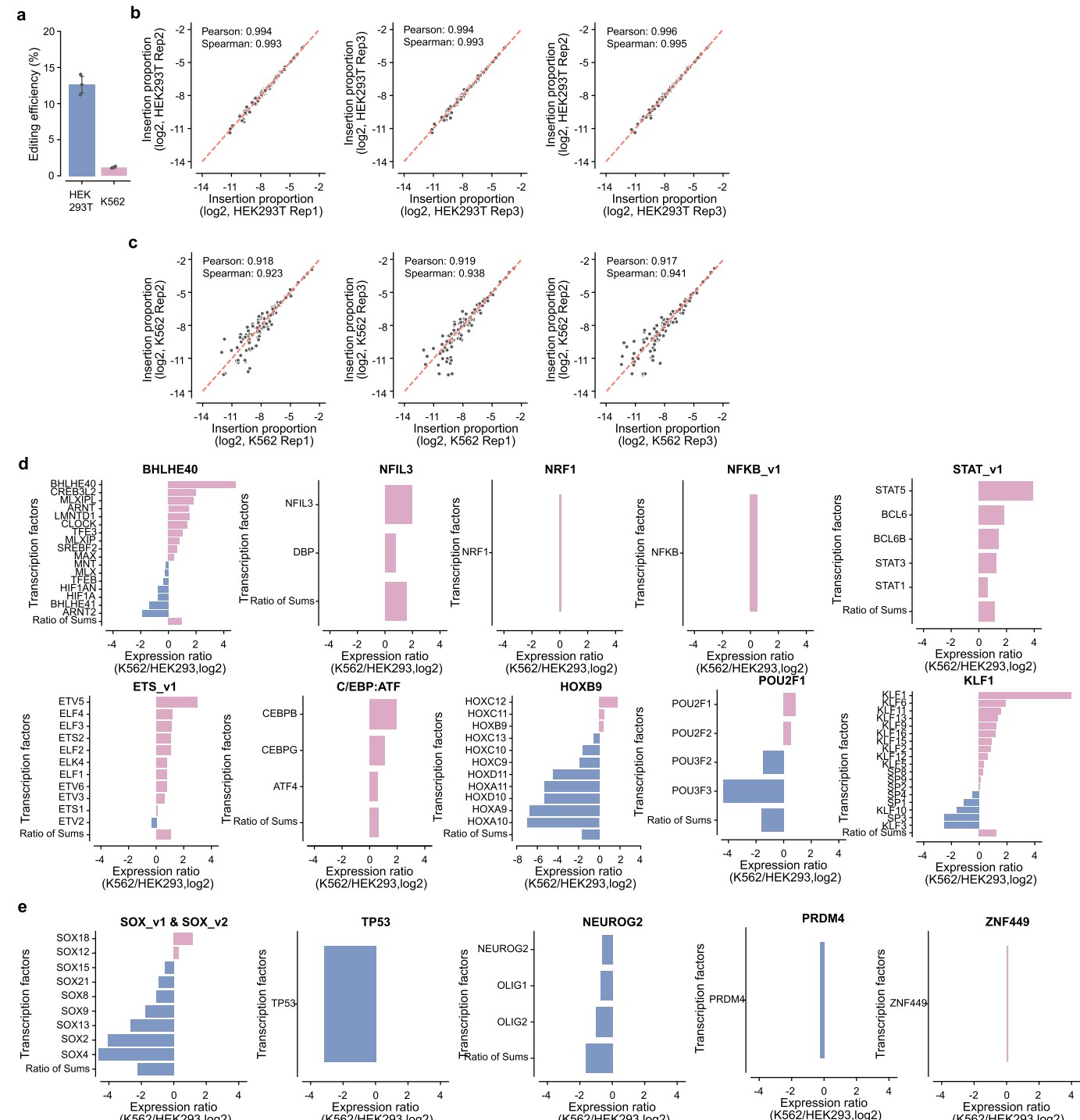

**Extended Data Fig. 5 | Multiplex recording of cell-type-specific activities of synthetic CREs with ENGRAM. (a)** Recording efficiency of synthetic CREs at the endogenous DNA Tape site (*HEK3* locus) in HEK293T (12.6%) and K562 (1.0%) cells. The difference in overall recording between cell lines is likely attributable to differences in transfection efficiency. The center and error bars correspond to mean and standard deviations, from n = 3 transfection replicates **(b-c)** Log-scaled insertion proportions for 5-mer barcodes linked to the 98 synthetic CREs were highly reproducible across transfection replicates for both HEK293T **(b)** and K562 **(c)** cells. Each value corresponds to the proportion of barcodes read out at the DNA level from the endogenous *HEK3* locus. As the same number of cells were sampled for recording, the lower reproducibility in K562 cells is likely secondary to lower transfection/editing efficiency. **(d-e)** Differential expression of TFs in HEK293T vs. K562 cells. As many TFs share

similar binding motifs, here we show expression ratios between the cell lines for all expressed TFs (normalized transcripts per million > 0.5 in one or both cell lines) assigned to the corresponding motif by JASPAR, for each of the 17 differentially active synthetic CRE recorders (Fig. 2g). In the bottom row of each plot, we show an expression ratio based on summing the read counts of all the motif-associated TFs in bulk RNA-seq data[52] from these cell lines, except for GCM1, as its JASPAR-associated TFs (GCM1, GCM2) are not detected as expressed in either cell line. Pink, higher expression in K562 cells; Blue, higher expression in HEK293T cells. Most K562-specific **(d)** and HEK293T-specific **(e)** recording activities are directionally concordant with the summed expression of the TFs that JASPAR associates with the motif embedded in the synthetic CRE (all but the HOXB9, POU2F1, ZNF449, and GCM1-named synthetic CRE recorders; 13/17; p = 0.02; binomial test).

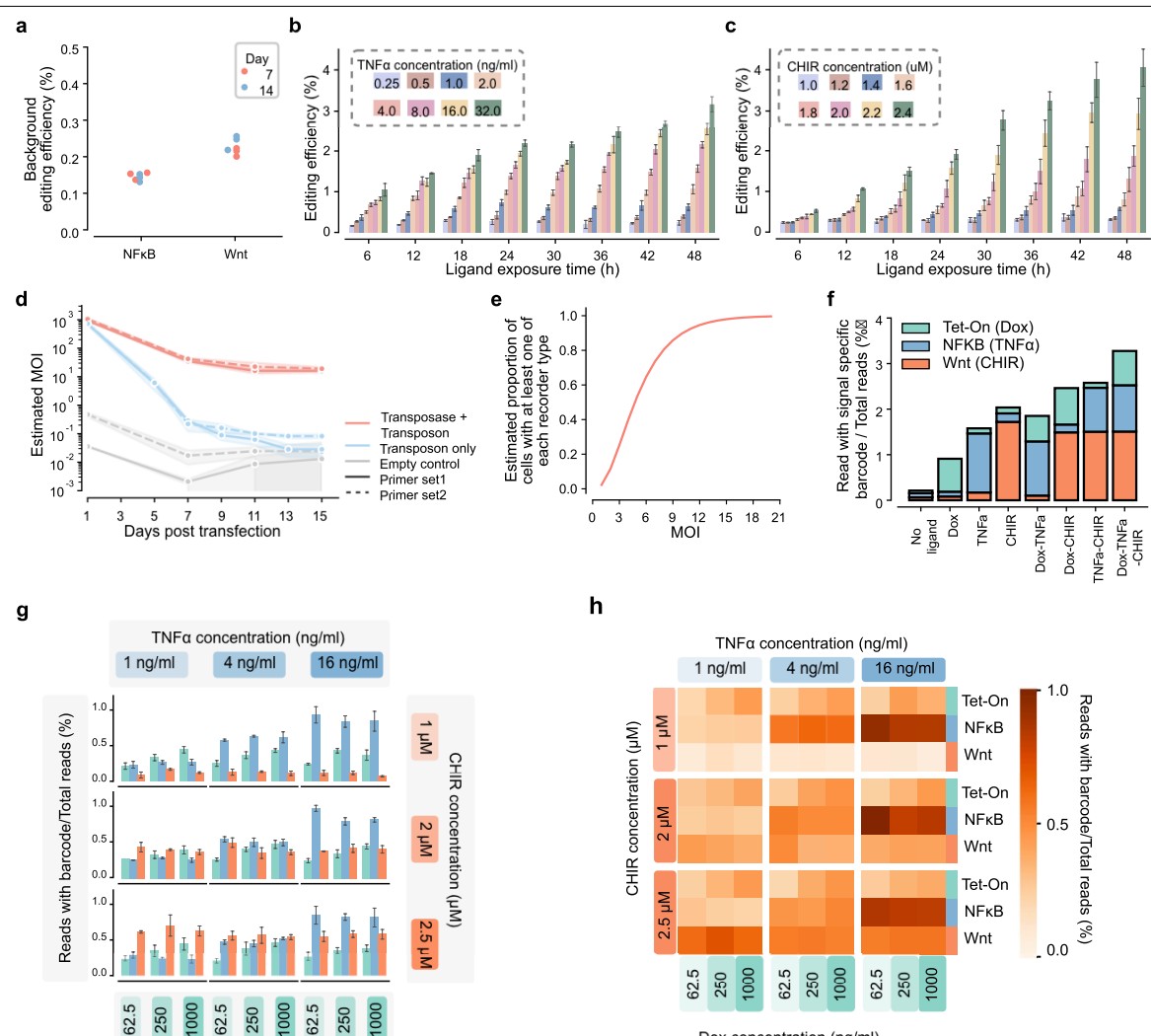

**Extended Data Fig. 6 | Recording of the intensity and duration of signaling pathway activity. (a)** We observed minimal background recording in the absence of stimulus with signal-responsive ENGRAM recorders after 7 or 14 days. This background did not accumulate over time, consistent with the hypothesis that it primarily accumulates shortly after transfection, potentially due to ORI-driven, plasmid-mediated transcription. Plotted points correspond to three integration replicates. **(b-c)** Histograms, broken out by ligand exposure time and agonist concentration, showing editing efficiencies resulting from matrix experiment on the NFκB **(b)** and Wnt **(c)** recorders, in which both stimulant concentrations and durations of exposure were varied (2 recorders x 8 concentrations x 8 durations x 3 integration replicates = 384 conditions). **(d)** Estimating the multiplicity of integration (MOI) of piggyBac transposase integration with qPCR. Cells were transfected with a piggyBac-GFP construct, either with or without piggyBac transposase. GFP DNA abundance was measured using qPCR with two pairs of GFP-specific primers (together with a pair of primers directed at native RPPH1 locus as an internal control) over the course of 15 days. The estimated levels of GFP abundance in the no-transposase control decreased to background levels after

7-9 days. DNA-level GFP abundance with transposase present, at timepoints where controls have gone to background, suggests that we were achieving an MOI on the order of 15-20. **(e)** The estimated proportion of cells with at least one copy of each of the three recorders as a function of MOI, assuming a Poisson distribution. At an MOI of 15, over 98% of cells are predicted to bear at least one copy of each of the three recorders. **(f)** Barcode composition of DNA Tape from cells treated with different combinations of stimuli. Of note, the recorders did not exhibit any discernible crosstalk, suggesting that the underlying signaling pathways are truly orthogonal (*e.g.* stimulating with CHIR does not lead to appreciable recording by the NF-κB recorder). **(g)** Cells bearing multiple recorders were exposed to all possible combinations of high, medium or low concentrations of three stimuli for 48 hrs, followed by harvesting and sequencing-based quantification of the levels of signal-specific barcodes. For Dox, 62.5, 250 or 1000 ng/ml were used; for TNFα, 1, 4 or 16 ng/ml; and for CHIR99021, 1, 2 or 2.5 μM. **(h)** Heatmap visualization of the data presented in panel **g**. The center and error bars in **b,c,g** correspond to mean and standard deviations, from n = 3 integration replicates.

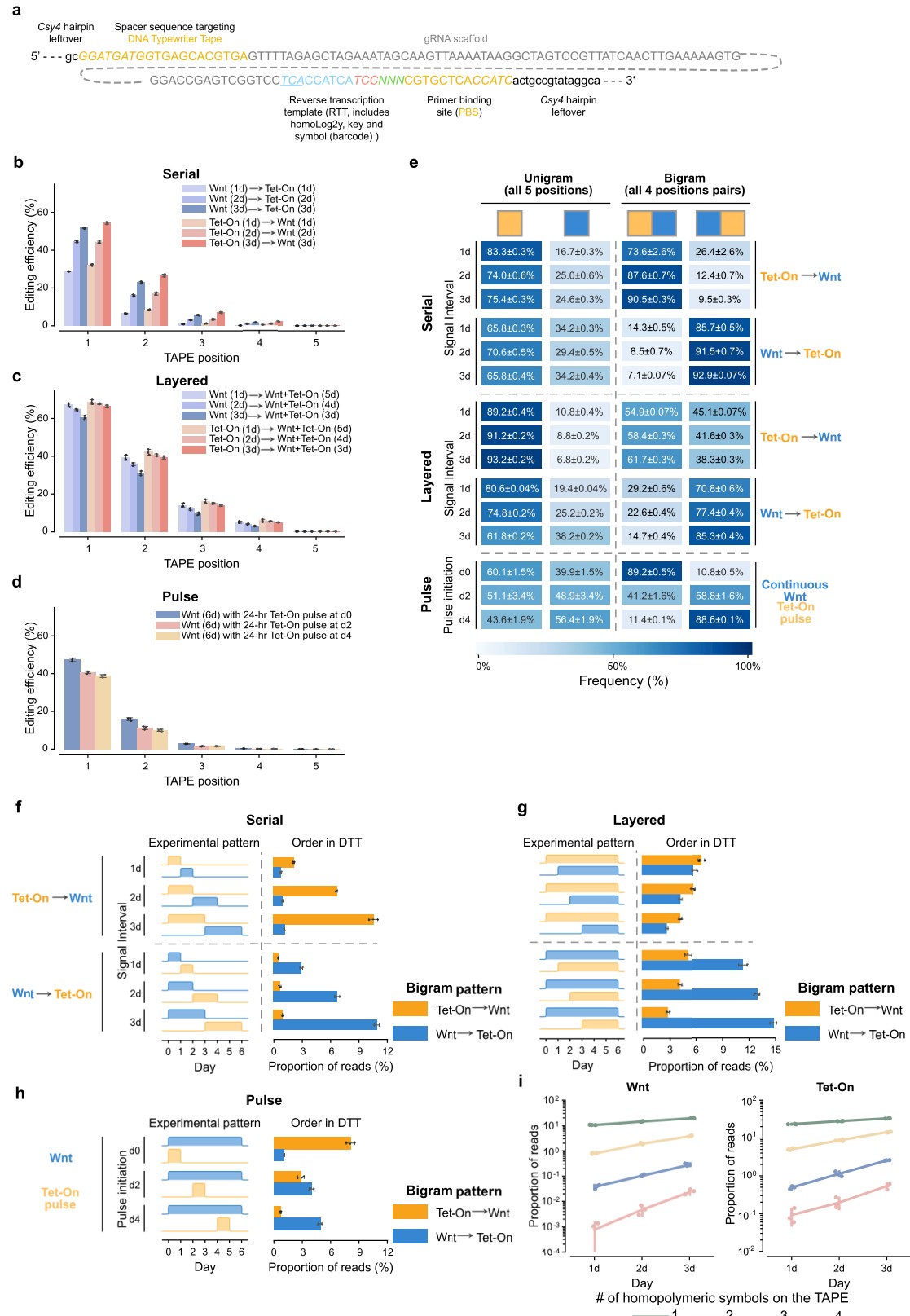

**Extended Data Fig. 7** | See next page for caption.

**Extended Data Fig. 7 | Combining ENGRAM and DNA Typewriter to record the temporal dynamics of biological signals. (a)** Sequence of the predicted pegRNA modified for compatibility with DNA Typewriter. This pegRNA is similar to the one shown in Extended Data Fig. 1a except that the spacer and PBS are modified to target the DNA Typewriter Tape's active type guide and the encoded insert is modified to include both a symbol (3-bp in this case) and a key sequence (3-bp). **(b-d)** Overall editing efficiencies for the serial **(b)**, layered **(c)** and pulse **(d)** programs, stratified by position along the 5-unit DNA Typewriter Tape. **(e)** Heatmap showing proportions at which two possible unigrams (left columns) or two possible heterogeneous bigrams (right columns) were observed. Note that proportions as shown here are calculated separately here for those classes in isolation, *i.e.* for each row, the left two columns sum to 100%, and the right two columns sum to 100%. **(f-h)** Modified ENGRAM recorders and 5-unit DTT were sequentially integrated to PEmax(+) HEK293T cells. We designed and tested serial **(f)** and layered **(g)** programs in which these cells were exposed to different patterns of 100 ng/ml doxycycline or 3 μM CHIR-99021 (left columns) across a total of 36 cell populations (2 patterns * 3 intervals * 2 possible orders * 3 integration replicates). Harvesting, amplifying and sequencing the DTT region after 6 days, we show here the absolute fractions of [Tet-On→Wnt] and [Wnt→Tet-On] bigrams at Sites 1-2 (right columns). See also Fig. 4b,c. **(h)** Same as panels f-g, but for pulse programs in which these cells were exposed to 500 ng/ml doxycycline for 24 hrs against the background of continuous stimulation with 3 μM CHIR-99021. A total of 9 cell populations are represented (3 pulse timings * 3 integration replicates). See also Fig. 4d. **(i)** Longer signal durations are associated with longer homopolymeric runs of the signal-specific symbol. Focusing on symbols corresponding to the first signal applied in Serial programs, we calculated the proportion of homopolymeric runs of various lengths, *i.e.* consecutive, identical symbols beginning at the first position in the DTT. Log-scaled proportions for homopolymeric runs of 1-4 are plotted for signal durations of 1, 2 or 3 days. We did not observe any "5-in-a-row" homopolymeric instances in the data. The center and error bars in **b-d, f-i** correspond to mean and standard deviations, from n = 3 integration replicates.

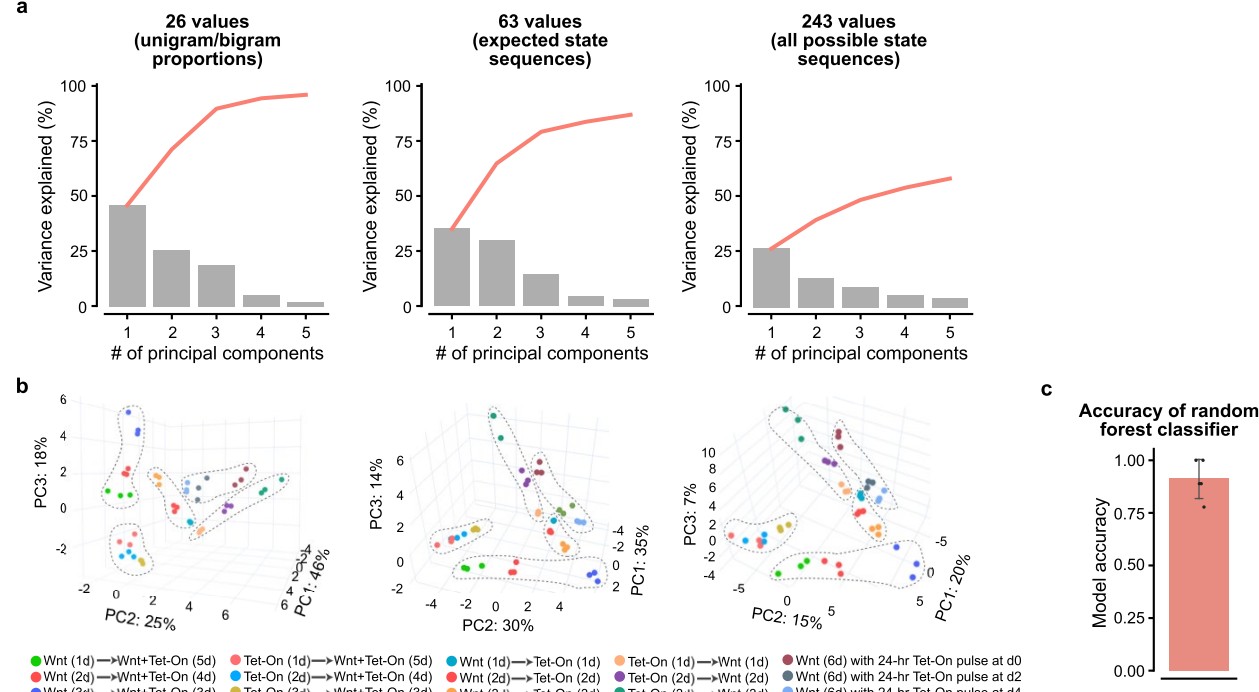

**Extended Data Fig. 8 | Decoding dynamic signaling programs based on ensembles of editing patterns resulting from the combination of ENGRAM and DNA Typewriter. (a-c)** Comparison of different encoding strategies. DTT recording data was encoded as either the proportions of various unigrams/bigrams at each position (2*5 = 10 unigrams; 4*4 = 16 bigrams; 26 values) **(left)**, the proportions of each possible state sequence that is consistent with ordered recording ($2^0 + 2^1 + 2^2 + 2^3 + 2^4 + 2^5 = 63$ values) **(middle)**, or the proportions of all possible sequences of three states (*unedited*, *Tet-on symbol*, *Wnt symbol*) across 5 positions (3^5 = 243 values) **(right)**. **(a)** Scree plots showing the proportion of variance explained by the top 5 principal components with each of the three strategies), which explain 96%, 89% and 50% of the variance, respectively. **(b)** PCA based on patterns observed in DTT across 45 cell populations subjected to various patterns of exposure to doxycycline and CHIR-99021 (15 programs, executed in triplicate). Circled subsets correspond to serial and layered programs in either order, or to pulse programs. The top three PCs are plotted for 26 value **(left)**, 62 value **(middle)** or 243 value **(right)** encoding. **(c)** Barplot showing the accuracy of applying a random forest classifier to assess which of the 15 signal programs an unseen set of sequenced tapes derived from. We randomly split the tape ensembles from 45 samples (15 programs x 3 integration replicates) into 5 groups and conducted 5-fold cross-validation, *i.e.* using each group once as a test set, while training on all other groups. The model achieved a mean accuracy of 0.91. The center and error bars correspond to mean and standard deviations, from n = 5 5-fold cross-validation.

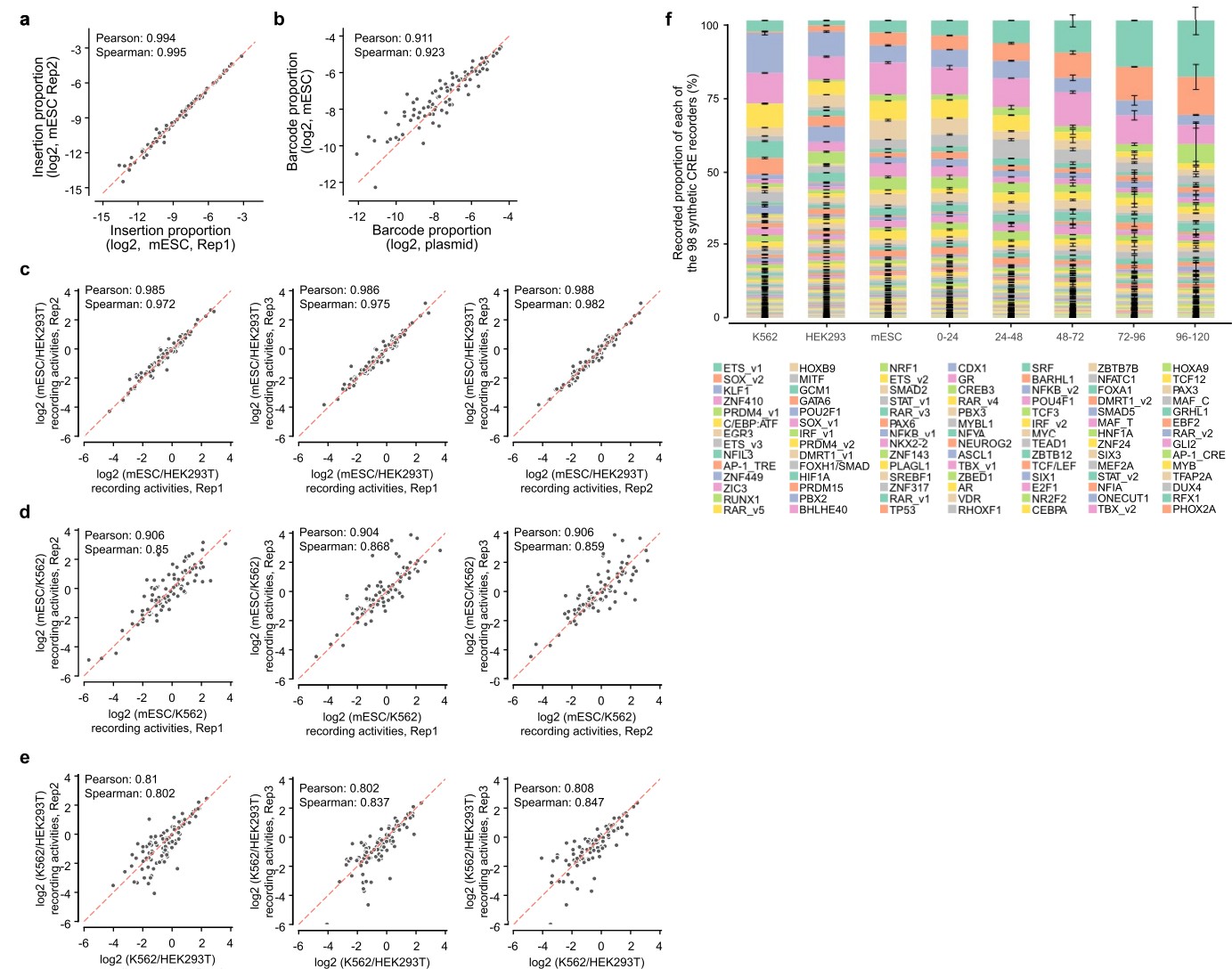

**Extended Data Fig. 9 | Multiplex recording of CRE activity in embryonic stem cells and differentiating gastruloids. (a)** Log-scaled insertion proportions for 5-mer barcodes linked to the 98 synthetic CRE-driven ENGRAM recorders were highly reproducible across integration replicates for cultured mESC, as read out by amplification and sequencing of synthetic DNA Tape. **(b)** Log-scaled proportions for 5-mer barcodes linked to the 98 synthetic CRE-driven ENGRAM recorders are well correlated between the original plasmid pool and genomic integrations in the polyclonal mESC line. **(c-e)** Log-scaled barcode proportion ratios, as calculated from one pair of replicates vs. as calculated from another pair of replicates, for mESC vs. HEK293T cells **(c)**, mESC vs. K562 cells **(d)**, or K562 vs. HEK293T cells **(e)**. Note that we corrected the mESC data for differences in relative abundances of recorders in the polyclonal mESC line vs. the plasmid pool used to transiently transfect K562 and HEK293T cells (as shown in Extended Data Fig. 9b), prior to performing these comparisons.

As the same number of cells were sampled for recording, the lower reproducibility for comparisons involving K562 cells is likely secondary to lower transfection/editing efficiency, as shown in Extended Data Fig. 5a. **(f)** Stacked bar plot showing the proportion of 5-mer barcodes associated with each of the 98 synthetic CRE-driven ENGRAM recorders in cell lines and gastruloids. Recorder activities are presented in the order of their maximal proportion across all samples. Error bars correspond to standard deviations across 3 transfection replicates (in K562 and HEK293T cells), or 3 integration replicates in mESCs and 2 integration replicates for gastruloid time-windows. Note that these labels are representative of TF(s) thought to bind each motif, and it remains uncertain which TF(s) are driving the activity of each synthetic CRE recorder. See Supplementary Table 3 for the corresponding consensus motifs, and the full list of TFs associated with each motif by the JASPAR database.

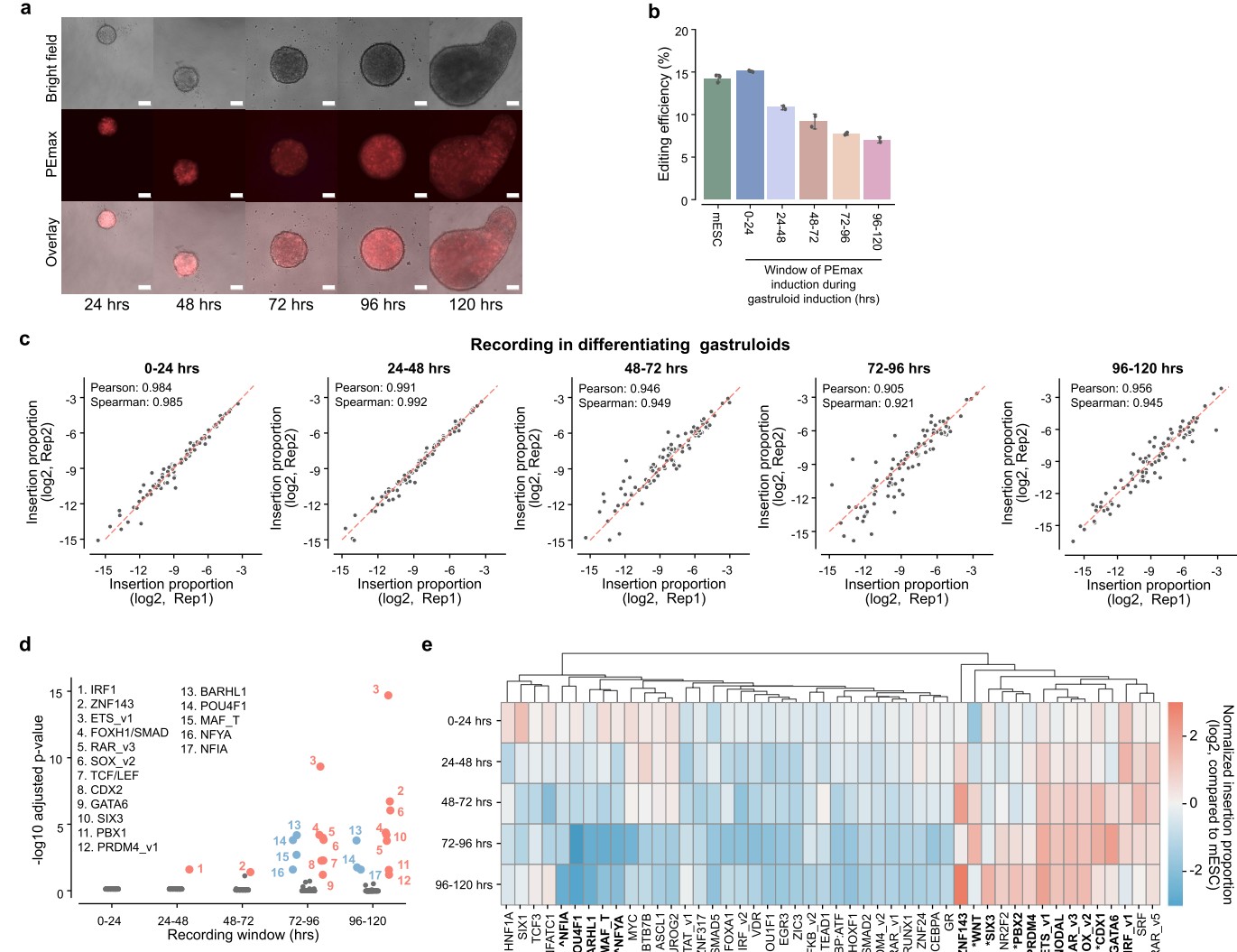

**Extended Data Fig. 10 | Multiplex recording of CRE activity in embryonic stem cells and differentiating gastruloids. (a)** Representative images of gastruloids induced from polyclonal ENGRAM mESCs, illustrating that the components of the ENGRAM recording system do not substantially impact the morphological development of gastruloids. Scale bar: 100 μm. **(b)** Overall ENGRAM recording efficiency at synthetic DNA Tape for cultured mESCs or differentiating mouse gastruloids in which PEmax was induced for a particular 24 hr window. The center and error bars correspond to mean and standard deviations, from n = 2-3 integration replicates. **(c)** Log-scaled insertion proportions for 5-mer barcodes linked to the 98 synthetic CRE-driven ENGRAM recorders were highly reproducible across integration replicates for differentiating gastruloids, as read out by amplification and sequencing of synthetic DNA Tape, for integration replicates in which doxycycline was used to induce PEmax during particular 24 hr windows. **(d)** Plot showing -log10 adjusted p-values of ENGRAM recorders (y-axis) across each 24 hr recording window (x-axis) during

gastruloid differentiation. A total of 17 ENGRAM recorders exhibiting significant and substantial differences (Wald-test with Benjamini-Hochberg correction P < 0.1 for a fold-difference >2) for a particular window, in comparison to recordings made from the same polyclonal ENGRAM mESCs under normal culture conditions. The 12 recorders with increased activity are labeled in red, and the 5 recorders with decreased activity are labeled in blue. **(e)** Heatmap presenting recorded activities across each 24 hr interval (rows) for 48 of the 98 ENGRAM recorders with substantial activity in any of the five windows (columns). Columns are hierarchically clustered for presentation purposes. Values are log-scaled barcode proportion ratios for gastruloids with windowed recording in a particular 24 hr interval vs. cultured mESCs. The 17 recorders whose activity exhibiting significantly and substantially increased (*) or decreased (^) activity in one or more of the 24 hr windows, relative to cultured mESCs, are bolded (Wald-test with Benjamini-Hochberg correction P < 0.1 for a fold-difference >2).

# Reporting Summary

## Statistics

For all statistical analyses, confirm that the following items are present in the figure legend, table legend, main text, or Methods section.

| n/a | Confirmed | |
|-----|-----------|---|
| ☐ | ☒ | The exact sample size ($n$) for each experimental group/condition, given as a discrete number and unit of measurement |
| ☐ | ☒ | A statement on whether measurements were taken from distinct samples or whether the same sample was measured repeatedly |
| ☐ | ☒ | The statistical test(s) used AND whether they are one- or two-sided *Only common tests should be described solely by name; describe more complex techniques in the Methods section.* |
| ☒ | ☐ | A description of all covariates tested |
| ☒ | ☐ | A description of any assumptions or corrections, such as tests of normality and adjustment for multiple comparisons |
| ☐ | ☒ | A full description of the statistical parameters including central tendency (e.g. means) or other basic estimates (e.g. regression coefficient) AND variation (e.g. standard deviation) or associated estimates of uncertainty (e.g. confidence intervals) |
| ☐ | ☒ | For null hypothesis testing, the test statistic (e.g. $F$, $t$, $r$) with confidence intervals, effect sizes, degrees of freedom and $P$ value noted *Give P values as exact values whenever suitable.* |
| ☒ | ☐ | For Bayesian analysis, information on the choice of priors and Markov chain Monte Carlo settings |
| ☒ | ☐ | For hierarchical and complex designs, identification of the appropriate level for tests and full reporting of outcomes |
| ☐ | ☒ | Estimates of effect sizes (e.g. Cohen's $d$, Pearson's $r$), indicating how they were calculated |

*Our web collection on statistics for biologists contains articles on many of the points above.*

## Software and code

Policy information about availability of computer code

| Data collection | Illumina Next generation sequencing platform (Nextseq 500, Nextseq 2000 and Miseq) are used for data collection. |
|-----------------|---|
| Data analysis | We have used bcl2fastq(v2.20), bwa (v0.7.1), NUPACK (4.0.0.27), custom Python code (Python 3.10) and custom R code (R 4.1) for analysis. Custom code is available at https://github.com/shendurelab/ENGRAM |

For manuscripts utilizing custom algorithms or software that are central to the research but not yet described in published literature, software must be made available to editors and reviewers. We strongly encourage code deposition in a community repository (e.g. GitHub). See the Nature Portfolio guidelines for submitting code & software for further information.

## Data

Policy information about availability of data

All manuscripts must include a data availability statement. This statement should provide the following information, where applicable:
- Accession codes, unique identifiers, or web links for publicly available datasets
- A description of any restrictions on data availability
- For clinical datasets or third party data, please ensure that the statement adheres to our policy

Raw sequencing data have been uploaded on Sequencing Read Archive (SRA) with associated BioProject ID PRJNA780310. Processed data is available at Github: https://github.com/shendurelab/ENGRAM. With the provided Jupyter notebook, the results and figures in the manuscript are fully reproducible. Plasmids for ENGRAM recorders (piggyBac-5'-ENGRAM and piggyBac-3'-FT-ENGRAM) have been deposited to Addgene (ID 179157 and ID 179158). TF motif data was obtained from JASPAR database (https://jaspar2020.genereg.net/)

# Field-specific reporting

Please select the one below that is the best fit for your research. If you are not sure, read the appropriate sections before making your selection.

☒ Life sciences   ☐ Behavioural & social sciences   ☐ Ecological, evolutionary & environmental sciences

For a reference copy of the document with all sections, see nature.com/documents/nr-reporting-summary-flat.pdf

# Life sciences study design

All studies must disclose on these points even when the disclosure is negative.

| | |
|---|---|
| Sample size | No sample size estimation was performed. In all experiments, we have performed 2-3 technical replicates for genome editing efficiency, which is sufficient to estimate the technical variation. Values measured in technical replications were used to calculate mean and standard deviations. |
| Data exclusions | No data was excluded |
| Replication | All experiments were performed at least with three replicates (transfection replicates, culture in separate wells) All attempts at replication were successful |
| Randomization | Randomization is not relevant to this study, as we only used human and mouse cells for measuring genome editing efficiencies. |
| Blinding | Blinding is not relevant to this study, as the values we measured in sequencing are quantitative and doesn't need subject judgment. |

# Reporting for specific materials, systems and methods

We require information from authors about some types of materials, experimental systems and methods used in many studies. Here, indicate whether each material, system or method listed is relevant to your study. If you are not sure if a list item applies to your research, read the appropriate section before selecting a response.

### Materials & experimental systems

| n/a | Involved in the study |
|---|---|
| ☒ | ☐ Antibodies |
| ☐ | ☒ Eukaryotic cell lines |
| ☒ | ☐ Palaeontology and archaeology |
| ☒ | ☐ Animals and other organisms |
| ☒ | ☐ Human research participants |
| ☒ | ☐ Clinical data |
| ☒ | ☐ Dual use research of concern |

### Methods

| n/a | Involved in the study |
|---|---|
| ☒ | ☐ ChIP-seq |
| ☒ | ☐ Flow cytometry |
| ☒ | ☐ MRI-based neuroimaging |

## Eukaryotic cell lines

Policy information about cell lines

| | |
|---|---|
| Cell line source(s) | HEK293T (ATCC), K562 (ATCC), Mouse embryonic stem cells (mESC, E14TG2a, gift from Dr. Christian Schröter |
| Authentication | All cell lines were used as received without further autherticcation. |
| Mycoplasma contamination | Cell lines were not detected for mycoplasma contamination |
| Commonly misidentified lines (See ICLAC register) | Commonly misidentified cell lines were not used in this study. |

