## [Peer Review File · Nature]

Manuscript Title: Symbolic recording of signaling and cis regulatory element activity to DNA

Editorial Notes:

Redactions – unpublished data

Reviewer Comments & Author Rebuttals

Reviewer Reports on the Initial Version:

Referees' comments:

Referee #1 (Remarks to the Author):

Chen et al. demonstrated a proof-of-principle DNA event recording system by harnessing a massively parallel reporter assay (MPRA) and CIRPSR Prime Editing (PE). In brief, their system, termed ENGRAM (Enhancer-driven genomic recording of transcriptional activity in multiplex), places multiple prime-editing guide RNAs (pegRNAs) sharing the same target sequence but with different small insert sequences which are molecular identifiers under the expression of corresponding enhancer element. ENGRAM uses a previously described gRNA processing strategy (Cys4 system) to mature pegRNAs from their enhancer-driven transcripts and confer editing. The authors also carefully tested various ENGRAM systems to derive an optimal 5' Recorder system. They demonstrated the 5' Recorder system to record the activities of ~300 enhancer regions in K562 cells and the correlation between the recorded DNA readouts and RNA sequencing-based readouts. While the analyses in Figures 1 and 2 seemed to be designed very carefully, Figures 3 and 4 are a bit rough. I understand how an ideal DNA-based recording would revolutionize biology, share the vision with the authors and realize that this is one of the first proof-of-demonstration studies. However, none of the observations derived by ENGRAM in the present study is what we cannot achieve by the other current technologies. I also feel they oversell the possibility of ENGRAM without sufficient datasets. I will also address these points one by one below.

Major comments:

1. "Overall, there remains a need for a means of capturing signaling and gene regulatory activity that is at once quantitative, reproducible, non-destructive, multiplexable, applicable to physically opaque biological systems and capable of integrating the temporal dynamics of large numbers of signals." I strongly agree with this point and this is the whole beauty of the recently emerging molecular recording concept. I also greatly appreciate that the authors carefully designed and established beautiful circuits. However, all the demonstrations were performed using cell culture. The CRE library recording was impressive but similar data can be obtained by RNA-seq of a subsampled population. According to what they claim as a unique function of ENGRAM, I wanted to see sequential recording and readout of enhancer activities in a way that existing technologies cannot achieve. Environmental recording with dox, NFkB, and Wnt have been well demonstrated by other studies using similar BE-based circuits. The author states the success of recoding the order of signaling events in Figure 4. However, I believe this does not showcase the ability of the system to record sequential events as they transfected circuits in an arbitrary order.
2. The fourth paragraph of the introduction does not flow well. They talked about SSR-based clone labeling (like Brainbow, needs to be cited) and then suddenly CRISPR-based event recording like

CAMERA. I was a bit confused why only these topics were introduced here. DOMINO needs to be cited.

3. The authors stated the previous CRISPR or BE-based system requires the pairing of gRNAs with their corresponding target sequences. This is not true. For example, homing gRNAs or stgRNAs can achieve the recording of events without such a constraint.

4. “For example, in the case of SSRs or the CAMERA proof-of-concept, enhancers are used to selectively drive the enzyme that mediates an alteration in DNA sequence. However, in this framing, each signal requires its own enzyme, and it is difficult to imagine how more than a handful of independent signals could be concurrently recorded within the same cell or population of cells... “ This seems a little overstated. Given that arrays of gRNA and their respective target site can be synthesized (Kim et al 2020 Nature Biotechnology) and gRNA can be expressed under the expression of pol2 promoters, we can imagine that multiplexed event recording is possible by coupling MPRA with the gRNA-based event recording strategy introduced in CAMERA.

5. “However, as noted above, MPRA depends on targeted RNA-seq of the barcodes, which is destructive and static.” Yes, but again, the paper did not obtain any experiments which cannot be obtained by MPRA.

6. Many of the statements are not supported by data—for example, Figure 1: “(a) Schematic of ENGRAM. Endogenous or synthetic cis-regulatory elements (CREs) drive activity-dependent transcription of a prime editing guide RNA (pegRNA) encoding a CRE-specific insertion. The insertion is written to a natural or engineered recording site within genomic DNA (“DNA Tape”). The recorded signal can subsequently be recovered by DNA sequencing or, potentially, by fluorescent in situ hybridization (FISH).” No endogenous CRE activity recording was demonstrated. No natural recording tape was implemented. No FISH-based readout was demonstrated (their recording sequence is only 5 or 6 nt long).

7. Although I liked Figures 1 and 2 and Supplementary Figures 1 and 2 and appreciated their careful experimental design and analyses, I was confused about the part where they attempted to reduce the background activity of ENGRAM. In Figures 2b-d, they analyzed the background activities of 5' Recorder, 3' Recorder, and 3'-FT Recorder and stated that 5' and 3'-FT recorders were optimal. However, they did not measure their enhancer recording activities side by side. They could be just overall diminished.

8. Although the authors hypothesized that these circuits make auto-regulatory negative feedbacks, another possibility is that the lower expression of Cys4 by the enhancer-minP just reduces functional pegRNA molecules.

9. The overall recording activity was only a few percent. The authors use the measurement unit of % of insertions (Figure 2f) or insertion proportion (log2). For me, this looked like an attempt to stay readership attentions from the low recording activity. They should be %inserted read and insertion proportion (log10). What is the absolute recording efficiency of each insert in Figure 1d?

10. “The intent of this non-targeting pegRNA was to compete for PE2 binding and thereby minimize any “pre-integration” activity from pegRNAs encoded by the ENGRAM recorders.” How was the integration confirmed? No transposon control? Can they employ lentiviral integration?

11. Maybe the enhancer recording library can be transfected into different cell lines like K562, HEK, and fibroblast cells to see if ENGRAM can capture cell line-specific enhancer activities.

12. Figure 3 needs more supportive evidence with proper controls. From Figure 3h, they stated the editing efficiency was proportional to the duration of ligand exposure. Did they establish the cells first all with the expected circuits? Is there a possibility that the cells with the expected edits

propagate better as they tend to have all the expected plasmids with necessary selection markers? The trend could be seen just by diminishing of cells that do not have all of the plasmids. The proper control experiments would be done by giving different exposures to different samples but after removal signals at different time points, all cell culture samples should be harvested at the same time (unifying of total cell culture/selection time).

13. It is unclear if the endogenous activity falls within the dynamic range in which the ENGRAM was shown to be functional or if the dynamics of endogenous activity fall within the time frame tested in Figure 3.

14. The authors integrated ENGRAM pegRNA recorders for TET-On, NFkB, and Wnt using piggyBac transposition in a population of HEK293T PE2 expression cells that had previously been derived by single-cell isolation. The authors did not do another round of single-cell isolation and select for cells that have all three co-integrated recorders but rather performed the experiment on a bulk population of integrated cells. As such, based on the current experimental design, it is difficult to say that each recorder is “competing to write to a shared DNA tape.” The authors could instead express all three ENGRAM pegRNAs from a single cassette.

15. Figure 4b is great but the following demonstration is not a recording of event artificial signals as they just tested different orders of pegRNA operons. The functionality of the cells with the pre-installed circuit should be tested. Furthermore, although they start this section by stating “in context of a multiplex signal recorder”, this section does not demonstrate the multiplexability of their technology that is stressed in the manuscript. Although the initial recording tape is not event-dependent, after the first edit, the proceeding pegRNA that can edit the tape becomes unique for each prior event. Hence, even if it is only for 2 events, it is difficult to see how the system described in this section could be more multiplexable/advantageous as compared to CAMERA or DOMINO.

16. Supplementary Figure 4a was interpreted as a result of crosstalks, but for me, this is a too strong assumption. They could be derived just by background activities.

17. Recording the order of CREs would be more impressive.

Minor comments:

18. “We speculated that this was due in part to the accumulation of background edits due to constitutive expression of Csy4, which may also have reduced growth of the monoclonal cell line due to cytotoxic effects” It would have been better if they tested other published means of expressing pegRNA from pol2 promoter such as through utilizing tRNA processing (Knapp et al 2019 Nature Communications). Or Ribozyme? Also, it would benefit the community how their system has an effect on cell viability and endogenous gene expression.

19. PE3 in the original PE paper or epegRNA efficiency is better than PE2. Is there any reason they used PE2?

20. Figure 3e needs color legends

21. Figure 2g: “1 barcode each”? or did you mean “2 barcodes each”?

22. Supplementary Figure 1a: Typo in the y-axis

23. Supplementary Figure 1e: The x-axis needs to be properly labeled.

24. The plasmid constructs should be deposited to the Addgene.

25. The sequencing data should be deposited to the NCBI SRA and provided with accession number.

Referee #2 (Remarks to the Author):

The submitted manuscript surely wins the 2021 award for creative toolbuilding in putting together new modalities of targeted genome engineering, new enzymes that target RNA, and next-generation sequencing to advance the rapidly growing field of using the cell's own genome to record a biological phenomenon – in this case, gene transcription in mammalian cells. As the authors point out, this is not a new idea, and many elegant studies have been done before Cas-driven editing was invented (in addition to the mouse literature the authors cite they may consider some beautiful examples from the yeast literature, eg <https://pubmed.ncbi.nlm.nih.gov/19543267/>).

The submitted manuscript describes in sometimes excessive – for a journal with a broad audience such as “Nature” – detail how this tool, thoughtfully named ENGRAM (see comment below), was optimized, and then put it to its key use: recording of multiple transcriptional outputs in multiplex. These data are shown in Fig 4 and lend a general sense of enthusiasm for the broad interest of this approach. In equally broad strokes, we remain woefully ignorant of how human gene regulation actually works (despite having enormous catalogs of component parts), and a tool such as ENGRAM could be a useful and powerful addition to our arsenal.

There are two gaps. The first – it's unclear how good of a ranking tool ENGRAM actually is? – can be addressed by reanalyzing the data in Figure 2H. The second is – in the opinion of this reviewer – key to demonstrating the utility of this approach to study biology of gene regulation at native loci, and requires additional – technically trivial – experimentation.

Major comments

Fig 2h represents a key dataset. ENGRAM “records” transcription indirectly: enhancer activates promoter, Csy4 is translated off the mRNA driven by the enhancer-promoter combination, then cleaves the transcript, the transcript arms the prime editor to edit the target, the editing takes place (in a way that is biased by the sequence of the barcode), and then NGS measures the % reads with and without enhancer. Every single step in this process potentially has a nonlinear component. To show correlation with actual level of transcription using a log₂/log₂ scatterplot for 300 enhancer constructs is simply not the appropriate way forward here. The authors need to present a more detailed and systematic analysis. Specifically: what exactly can the assay detect in terms of relative enhancer activities? The simplest way to do this is to extract 8 random enhancers from each quartile of normalized expression (by RNA), rank them using RNA levels, and next to those 8 bars of data show the cognate ranking by ENGRAM. The authors have all those data, it is simply a matter of presenting them in way where the reader can for themselves ascertain whether they should be using ENGRAM to study relative enhancer activities.

Every experiment in the manuscript is done on reporters – randomly integrated via PiggyBack, no less. This is, in plain terms, a cross between an a necessity-driven compromise and an atavism. The genome regulatory literature is equal-parts replete with massively parallel reporter assays for regulatory element activity and data showing such reporters offer modest insight into what actual enhancers do in their native context. The advent of genome editing has rapidly relieved the field of the yoke of random integration and enhancer fragments removed from their native context. It is technically trivial for the authors to use plain-vanilla editing to integrate their Csy4-pegRNA tandem into stop codons of any number of native silent human genes (Manu Leonetti at the Biohub and Ru Garawande at the Allen have tagged near-everything at the native locus), and then activate them. A conceptually trivial set of targets for this would be genes regulated, for instance, by type 2 NHRs,

where baseline level of activity is essentially zero. It would be, in the opinion of this reviewer, important that the authors add a Fig 5 to the manuscript in which they take 2-3 genes responsive to NF- κ B, or Wnt, or the thyroid hormone, integrate their reporter into the stop codon of those genes with a trivial 2A, and see how well the recording works. One does not need a single-cell-derived clone (which the authors refer to – in a puzzling lexical crossover with the field of immunology – monoclonal) line for this effort; – a pool with a sufficiently high level of the “gene->reporter” allele would suffice. This simple experiment – if successful – would make a bona fide impactful addition to the field of studying human gene control.

Minor

The acronym ENGRAM elegantly evokes the orthographically congruent term from neuroscience. The authors may wish to point that out explicitly to those readers who are unfamiliar with what that word means.

The authors – appropriately so – devote a considerable length of time on pp 2-3 to comparing ENGRAM to other methods and then pointing out – using a traditional “However” clause – why ENGRAM is better. This is appropriate but makes for difficult reading. The authors may consider relegating all that to one paragraph in the Discussion, eg “The field has developed methods X, Y, and Z. In contrast, ENGRAM attains X’, Y’, and Z’.”

Fig 1a is confusing – the text refers to a “common DNA tape,” Fig 1a shows three such loci / tapes. Further, the idea to use Csy4 is elegant but Fig 1b is confusing. The authors should consider having a figure that explains, at a high level, both the system and the process of recording. Ie, here is the reporter construct (a given enhancer drives expression of a transcript that is processed, by Csy4, to yield a “liberated pegRNA”); the cell contains a prime editor that binds to that pegRNA and inscribes the cognate edit into a recording locus. Sequencing the locus determines whether the enhancer was active, and if there are multiple enhancer constructs in the same cell, or pool of cells, you can measure their relative activities. Essentially the figure needs to represent the first 4 sentences of the Discussion section. The current Fig 1a does not entirely do ENGRAM justice – certainly not for a broad audience.

Fig 1d shows that hexamers are recorded at vastly different efficiencies – it is unclear at this point in the manuscript what implications this experiment has for subsequent work shown. A way to address this would be to say – “we rigorously controlled for this bias in subsequent experiments.”

Supp fig 1 refers to 4% editing as being of “reasonable efficiency.” Without wishing to quibble over semantics, in December 2021, 4% editing in transformed cells should not be described by the epithet “reasonable” – 40% would be. Vide infra for implications of this low efficiency.

Fig 1f shows the dynamic range of recording in this 1.0 version of the ENGRAM system is a low one – as the authors correctly state. The explanation given – “due in part to the accumulation of background edits due to constitutive expression of Csy4, which may also have reduced growth of the monoclonal cell line due to cytotoxic effects” – is hard to understand and may be best omitted. The efficiency of editing is due to the amount of pegRNA (one presumes) – which, in turn, is due to transcriptional activity of the minimal promoter with and without enhancers. Surely it’s the latter that is the most parsimonious reason for the “considerably more modest S/N” – rather than cytotoxicity?

The system optimization described in the first two paragraphs of page 7 is of interest only to a specialist audience and can be relegated to supplementary information.

The purpose of the effort in Fig 2 was to address the problem of low dynamic range shown in Fig 1f. Why are the data in Fig 2f represented using a different metric - % insertions – than Fig 1f (editing

with and without enhancer)? Are the data cognate in Fig 1f – in supp fig 2b?

“To push the system further” is inappropriately colloquial and imprecise.

Could the authors briefly clarify what they mean by “ORI-driven transcription” on p 10? Plasmids lacking viral origins of replication (eg EBV) do not replicate in mammalian cells using defined origins (ORI) of replication unless a factor like EBNA is present. In other words, generic plasmids do not have an “ORI” that functions as such in mammalian cells. Do the plasmids being used have such ORIs? Or is ORI an acronym for something else?

While the sigmoidal curves in Fig 3b are impressive, as the authors know, a difference between 1.5% and 3% editing is not what one wants for an editing-based readout (also vide supra). The actual dynamic range of editing by NGS is between 0.5% (amplicon resequencing without UMIs) and 95% indels (presumably for PE this is lower). It’s impressive the triplicates (unclear if biological – ie different experiments start to finish, or NGS in triplicate) are robust, but for the average user of this method, working with NGS “dynamic ranges” between 1.5% and 3% will be very hard. How many reads per amplicon did the authors have to use in order to get the numbers they got?

The interesting experiment in Fig 4 is missing a key technical bit of editology: are the authors counting individual haplotypes (ie alleles in a given cell) for evidence of Boolean logic such as shown in Fig D? If yes, can the authors show details of the “shared DNA tape”? Where is the gRNA binding site, what are the sequences the pegRNA for each signal encode, and what happens when multiple pegRNAs compete, in the same cell, to write different alleles of the DNA tape? Conversely, are the individual recordings one and done, and in this case, the authors are reading this out on a pool level (ie counting stimulus-specific barcodes)? This is confusing even for this reviewer – who is familiar with the underlying technologies – and the authors should clarify.

// Fyodor Urnov – Innovative Genomics Institute and UC Berkeley //

Referee #3 (Remarks to the Author):

This is a very creative and elegant new approach to record the activity of selected regulatory elements (and hence their upstream pathways). Enhancer activities are recorded and integrated over time and intensity, and a proof-of-principle experiment demonstrates that it is also possible to record the temporal order of signalling events. The method is highly multiplexable.

Strengths of the manuscript are its high conceptual novelty, the careful testing and iterative optimisation of the methodology, the overall high quality of the data, and the clarity of the text and figures. This new approach will inspire other scientists to think along similar lines and perhaps develop further variants of this technique.

But for publication in Nature, I feel that something is missing. While the ENGRAM technology is very cleverly designed, the manuscript does not really demonstrate its practical utility for solving new biological problems. The manuscript remains very conceptual; no new biological insights are obtained. I took some time to think about this, but I cannot think of a biological question in cultured cells that cannot be addressed equally well (or even better) with conventional MPRAs. But I would be happy to be convinced otherwise, by one or more clear examples.

It seems to me that the real promise of ENGRAM lies in future applications in transgenic animals, to study signalling and gene regulation during morphogenesis and tissue development, in growing tumors, various disease models, etcetera. However, as pointed out by the authors in the Discussion, multiplexing would require integration of large sets of recorders into the genome ("a recorder locus"), and as yet this is still a major challenge.

The authors also mention that ENGRAM may record "cellular histories". This sounds exciting, but again there still seems a big gap between the promise and the realisation. Related to this, I am a bit worried about the intrinsic property of ENGRAM that it integrates signals both over time and intensity (Figure 3j-k). This makes it difficult to reconstruct how a signal of medium strength arose: was the enhancer stably active at moderate levels, or was it very active but only part of the time? Both scenarios could give the same output. To deconvolve this, one would have to do time series experiments - but then the added value over standard reporter assays seems lost?

Thus, this is an elegant and interesting new technology, but what will it be used for in the foreseeable future, and how does it outperform existing technologies? I would not hesitate to recommend publication of this work in *Nature Methods* or *Nature Biotechnology*. Readers of these journals will be inspired by the technological innovation, even if the immediate advantages over existing tools are not obvious yet, and no new biological insights are included as evidence. But for publication in *Nature*, I feel that one or more concrete examples are needed to demonstrate that new biological problems can be solved that could not be solved by MPRA or other reporter assays.

Other major points:

The text is easy to follow, and in principle it is very informative to learn about the sequential improvement steps that the authors went through. But I do not think that this chronological and rather technical account is suitable for the broad readership of *Nature*. If the Editor decides to move forward with this manuscript, I recommend a major rewriting/shortening of the text, focusing on the well-working versions of the method. The chronological account of the step-by-step improvements is only important for readers who want to implement the method themselves, and should therefore be provided as supplementary info.

The Materials and Methods section is far too minimal. Since this manuscript describes an entirely new method, a much more detailed protocol should be provided.

I was unable to find the datasets. For example, a list of all tested enhancers and their precise sequences, and the data of Figs 2h and SuppFig 2.

Minor points:

Figure 1a: The use of FISH as a readout seems too optimistic to me. To my knowledge, DNA FISH still

does not work for such single, very short sequences.

Please define term "recorder" as used throughout the text. Is it the vector that drives pegRNA expression, or its combination with the prime editor vector? Or does it also include the DNA tape? Please check that the term is used consistently.

It was a good idea to reduce background by putting Csy4 expression under control of the same enhancer as the pegRNA. However, this background reduction may fail in multiplexed setting, as the Csy4 expression will be driven by many enhancers. What if one of these enhancers activates at an early time point -- will this mean higher background signals for the recording of the enhancers that activate later (or not at all)? Please comment on this potential problem.

In the multiplexed setting of Figure 2g/h, I understand that the editing of the DNA tape is competitive. In this setting, what would happen if enhancer A is active earlier than enhancer B? Would the editing driven by B erase some of the edits driven by A? Please comment on this.

Please clarify in the main text: for the multiplexing with 300 enhancers as in figure 2h, how many cells were needed? How does this compare to the number of cells needed for measuring the activity directly from barcoded RNA (i.e. a conventional MPRA)?

===== Reviewed by Bas van Steensel. Assignment accepted: 26 Nov 2021; completed: 28 Nov 2021. It is my standard policy to sign and date **all** of my manuscript reviews, irrespective of my comments and recommendations. All correspondence should go via the editor. PLEASE DO NOT REMOVE THIS NOTE. =====

Response to Reviewers

We thank the three reviewers for their highly constructive feedback on the submitted version of the manuscript.

We also apologize for the extensive delay in the development and submission of this revision, which was due in part to personal circumstances.

In this point-by-point response, the original reviewer comments are in **blue text**, while our responses are inline in **black text**. Referee #1: pages 2-26; Referee #2: pages 27-46; Referee #3: pages 46-57.

A summary of the major new experiments or analyses is as follows:

1. We now demonstrate ENGRAM-based temporal recording of biological signals via an approach that has much greater potential than the strategy previously presented. Specifically, we successfully combined ENGRAM with DNA Typewriter, a related technology recently developed by our lab for time-resolved, multi-symbol molecular recording, to record and subsequently recover the temporal order of biological signals. See new **Figure 4, Supplementary Figure 7** and corresponding additions to text. The combination of ENGRAM and DNA Typewriter provides a clear path forward for the time-resolved, multi-channel recording of biological signals, potentially in combination with cell lineage.
2. We added several experiments that could not have been done with MPRAs. This includes the aforementioned integration of ENGRAM and DNA Typewriter to longitudinally record the order of biological signals within the same cells, and also new experiments in which we perform multiplex “integration” of the activity of synthetically designed CREs over 24-hr intervals collectively spanning a differentiation time-course of mouse embryonic stem cells (mESCs) into gastruloids, a multi-germ-layer organoid model of early mammalian development. See new **Figures 4-5, Supplementary Figures 7-8** and corresponding additions to text. In the revised Discussion, we have sought to better emphasize key distinctions between ENGRAM and MPRA)s, including that: (a) only ENGRAM permits integration of transient signals over time; and (b) only ENGRAM permits correlation of present state with past state(s) within the same system.
3. As requested by the reviewers, we performed a new experiment in which we record (in multiplex) the cell type-specific activities of synthetically designed CREs, comparing K562s, HEK293Ts and mESCs, and observe clear, reproducible differences. See new **Figure 2g**, revised **Supplementary Figure 5**, and corresponding additions to text.
4. We tested the efficiency of ENGRAM with alternative or optimized prime editing components, including tRNA-based processing, an enhanced pegRNA backbone (“epegRNA”) and an optimized prime editor enzyme (“PEmax”), and observe substantial improvements with PEmax in particular (a 1.7-fold and 9.3-fold increase in editing efficiency with PEmax over PE2 in K562 cells and mESCs, respectively). See **Supplementary Figure 1** and corresponding additions to text.
5. We performed a new experiment directly inspired by, although not precisely corresponding to, a request from Reviewer #2, in which we randomly integrated “minimal” ENGRAM recorders into the native genome. i.e. such that individual recorders were embedded within the gene bodies of thousands of native genes, or alternatively at intergenic locations. However, their behavior in these contexts was NOT as expected. For the time being, we report them only here and have left them out of the revised manuscript. However, we are open to including them as a Supplementary Note pending the reviewer’s input.

6. We updated our analyses and presentation of the 300-plex CRE recording data along the lines suggested by Reviewer #2. See **Figure 2, Supplementary Figures 4 & 5** and corresponding additions to text.

Other major changes include:

1. We have sought to substantially tone down the manuscript with respect to the concern that we are overselling it relative to the experiments shown.
2. We have shortened the description of the development and optimization of ENGRAM, moving details of early versions to the supplement. We also consolidated Figures 1 & 2 from the original paper into a single figure. See **Figure 1** of the revised manuscript.
3. We have substantially expanded the **Methods** section with respect to details of both experimental and computational analyses. We have also added more details in the form of Jupyter notebooks to ensure the reproducibility of computational analysis and figure production. Finally, we added plasmid maps for most of the recorders that we describe in the manuscript. Both the Jupyter notebooks and plasmid maps are at: <https://github.com/shendurelab/ENGRAM>

Referee #1 (Remarks to the Author):

Chen et al. demonstrated a proof-of-principle DNA event recording system by harnessing a massively parallel reporter assay (MPRA) and CIRPSR Prime Editing (PE). In brief, their system, termed ENGRAM (Enhancer-driven genomic recording of transcriptional activity in multiplex), places multiple prime-editing guide RNAs (pegRNAs) sharing the same target sequence but with different small insert sequences which are molecular identifiers under the expression of corresponding enhancer element. ENGRAM uses a previously described gRNA processing strategy (Cys4 system) to mature pegRNAs from their enhancer-driven transcripts and confer editing. The authors also carefully tested various ENGRAM systems to derive an optimal 5' Recorder system. They demonstrated the 5' Recorder system to record the activities of ~300 enhancer regions in K562 cells and the correlation between the recorded DNA readouts and RNA sequencing-based readouts. While the analyses in Figures 1 and 2 seemed to be designed very carefully, Figures 3 and 4 are a bit rough. I understand how an ideal DNA-based recording would revolutionize biology, share the vision with the authors and realize that this is one of the first proof-of-demonstration studies. However, none of the observations derived by ENGRAM in the present study is what we cannot achieve by the other current technologies. I also feel they oversell the possibility of ENGRAM without sufficient datasets. I will also address these points one by one below.

We thank the reviewer for these constructive comments. As outlined further below, in revising the manuscript, we have sought to better articulate the unique strengths of ENGRAM as well as to add new experiments showcasing its potential. We have also toned down our language throughout the manuscript in hopes of addressing the concern that we are overselling it.

Major comments:

1. "Overall, there remains a need for a means of capturing signaling and gene regulatory activity that is at once quantitative, reproducible, non-destructive, multiplexable, applicable to physically opaque biological systems

and capable of integrating the temporal dynamics of large numbers of signals.” I strongly agree with this point and this is the whole beauty of the recently emerging molecular recording concept. I also greatly appreciate that the authors carefully designed and established beautiful circuits.

We are glad that the reviewer appreciates the potential of DNA-based recording to revolutionize biology. We also thank the reviewer for the kind words about the experimental design of our circuits.

However, all the demonstrations were performed using cell culture. The CRE library recording was impressive but similar data can be obtained by RNA-seq of a subsampled population. According to what they claim as a unique function of ENGRAM, I wanted to see sequential recording and readout of enhancer activities in a way that existing technologies cannot achieve. Environmental recording with dox, NFkB, and Wnt has been well demonstrated by other studies using similar BE-based circuits. The author states the success of recoding the order of signaling events in Figure 4. However, I believe this does not showcase the ability of the system to record sequential events as they transfected circuits in an arbitrary order.

The first part of this comment focuses on the fact that the experiments described in the original submission were limited to static measurements in cell culture that could have been achieved by MPRA. This is similar to Comment #5 from this reviewer, and is therefore addressed further below.

Focusing on the remainder of this comment, the reviewer correctly points out that our serial transfections of pegRNA-encoding plasmids limits the extent to which we can claim to have demonstrated sequential recording of biological signals (as plasmid transfections are not biological signals). A related limitation, raised by Comment #15 from this reviewer, is that our original strategy for sequential recording does not readily scale to larger numbers of signals. In particular, our original strategy relied on pegRNAs that were “pre-programmed” to detect particular orders. More specifically, each signal drives multiple pegRNAs, but the activity of some of these pegRNAs are dependent on pegRNAs driven by other signals having already edited the DNA Tape. Although this strategy works well for a few signals, it has poor scalability, as the number of pegRNAs required to record all possible orders grows exponentially with the number of signals. The CAMERA and DOMINO methods demonstrate ordered recording, but are similarly constrained in that a template for each potential order must be pre-programmed. Pre-programmed circuits for detecting order may also be limited in terms of their ability to capture temporal dynamics of even two signals, *i.e.* duration, overlap, intensity, etc.

We recently reported a method called DNA Typewriter¹ for the ordered recording of insertional prime edits. With DNA Typewriter, the blank recording medium (multi-unit DNA Typewriter Tape or DTT) consists of a tandem array of partial CRISPR–Cas9 target sites, with all but the first site truncated at their 5’ ends and therefore inactive. Short insertional edits serve as symbols that record the identity of the pegRNA mediating the edit while also shifting the position of the ‘type guide’ by one unit along the DTT, *i.e.* sequential genome editing. In our published work¹ describing DNA Typewriter, we demonstrated that we can record the order of stochastically driven barcode insertions, with the goal of recording cell lineage.

In new experiments, we sought to combine ENGRAM and DNA Typewriter to record the sequential order of biological signals (see **Figure 4**, reproduced below, and corresponding additions to text).

Figure 4. Combining ENGRAM and DNA Typewriter to record the temporal dynamics of biological signals. (a) Signal-responsive ENGRAM recorders can be modified to drive the expression of DNA Typewriter-compatible pegRNAs (DTT-pegRNA), each encoding signal-specific insertions to be written to the current type guide location of a multi-unit DNA Typewriter Tape (DTT). The insertions record the identity of the signal but also include a ‘key’ that shifts the location of the type guide by one position. Here we modified the pegRNAs encoded by Tet-On (orange) and Wnt (blue) ENGRAM recorders for DNA Typewriter compatibility. Temporal dynamics (in the simplest case, the order of two serially applied agonists) should be captured in the order in which the corresponding symbols appear in DTT. (b-c) These modified ENGRAM recorders and 5-unit DTT were sequentially integrated to PEmax(+) HEK293T cells. We designed and tested

serial **(b)** and layered **(c)** programs in which these cells were exposed to different patterns of 100 ng/ml doxycycline or 3 μ M CHIR-99021 (left columns), as described in the text, across a total of 36 cell populations (2 patterns * 3 intervals * 2 possible orders * 3 replicates). Harvesting, amplifying and sequencing the DTT region after 6 days, we calculated the log₂ ratio of [Tet-On→Wnt] vs. [Wnt→Tet-On] bigrams at Sites 1-2, predicting and observing positive values when Tet-On activation preceded Wnt activation, and negative values when Wnt activation preceded Tet-On activation (right columns). Error bars correspond to standard deviations across 3 transfection replicates. **(d)** Same as panels b-c, but for pulse programs in which these cells were exposed to 500 ng/ml doxycycline for 24 hrs against the background of continuous stimulation with 3 μ M CHIR-99021. A total of 9 cell populations are represented (3 pulse timings * 3 replicates). **(e)** PCA based on proportions of unigrams and bigrams observed at each of 5 DTT positions and 4 DTT position-pairs, respectively, across 45 cell populations subjected to various patterns of exposure to doxycycline and CHIR-99021 (15 programs, executed in triplicate). Circled subsets correspond to serial and layered programs in either order, or to pulse programs. The top three PCs are plotted, which collectively explain 90% of the variance.

In brief, in these experiments, Wnt and TetOn ENGRAM recorders are driving pegRNAs that each encode a signal-specific barcode to be written to DTT, but also the key sequence that shifts the type guide forward (**Figure 4a**). All relevant components (ENGRAM recorders, PEmax, DTT) were first integrated to the genomes of HEK293T cells. Activation of the Wnt and TetON ENGRAM recorders were controlled by the chemical ligands CHIR and doxycycline (Dox), respectively. We then executed a variety of programs in which these chemical ligands were added or removed in different temporal patterns.

In “serial” programs, the first and second signals are not overlapping, but their order and duration is varied (**Figure 4b**, left). In the “layered” programs, the second signal is added in, after some delay, to a constitutive first signal, with both the order and duration of delay varied (**Figure 4b**, right). In “pulse” programs, the TetON signal is added and then withdrawn, in the background of a constitutive Wnt signal, but the timing of the pulse is varied (**Figure 4d**).

Altogether, we tested 15 temporal programs, each in triplicate (45 experiments total). Within the serial and layered programs, we were readily able to determine which signal was introduced first, simply based on the ratio of DNA Tapes bearing the Wnt-specific barcode before vs. after the TetON-specific barcode (TetOn→Wnt / Wnt→TetOn in adjacent units of the DTT; **Figure 4b,c**). Moreover, the magnitude of this ratio was informative with respect to the length of time before the second signal was introduced, *i.e.* longer intervals yielding a greater signal-to-noise. Although we only tested intervals down to 24 hrs, the data suggests that we could likely distinguish orders for even shorter intervals, particularly in the case of serial signals (**Figure 4b**). Similarly, in the case of pulse programs, this same ratio was highly informative with respect to whether the second signal was pulsed early, midway or late in the experiment (**Figure 4d**).

On the other hand, this ratio in isolation was not sufficient to distinguish different pattern classes from one another, *e.g.* serial vs. layered patterns. To assess whether all 15 programs were distinguishable on the basis of the ensemble of patterns observed in DTT, we performed principal components analysis (PCA) on 26 variables observed in each of 45 treatments (15 programs * 3 replicates each). These included the proportions of the two symbols at each of the five DTT sites (2 symbols * 5 positions) and the proportions of each possible bigram (Tet-On→Tet-On, Tet-On→Wnt, Wnt→Tet-On, Wnt→Wnt) at pairs of adjacent sites (4 bigrams * 4 position-pairs). Upon plotting the top three PCs, which together explain 90% of the variance, we observe that replicates are tightly clustered (**Figure 4e**). This suggests that the 15 temporal programs of signal transduction

give rise to reproducibly distinct ensembles of symbol patterns, as recorded to 5-unit DTT via the combination of ENGRAM and DNA Typewriter.

In summary, these experiments demonstrate how the combination of ENGRAM and DNA Typewriter enable a diversity of temporal patterns involving two distinct signal recorders to be distinguished via the symbols that are written to a common substrate, DTT. DTT is fundamentally different from other forms of multi-unit DNA-based recording media in that it is written to in an ordered fashion, with the activation of site $n+1$ depending only on site n having been written to but not on what symbol was written to it. Of course, there are limitations to what we have achieved so far in combining ENGRAM and DNA Typewriter. We relied on an ensemble of cells, rather than a single cell, to extract each temporal signal pattern, and we have yet to develop algorithms or models that would allow one to directly infer signaling histories from a newly observed, complex patterns (as opposed to simply showing after the fact that the patterns are distinct, as we do in **Figure 4e**). On the other hand, these experiments only partly showcase the full potential of this combination, as vastly more signals and orders could potentially be accommodated by this same framework. By introducing and recovering more DNA Tapes into each cell, we can also imagine how the number of cells required for complex pattern inference might be greatly reduced. We also envision that the combination of ENGRAM and DNA Typewriter might facilitate the concurrent recording of biological signals and cell lineage, *i.e.* with both stochastic and biologically conditional recorders all writing in an ordered fashion to a shared set of DNA Tape(s).

Altogether, we hope that the reviewer agrees that this combination of ENGRAM and DNA Typewriter unlocks the path to a bright future for DNA-based, time-resolved, multi-channel recording of biological signals.

2. The fourth paragraph of the introduction does not flow well. They talked about SSR-based clone labeling (like Brainbow, needs to be cited) and then suddenly CRISPR-based event recording like CAMERA. I was a bit confused why only these topics were introduced here. DOMINO needs to be cited.

We apologize for missing key citations here. We have now added citations for both Brainbow and DOMINO, and we have also sought to separate out our summaries of SSR- and CRISPR-based systems more clearly as separate paragraphs. The modified text is reproduced below:

“DNA is the natural medium for biological information storage and is easily “read” by sequencing. Various enzymatic systems have been used to alter genomic DNA in a signal-responsive manner, most prominently site-specific recombinases (SSRs)^{2,3} and CRISPR genome editing⁴. For example, the Cre SSR is typically expressed under the control of a cell type-specific enhancer. In cells or tissues where that enhancer is active, Cre-mediated recombination at a target locus excises sequences between pairs of *lox* sites, resulting in the expression of a fluorescent reporter (or combinations of fluorescent reporters, in the case of Brainbow)⁵⁻⁷. The enzymatic event is irreversible, *i.e.* the descendants of those cells in which the DNA rearrangement occurred continue to express the reporter regardless of whether the gating regulatory element remains active or not.

CRISPR genome editing has also been adapted to signal-specific recording. A first class of methods repurposes the CRISPR-Cas spacer acquisition system to “log” events or content in prokaryotic systems, *e.g.* DNA, RNA or metabolites⁸⁻¹². A second class of methods, including CAMERA and DOMINO, links the activity of CRISPR base editors to the presence of specific small molecules, or to the activity of specific signaling pathways, such that observed base edits serve as a stable, heritable record of those signals^{13,14}.”

3. The authors stated the previous CRISPR or BE-based system requires the pairing of gRNAs with their corresponding target sequences. This is not true. For example, homing gRNAs or stgRNAs can achieve the recording of events without such a constraint.

The reviewer is accurate that homing gRNAs or stgRNAs do not require the separate introduction of a target sequence, because the gRNA-expressing cassette also serves as the target. However, the limitation that we were attempting to articulate (albeit awkwardly) is still there, because in trying to “read” each signal, there is still a separate target sequence for each homing gRNA. In contrast, with ENGRAM, signal-specific insertional edits driven by different guides are written to a shared target, facilitating their ability to be efficiently read/recovered with sequencing. For example, if dozens of homing gRNAs were used for different signals, each writing back to the genomic locations from which they were expressed, recovering recorded events from so many such sites by methods such as single-cell RNA-seq (sc-RNA-seq) would be very challenging. In contrast, with ENGRAM, events driven by any number of signal-specific pegRNAs can all be written to a shared target site or compact array of target sites (or, as we now show, the ordered recording substrate of DNA Typewriter), which makes for much more straightforward, optimizable recovery by sc-RNA-seq or related methods. Furthermore, it enables relative signal strengths across experiments to be quantitatively compared in an internally normalized fashion, *i.e.* ratios of signal-specific insertions (**Figure 3**) or insertion orders (**Figure 4**) at the shared target site. With the goal of being more accurate while still trying to make this point, we have revised the text to read as follows:

“However, contemporary SSR and CRISPR-based molecular recording systems are sharply constrained with respect to multiplexability, *i.e.* the number of independent signals that can be recorded at once. For example, in the case of SSRs, a CRE is used to selectively drive the enzyme that mediates an alteration in DNA sequence. However, each signal requires its own enzyme, and it is hard to imagine, with SSRs, how more than a handful of independent signals could be concurrently recorded within the same cell, let alone how extensive, concurrent recording of a very large number of biological signals could be achieved throughout the development of a multicellular organism. Also, SSR-based recording systems do not readily capture the temporal order in which multiple signals of interest occurred.

In the case of CRISPR-based molecular recording systems such as CAMERA and DOMINO, Pol-II promoters can be used to drive single guide RNA (sgRNA) expression¹⁵, such that, at least in principle, multiple signal-specific reporters could be deployed for molecular recording within the same cell, *i.e.* by leveraging a separate set of CRE, guide RNA, and target site, per signal^{16,17}. However, in practice, such multiplexing would be limited by the fact the information is effectively captured by the location(s) of the edited site(s), rather than the nature of the edit itself. Even with homing or self-targeting guide RNAs^{18,19}, the “write” events corresponding to each signal would occur at different locations in the genome. This limitation is particularly evident in the context of temporally ordering multiple signals, as each possible order (the number of which scales exponentially with the number of signals) must be “pre-coded” as an editable template^{13,14} (**Supplementary Table 1**). In an ideal multiplex biological recording system, distinct signals would be written in an ordered manner to a shared genomic location, such that signal-specific edits, and their temporal dynamics, could be efficiently recovered via methods such as single cell RNA-seq (scRNA-seq).”

Towards conveying some of these nuances, we also added a new table (**Supplementary Table 1**, reproduced below) that compares features of traditional CRISPR cut-based recording, base editing-based recording, and prime editing-based recording. We certainly welcome feedback on the content and clarity of this new table:

Supplementary Table 1 Comparison of strategies for CRISPR-based molecular recording.				
		CRISPR cut	Base editing	Prime editing
Writing	Editing modality	Cas9 nuclease	Base editor	Prime editor
	Other writing components	gRNAs + targets OR homing gRNAs/stgRNAs	gRNAs + targets OR homing gRNAs/stgRNAs	pegRNAs + shared target
	Information content of edit sequence	pseudorandom	pseudorandom	signal specific
	Potential for recording temporal order of biological signals	Poor (current methods do not facilitate ordered recording)	Moderate (system complexity grows exponentially with number of biological signals, because guides/targets must be preprogrammed for each order)	High (system complexity grows linearly with number of biological signals, with ENGRAM + DNA Typewriter)
Storing	Signal location	Signals write to distinct sites or sets of sites	Signals write to distinct sites or sets of sites	Signals write to a shared site or sets of sites
Reading	Ease of recovery of target site(s)	Poor (target site(s) corresponding to each signal must be recovered separately; further challenge of DSB-mediated deletions deleting information and/or compromising primer sites)	Moderate (target site(s) corresponding to each signal must be recovered separately)	High (shared target site(s); compatible with sc-RNA-seq)
	References	(McKenna et al. 2016; Perli et al. 2017; Frieda et al. 2017; Chan et al. 2019; Bowling et al. 2020)	(Tang and Liu 2018; Farzadfard et al. 2019)	This paper

4. “For example, in the case of SSRs or the CAMERA proof-of-concept, enhancers are used to selectively drive the enzyme that mediates an alteration in DNA sequence. However, in this framing, each signal requires its own enzyme, and it is difficult to imagine how more than a handful of independent signals could be concurrently recorded within the same cell or population of cells...” This seems a little overstated. Given that arrays of gRNA and their respective target site can be synthesized (Kim et al 2020 Nature Biotechnology) and gRNA can be expressed under the expression of pol2 promoters, we can imagine that multiplexed event recording is possible by coupling MPRA with the gRNA-based event recording strategy introduced in CAMERA.

Apologies for our lack of clarity in this passage. We have deleted “or the CAMERA proof-of-concept” from the quoted text, such that it is clearer that the limitation cited is solely in relation to SSR-based recording. We have also added citations to both Kim et al. Nature Biotechnology (2020) and Knapp et al. Nature Communications (2020) in some new text written in response to this reviewer’s comment #3 (see above), as part of a more generally updated introduction. We welcome further feedback on whether we have better hit the mark.

5. “However, as noted above, MPRA depends on targeted RNA-seq of the barcodes, which is destructive and static.” Yes, but again, the paper did not obtain any experiments which cannot be obtained by MPRA.

The reviewer agrees with us with respect to the cited limitation of MPRA, but fairly points out that in the original submission, we did not show any results which could not have been obtained by an MPRA. To address this, we now present two sets of new experiments whose results could not have been obtained by an MPRA.

(1) The first set of new experiments involves the integration of ENGRAM and DNA Typewriter, which enables decoding of the order and/or timing of signals in the context of diverse temporal “programs”. For details, see our response to Comment #1 from this reviewer above, as well as new **Figure 4** and the corresponding additions to the main text. In these experiments, we are obtaining results which could not have been obtained via an MPRA. In particular, we observe patterns with a DNA Typewriter Tape that document the recording of one signal followed by another signal within the same cell. In contrast, even new single cell MPRA methods measure only a cell’s state at a single point in time, *i.e.* the moment of lysis and RNA purification.

(2) The second set of new experiments involves using ENGRAM recorders while differentiating mouse embryonic stem cells (mESCs) into gastruloids, a multi-germ-layer organoid model of early mammalian development. Specifically, we leveraged ENGRAM to “integrate” the cumulative activity of each of 98 synthetic CRE ENGRAM recorders within 24-hr windows collectively spanning a 5-day time-course of gastruloid induction. For these experiments, all recording components, including a dox-inducible PEmax (TRE-PEmax), a library of ENGRAM recorders (98 synCRE ENGRAM recorders), and a DNA Tape bearing the HEK3 target (synHEK3-Tape), were integrated into mESCs at high MOI (**Figure 5a**, reproduced below). These mESCs were then successfully induced to form gastruloids that were morphologically indistinguishable from conventional gastruloids, suggesting that the system’s components did not appreciably interfere with the model. The recording window was controlled by adding dox to the medium for a particular 24 hr window.

To evaluate dynamics, we compared the levels of recorded activity for each synthetic CRE, as integrated across each 24 hr post-induction window, against recorded activity in cultured mESCs (**Figure 5d**, reproduced below). This identified 17 of the 98 recorders as exhibiting significantly and substantially activity dynamics during one or more 24 hr windows of gastruloid differentiation (**Figure 5e**; **Supplementary Figure 8e-f**, both reproduced below; Wald-test with B-H correction $P < 0.1$ for a fold-difference > 2 relative to cultured mESCs). The dynamics of four major developmental signaling pathways—Wnt, Nodal, FGF and retinoic acid—were probed by subsets of the 98 recorders bearing motifs for their effector TFs. The other six differentially active ENGRAM recorders were driven by CREs bearing consensus motifs for core developmental TFs or TF families including CDX, GATA, PBX, PRDM4, SIX and SOX, suggesting the potential of recording lineage-specific activities with ENGRAM.

The experiment highlights the ability of ENGRAM to capture the cumulative activity of a signaling pathway or a TF over an extended period of time (*e.g.* here, 5 x 24 hr windows that cumulatively “cover” the full 120 hrs), something that is not possible with an MPRA. Of course, one could perform MPRA with an extremely large number of sampled timepoints, analogous to the infinitesimal sampling that eventually gives rise to the integral in calculus. However, this would be impractical, particularly for organoid models in which only a limited number of organoids can be reasonably cultured. For more details, please see the entirety of new **Figure 5** and **Supplementary Figure 8** (both only partially reproduced below), as well as the corresponding parts of the revised results section.

Looking further ahead, we envision that the combination of ENGRAM, DNA Typewriter and single cell RNA-seq, in dynamic multi-cell-type *in vitro* models such as gastruloids, as well as *in vivo* models such as mouse, will further reinforce the distinction between what is possible with “temporally integrative and ordered” multi-channel *recording* vs. MPRA-based single-time-point *reporting*.

Figure 5. Multiplex recording of CRE activity in embryonic stem cells and differentiating gastruloids. (a) Schematic of polyclonal ENGRAM mESCs, with each cell bearing multiple copies of doxycycline-inducible PEmax, ENGRAM recorders, and synthetic DNA Tape (*HEK3*). **(d)** Schematic of experiment in which polyclonal ENGRAM mESCs were differentiated to gastruloids, with windowed induction of PEmax across a specific 24 hr interval with doxycycline. The induction of mouse gastruloids starts with aggregation of ~300 mESCs in round bottom wells, which are subjected to 24 hrs of Wnt stimulation at 48-72 hrs post-aggregation. At 120 hrs post-aggregation, gastruloids are elongated and include multiple cell types, including somites, neural tubes, cardiomyocyte-like cells, and endoderm-like cells^{20,21}. For each of five 24-hr windows, PEmax was activated by adding 50 ng/ml doxycycline and differentiating gastruloids harvested 24 hrs later, followed by isolation of genomic DNA and amplification and sequencing of DNA Tape. Each recording window was assayed in duplicate. **(e)** Hierarchically clustered heatmap presenting recorded activities across each 24 hr interval (rows) for 17 of the 98 ENGRAM recorders with exhibiting significant and substantial differences (Wald-test with Benjamini-Hochberg correction $P < 0.1$ for a fold-difference > 2) in one or more of the five windows (columns). Values are log2-scaled barcode proportion ratios for gastruloids with windowed recording in a particular 24 hr interval vs. cultured mESCs. *, **, ***, represents $P < 0.1$, $P < 0.01$ and $P < 0.001$, respectively.

Supplementary Figure 8. Multiplex recording of CRE activity in embryonic stem cells and differentiating gastruloids. (e) Plot showing $-\log_{10}$ adjusted p-values of ENGRAM recorders (y-axis) across each 24 hr recording window (x-axis) during gastruloid differentiation. A total of 17 ENGRAM recorders exhibiting significant and substantial differences (Wald-test with Benjamini-Hochberg correction $P < 0.1$ for a fold-difference > 2) for a particular window, in comparison to recordings made from the same polyclonal ENGRAM mESCs under normal culture conditions. The 12

recorders with increased activity are labeled in red, and the 5 recorders with decreased activity are labeled in blue. **(f)** Heatmap presenting recorded activities across each 24 hr interval (rows) for 48 of the 98 ENGRAM recorders with substantial activity in any of the five windows (columns). Columns are hierarchically clustered for presentation purposes. Values are log₂-scaled barcode proportion ratios for gastruloids with windowed recording in a particular 24 hr interval vs. cultured mESCs. The 17 recorders whose activity exhibiting significantly and substantially increased (*) or decreased (^) activity in one or more of the 24 hr windows, relative to cultured mESCs, are bolded (Wald-test with Benjamini-Hochberg correction $P < 0.1$ for a fold-difference > 2).

6. Many of the statements are not supported by data—for example, Figure 1: “(a) Schematic of ENGRAM. Endogenous or synthetic cis-regulatory elements (CREs) drive activity-dependent transcription of a prime editing guide RNA (pegRNA) encoding a CRE-specific insertion. The insertion is written to a natural or engineered recording site within genomic DNA (“DNA Tape”). The recorded signal can subsequently be recovered by DNA sequencing or, potentially, by fluorescent in situ hybridization (FISH).” No endogenous CRE activity recording was demonstrated. No natural recording tape was implemented. No FISH-based readout was demonstrated (their recording sequence is only 5 or 6 nt long).

We apologize for the confusing terminology. By endogenous CREs, we were referring to the naturally occurring sequences (not locations) of CREs which are then tested “out of context” with either MPRA or ENGRAM. In using the terms “endogenous CRE” vs. “synthetic CRE”, we were attempting to distinguish between naturally occurring sequences vs. designed sequences (e.g. spacer-separated repeats of TF binding sites). We did in fact test natural CRE sequences in the MPRA vs. ENGRAM experiments -- albeit, out of the context of their native locus. To the next point, in saying “natural or synthetic recording site”, we were referring to either the endogenous *HEK3* locus (“natural recording site”) vs. a copy of the *HEK3* target site, introduced transiently or stably (“synthetic recording site”).

In retrospect, we recognize how both sets of terms could be easily misunderstood, and in the revised version of **Figure 1a**, reproduced below, we now refer to “naturally occurring” and “designed” CRE sequences (both of which are successfully used upstream of ENGRAM recorders in the manuscript) and explicitly define what we mean by these terms in the legend. We also refer to the DNA Tape being located at an endogenous genomic locus (‘endogenous DNA Tape’) or exogenously introduced via piggyBac (‘synthetic DNA Tape’) in the legend, as both kinds of DNA Tape are used in the manuscript.

Finally, the reviewer is correct that no FISH-based readout was shown. Rather, we were trying to highlight the potential. In retrospect, we agree that this is not appropriate, particularly for an introductory figure, as we did not actually demonstrate it. Thus, we have also deleted any reference to FISH from **Figure 1a**, although we still briefly allude to it in the Discussion as a future possibility.

Figure 1. ENhancer-driven Genomic Recording of transcriptional Activity in Multiplex (ENGRAM). (a) Schematic of ENGRAM. Naturally occurring or designed cis-regulatory elements (CREs) drive activity-dependent transcription of a prime editing guide RNA (pegRNA) encoding a CRE-specific insertion. The CRE could be a natural enhancer sequence (notably tested out of context, as in a conventional or massively parallel reporter assay) or a designed sequence (e.g. spacer-separated repeats of a transcription factor motif). The pegRNA is embedded within a Pol-2 transcript, flanked by two 17bp *csy4* hairpin such that it can be released by Csy4 ribonuclease. The liberated pegRNAs are designed to write a CRE-specific insertional barcode to a common DNA Tape site, which could be an endogenous genomic locus (e.g. the native *HEK3* site) or a synthetic DNA Tape (e.g. exogenously introduced via lentiviral or piggyBac integration). Sequencing of the DNA Tape reveals which CREs were active; if there are multiple ENGRAM reporters in the system, their relative activities can be measured across a pool of cells and/or multiple DNA Tape sites in each cell.

7. Although I liked Figures 1 and 2 and Supplementary Figures 1 and 2 and appreciated their careful experimental design and analyses, I was confused about the part where they attempted to reduce the background activity of ENGRAM. In Figures 2b-d, they analyzed the background activities of 5' Recorder, 3' Recorder, and 3'FT Recorder and stated that 5' and 3'-FT recorders were optimal. However, they did not measure their enhancer recording activities side by side. They could be just overall diminished.

We thank the reviewer for this insightful comment. To address this, we performed a new experiment with the three ENGRAM designs, side-by-side exactly as suggested. Specifically, we cloned the same NFκB response element upstream of minP in the three versions of ENGRAM 2.0, and then measured both recording efficiency upon TNFα stimulation, relative to background activity. As shown in the figure panels reproduced below (**Figure 1b-d** of the revised manuscript), we observe strong signal-to-noise only with the 5' ENGRAM and 3'-FT ENGRAM recorders, with these recorders exhibiting both higher absolute recording efficiency and lower background activity than the 3' ENGRAM recorder, and consistently so across three replicates per design. In particular, while the 3' ENGRAM construct exhibited only a 1.4-fold signal-to-noise ratio, the 5' ENGRAM and 3'-FT ENGRAM recorders exhibited 13.3-fold and 23.8-fold signal-to-noise ratios, respectively.

Figure 1. ENhancer-driven Genomic Recording of transcriptional Activity in Multiplex (ENGRAM). (b) Three distinct ENGRAM 2.0 architectures with the *csy4* hairpin-flanked *pegRNA* embedded in the 3' UTR (top) or 5' UTR (middle) of a transcript encoding *Csy4*, or 3' ENGRAM 2.0 with an additional *csy4* hairpin in the 5' UTR (bottom). The presumed cleavage of the *csy4* hairpin at the 5' UTR in both the 5' ENGRAM and 3'-FT ENGRAM architectures may impose an auto-regulatory negative feedback on *Csy4* levels. (c) All three ENGRAM 2.0 recorders were integrated via piggyBac into PE2-expressing cells in triplicate, each driving 1,024 potential barcodes (5N degenerate insertion) with minP. The background editing efficiency was periodically checked over 20 days. Error bars correspond to standard deviations across 3 transfection replicates. (d) An NF- κ B response element was cloned upstream of minP in each of the three ENGRAM 2.0 designs. These ENGRAM recorders were separately integrated into the genomes of PE2(+) HEK293T cells via piggyBac. We then measured recording at the endogenous *HEK3* locus in the absence vs. presence of 10ng/ml of TNF α in triplicate. Error bars correspond to standard deviations across 3 replicates. P-values were obtained using the two-tailed Student's t-test.

8. Although the authors hypothesized that these circuits make auto-regulatory negative feedbacks, another possibility is that the lower expression of *Cys4* by the enhancer-minP just reduces functional *pegRNA* molecules.

We thank the reviewer for suggesting this alternative explanation. Compared to the architecture of ENGRAM 1.0 in which the *Csy4* was constitutively expressed under PGK promoter, we anticipate that *Csy4* levels in the absence of signal should indeed be lower with ENGRAM 2.0. To address this comment, together with this reviewer's previous comment, we performed a new experiment in which three ENGRAM 2.0 designs were compared side-by-side in a single experiment, all driven by the same NF κ B response element upstream of minP. As shown in response to the comment immediately above, the two architectures subject to potential auto-regulatory negative feedback on *Csy4* levels (dashed lines in schematic shown in **Figure 1b** above) exhibited much lower background than the design in which this feedback is not predicted (**Figure 1c-d** above). Although there may be other explanations, this result suggests that their relative transcription levels are not the explanation, as all three designs bear identical upstream architecture in this side-by-side experiment.

9. The overall recording activity was only a few percent. The authors use the measurement unit of % of insertions (Figure 2f) or insertion proportion (log2). For me, this looked like an attempt to stay readership attentions from the low recording activity. They should be %inserted read and insertion proportion (log10). What is the absolute recording efficiency of each insert in Figure 1d?

We see the reviewer's point (similarly raised by Reviewer #2) and apologize. To improve transparency, we now only present absolute editing efficiency. The referenced main text figure (Figure 2f) has been moved to the supplement and is now **Supplementary Figure 3b**, reproduced below. We also highlight that with recent development of PEmax, recording efficiency improved 1.7-fold or 9.3-fold over PE2 in K562s or mESCs,

respectively, as shown in **Supplementary Figure 1h**, also reproduced below. Of note, in mESCs that are stably expressing PEmax and writing to a synthetic DNA Tape (synthetic *HEK3* site integrated via piggyBAC), recording levels are on the order of 10-15% rather than a few %, three days after transient transfection with an ENGRAM recorder.

Supplementary Figure 3. Benchmarking of ENGRAM against reporter assays. (b) Barcodes corresponding to the highly active CRE were 15.1-fold, 23.6-fold, and 41.3-fold more abundant than barcodes corresponding to lowly active CRE, minP-only or promoter-less controls, respectively. Error bars correspond to standard deviations across 3 transfection replicates. P-values were obtained using the two-tailed Student's t-test.

Supplementary Figure 1. Optimization of ENGRAM architecture. (h) Comparison of recording efficiency between prime editor variants. PE2 or PEmax, together with a library of constitutive ENGRAM recorders programming a degenerate 5N insertion to the *HEK3* locus, were co-transfected into K562 cells. In a separate experiment, an ENGRAM recorder encoding a 5-bp insertion was transiently transfected into mESCs bearing a synthetic DNA Tape (*HEK3*) and stably expressing PE2 or PEmax. Genomic DNA was harvested 3 days post-transfection and insertions at endogenous (K562) or synthetic (mESC) *HEK3* loci were quantified. We observed that PEmax drove 1.7-fold and 9.3-fold more efficient recording than PE2 in K562 cells and mESCs, respectively. Although we used PE2 for many of the experiments reported in this paper (as PEmax had yet to be described at the time that they were performed), we recommend using PEmax for all future recording assays. Error bars correspond to standard deviations across 3 transfection replicates. P-values were obtained using the two-tailed Student's t-test.

10. “The intent of this non-targeting pegRNA was to compete for PE2 binding and thereby minimize any “pre-integration” activity from pegRNAs encoded by the ENGRAM recorders.” How was the integration was confirmed? No transposon control? Can they employ lentiviral integration?

We apologize for not being clearer on this detail. Of note, the non-targeting pegRNA results have been removed during revisions, as we identified conditions under which background could be reduced without it (see our response to Comment #20 from this reviewer). However, to answer the question about integration, we did not use selection, but rather no-transposase controls to define the amount of time required for unintegrated plasmid to be diluted out. Specifically, a piggyBac-GFP vector was transfected with vs. without transposase. GFP DNA abundance was measured using qPCR with two pairs of GFP-specific primers (together with a pair of primers directed at native RPPH1 locus as an internal control) over the course of 15 days. The estimated levels of GFP abundance in the no-transposase control decreased to noise levels after 7-9 days, while GFP abundance with transposase present indicated that we were achieving an MOI on the order of 15-20. This experiment is now included in the manuscript as **Supplementary Figure 6d-e**, reproduced below. Similar transfection conditions (including with respect to the size of the vector being transfected), together with a delay of at least 10 days after transfection, were used for all subsequent experiments involving integration.

Supplementary Figure 6. Multi-channel recording of the intensity and duration of signaling pathway activity. (d) Estimating the multiplicity of integration (MOI) of piggyBac transposase integration with qPCR. Cells were transfected with a piggyBac-GFP construct, either with or without piggyBac transposase. GFP DNA abundance was measured using qPCR with two pairs of GFP-specific primers (together with a pair of primers directed at native RPPH1 locus as an internal control) over the course of 15 days. The estimated levels of GFP abundance in the no-transposase control decreased to background levels after 7-9 days. DNA-level GFP abundance with transposase present, at timepoints where controls have gone to background, suggests that we were achieving an MOI on the order of 15-20. **(e)** The estimated proportion of cells with at least one copy of each of the three recorders as a function of MOI, assuming a Poisson distribution. At an MOI of 15, over 98% of cells are predicted to bear at least one copy of each of the three recorders.

To the last point, we agree we could have used lentivirus to deliver these constructs. However, we choose piggyBac because we find it to be more efficient, more straightforward to use, and less prone to silencing than lentivirus (particularly in differentiating mESCs, used in the new gastruloid experiments). It would also have introduced other issues, e.g. for the ~300-plex CRE ENGRAM experiment, lentiviral packaging would have risked scrambling CRE and barcode associations due to the template-switching phenomenon²². For such reasons, across all projects, our lab has almost entirely switched away from lentivirus and over to piggyBac.

11. Maybe the enhancer recording library can be transfected into different cell lines like K562, HEK, and fibroblast cells to see if ENGRAM can capture cell line-specific enhancer activities.

This is a terrific idea. In addressing this, rather than the 300 CRE library from the original manuscript, we newly designed and synthesized a library of ENGRAM recorders driven by 98 synthetic *cis*-regulatory elements (synCREs), each bearing homotypic repeats of a single TF motif. In brief, these synCREs were designed by clustering the 841 vertebrate motifs in JASPAR database by similarity and then manually curating the resulting motifs to a set of 98 (6-20 bp), each representing a single TF or a TF family (**Supplementary Table 3**). Each designed synCRE in this set is composed of 6 repeats of one of the 98 motifs, embedded within an inactive DNA sequence and separated by 4-bp spacers. The 98 synCREs were then cloned into the ENGRAM construct, each driving a CRE-specific 5-bp insertion to be written to the DNA Tape (*HEK3*).

In a first experiment, directly inspired by this comment, the synCRE ENGRAM recorders were transiently transfected to PE2(+) HEK293T or K562 cells in triplicate, and the cells harvested 2 days post-transfection, followed by amplification and sequencing of endogenous DNA Tape (*HEK3*) (**Figure 2f**, reproduced below). In a second experiment, described later in the paper in the context of setting up for the gastruloid experiments described above in response to Comment #5 from this reviewer, we also stably introduced these same synCRE ENGRAM recorders into mouse embryonic stem cells (mESCs) bearing both dox-inducible PEmax and synthetic DNA Tapes (*HEK3*) (**Figure 5a**, reproduced below). Recording was controlled by adding dox to the medium and cells were harvested after 1 day of recording, followed by amplification and sequencing of the synthetic DNA Tapes.

In all three cell lines (which also vary with respect to transient vs. stable introduction of the recorders, PE2 vs dox-inducible PEmax, and endogenous vs. synthetic DNA Tape), the proportions at which various synCREs drove recording to DNA Tape were highly reproducible (**Supplementary Figure 5b-c**; **Supplementary Figure 8a**, both reproduced below), albeit somewhat less so in K562 cells probably due to lower transfection efficiency.

Of the 98 synthetic CRE ENGRAM recorders, 15 (HEK293T v.s. K562), 28 (mESC v.s. K562), and 18 (mESC v.s. HEK293T) CREs exhibited significant and substantial differences in recorded activity (Wald-test with Benjamini-Hochberg correction $P < 0.001$ for a fold-difference > 2 ; **Figure 2g**, **Figure 5b-c**, all reproduced below). Most of the differential recording patterns in K562 and HEK293T cells were concordant with the differential expression of at least one TF that putatively binds the corresponding TF motif (**Supplementary Figure 5d**, reproduced below).

In our view, these experiments clearly demonstrate the ability of a modestly sized (and generic) library of designer ENGRAM recorders to record cell type-specific patterns of CRE activity to DNA Tape.

Figure 2. Multiplex recording of natural or synthetic CRE activity with ENGRAM. (f) Schematic of experiment directed at recording the cell type-specific activities of 98 synthetic CREs. Each synthetic CRE is composed of 6 repeats of known transcription factor (TF) binding motif (6-20 bp) with 4-bp spacers. These synthetic CREs were cloned to the ENGRAM construct such that each was associated with a unique 5-bp barcode designed to be written to endogenous DNA Tape by Csy4-released pegRNA. This construct library was transiently transfected to PE2(+) K562 or HEK293T cells in triplicate. Genomic DNA was harvested 48 hours later, followed by PCR and sequencing. (g) Volcano plot of differential recorded activity of synthetic CREs in K562 vs. HEK293T cells. Each plotted point represents a synthetic CRE, labeled by the single TF or a representative of the TF family associated with the corresponding motif (**Supplementary Table 3**). Red labeled points are 15 ENGRAM recorders exhibiting significant and substantial differences (Wald-test with Benjamini-Hochberg correction $P < 0.001$ for a fold-difference > 2). Many of these differences can potentially be explained by the differential expression of the corresponding TFs (**Supplementary Figure 5d**).

Supplementary Figure 5. Multiplex recording of cell-type-specific activities of synthetic CREs with ENGRAM. (a) Recording efficiency of synthetic CREs at the endogenous DNA Tape site (*HEK3* locus) in HEK293T (12.6%) and K562 (1.0%) cells. The difference in overall recording between cell lines is likely attributable to differences in

transfection efficiency. **(b-c)** Log-scaled insertion proportions for 5-mer barcodes linked to the 98 synthetic CREs were highly reproducible across transfection replicates for both HEK293T **(b)** and K562 **(c)** cells. Each value corresponds to the proportion of barcodes read out at the DNA level from the endogenous *HEK3* locus. As the same number of cells were sampled for recording, the lower reproducibility in K562 cells is likely secondary to lower transfection/editing efficiency. **(d)** Differential expression of TFs in HEK293T and K562 cells. The differential recording activity of most synthetic CREs **(Figure 2g)** can be explained by the differential expression of the corresponding TFs in HEK293T vs. K562 cells.

Figure 5. Multiplex recording of CRE activity in embryonic stem cells and differentiating gastruloids. **(a)** Schematic of polyclonal ENGRAM mESCs, with each cell bearing multiple copies of doxycycline-inducible PEmax, ENGRAM recorders, and synthetic DNA Tape (*HEK3*). **(b-c)** Volcano plot of differential recorded activity of ENGRAM recorders in cultured mESCs vs. K562 **(b)** or cultured mESC vs. HEK293T **(c)** cells. Each plotted point represents a distinct ENGRAM recorder driven by a synthetic CRE composed of a homotypic array of the consensus DNA binding motif for a TF or TF family (**Supplementary Table 3**). Red, labeled points are ENGRAM recorders exhibiting significant and substantial increasing activity (Wald-test with Benjamini-Hochberg correction $P < 0.001$ for a fold-difference > 2).

Supplementary Figure 8. Multiplex recording of CRE activity in embryonic stem cells and differentiating gastruloids. **(a)** Log-scaled insertion proportions for 5-mer barcodes linked to the 98 synthetic CRE-driven ENGRAM recorders were highly reproducible across biological replicates for cultured mESC, as read out by amplification and sequencing of synthetic DNA Tape.

12. Figure 3 needs more supportive evidence with proper controls. From Figure 3h, they stated the editing efficiency was proportional to the duration of ligand exposure. Did they establish the cells first all with the expected circuits? Is there a possibility that the cells with the expected edits propagate better as they tend to have all the expected plasmids with necessary selection markers? The trend could be seen just by diminishing of cells that do not have all of the plasmids. The proper control experiments would be done by giving different exposures to different samples but after removal signals at different time points, all cell culture samples should be harvested at the same time (unifying of total cell culture/selection time).

We apologize for our lack of clarity on this point. For the experiments shown in the original Figure 3, all recorders/circuits were integrated into a single cell derived line of HEK293T that had already been engineered to constitutively express PE2. Cells were in culture for at least 10 days before the experiment to dilute out unintegrated plasmids, with the timeframe for that dilution (a minimum of 10 days) set by the experiments shown in our response to this reviewer's Comment #10 (see above). No selection was applied in any experiment in either the original or revised version of the manuscript. Because there is no selection, we do not anticipate any growth bias for edited or 'all component containing' cells in these or other experiments. We have updated the corresponding description of the experiment and **Methods** to be clearer on these points, and the experiment demonstrating the dilution timeframe and estimated MOI under these conditions is now included as **Supplementary Figure 7d-e** (reproduced in response to this reviewer's Comment #10 above).

13. It is unclear if the endogenous activity falls within the dynamic range in which the ENGRAM was shown to be functional or if the dynamics of endogenous activity fall within the time frame tested in Figure 3.

This point touches on both dynamic range and time-frame of endogenous signaling activity.

Regarding dynamic range, for context, we relied on signal-specific response elements that have classically been via reporter assays to measure endogenous signaling activities in response to agonist concentrations in similar ranges to what we use here. For Wnt signaling, 3 μM of CHIR99021 is used to induce gastruloid formation²³, which falls into 1-4 μM tested by us. For NF κB signaling, we used TNF α as an agonist. Although the sensitivity of this pathway to this agonist varies across cell types, it has been shown in reporter assays that the NF κB response element has an EC_{50} of 3.34 ng/ml²⁴ for TNF α in HEK293T cells, which falls within the 0.1-16 ng/ml range over which we show ENGRAM is sensitive (and indeed, the "half maximal" recording rate for TNF α was 2.5 ng/ml, which is close to its reported EC_{50}). As such, we believe that the sensitivity, at least of these two particular recorders, is limited by the signal-specific transcriptional response elements rather than the ENGRAM architecture. That we are operating in the concentration ranges in which these canonical pathway-specific agonists are typically used^{20,21,23,24} is now mentioned in the revised figure legend.

Regarding time-frame, ENGRAM is similarly constrained by the fact that it leverages enhancer-like response elements to drive transcription. These particular pathways (Wnt, NF κB) do act via effects on transcriptional programs (indeed, the basis for the fact that there are well-defined, pathway-responsive CREs), and this presumably shapes the timeframe that is relevant to their activities. Here we demonstrate sensitivity over a range of 6 to 48 hours of agonist exposure, which is on par with the stimulation time-frames in which these same response elements have been used in reporter assays (typically, ~ 24 hours)²⁵⁻²⁸.

Of course, there are aspects of these or other signaling pathways that do not operate by modulating transcription, and we are currently blind to those mechanisms in the same way that classic reporter assays or

MPRAs that rely on these same CREs are also blind to them. However, we can at least imagine alternative ways in which pegRNA activity might be made biologically conditional, in order to record signaling or biological phenomena without relying on new transcription, e.g. we recently posted a preprint²⁹ on a molecular proximity sensor based on an engineered, dual-component guide RNA, which could in principle be coupled to ENGRAM.

We now raise these points explicitly in highlighting key limitations of ENGRAM in the revised Discussion:

“First, like MPRAs, ENGRAM relies on CRE-mediated enhancement of Pol-2 transcription. As such, it is not well-suited to biological signals or states that are not readily coupled to CREs, nor to recording at fast timescales. In the future, these challenges might be addressed in part with heterologous signal conversion (e.g. Tet-On) or with entirely different strategies for biologically conditional prime editing²⁹.”

14. The authors integrated ENGRAM pegRNA recorders for TET-On, NFkB, and Wnt using PiggyBac transposition in a population of HEK293T PE2 expression cells that had previously been derived by single-cell isolation. The authors did not do another round of single-cell isolation and select for cells that have all three co-integrated recorders but rather performed the experiment on a bulk population of integrated cells. As such, based on the current experimental design, it is difficult to say that each recorder is “competing to write to a shared DNA tape.” The authors could instead express all three ENGRAM pegRNAs from a single cassette.

We see the reviewer’s point that we did not perform an additional round of single cell isolation & selection, but as we consistently used conditions that gave rise to an estimated MOI of 15-20 integrations per cell, we expect that the ENGRAM recorders are competing in the vast majority of the cells. Specifically, assuming a Poisson distribution of integrations, we expect that >98% of cells to have at least one copy of each of the three recorders at an MOI of 15.

The corresponding main text sentence has been updated as follows. **Supplementary Figure 6d-e** is reproduced above in response to Comment #10 from this reviewer.

“Although we did not construct a monoclonal cell line or explicitly confirm that individual cells contain integrated copies of all three recorders, the conditions used here are estimated to yield 15-20 integrations per cell, such that the vast majority of cells should contain at least one copy of each of the three recorders (**Supplementary Figure 6d-e**)”

We also note that although we do not “prove” that multiple ENGRAM recorders are competing to write to the same tape in these specific experiments, we do note that this is unequivocally the case in the new ENGRAM + DNA Typewriter experiments, as there we are leveraging the 5’ to 3’ order of which signal-specific insertion precedes the other signal-specific insertion to decode signal order (see **Figure 4**, reproduced above in response to Comment #1 from this reviewer).

15. Figure 4b is great but the following demonstration is not a recording of event artificial signals as they just tested different orders of pegRNA operons. The functionality of the cells with the pre-installed circuit should be tested. Furthermore, although they start this section by stating “in context of a multiplex signal recorder”, this section does not demonstrate the multiplexability of their technology that is stressed in the manuscript. Although the initial recording tape is not event-dependent, after the first edit, the preceding pegRNA that can edit the tape becomes unique for each prior event. Hence, even if it is only for 2 events, it is difficult to see how

the system described in this section could be more multiplexable/advantageous as compared to CAMERA or DOMINO.

These are great points, and this comment pushed us to pursue the integration of ENGRAM and DNA Typewriter, as described in our response to Comment #1 from this reviewer. We believe that this combined system has massive advantages over CAMERA and DOMINO, as well as over our previously described circuits for ordered recording, as with ENGRAM + DNA Typewriter, including: (1) All possible signal orders are recorded to a common “media”, DTT; (2) Although the number of possible signal orders grows combinatorially as more signals/recorders are added to the system, the same DTT can be used to capture all potential orders; (3) Each new signal to be recorded only requires one more ENGRAM recorder to be added to the system, rather than multiple gRNAs or pegRNAs encoding various dependencies; and (4) As the signal-specific portion of the pegRNA-mediated insertion can be 5-6 bp, hundreds or even thousands of signals can potentially be recorded, *i.e.* that aspect of the system is not rate-limiting.

We believe that the combination of ENGRAM and DNA Typewriter provides a clear path forward for the time-resolved, multi-channel recording of biologically conditional signals, potentially in combination with cell lineage, with markedly more long-term potential than alternative DNA-based recording systems.

16. Supplementary Figure 4a was interpreted as a result of crosstalks, but for me, this is a too strong assumption. They could be derived just by background activities.

The reviewer may have misunderstood our language, as we believe that we are in full agreement. We interpreted this data as “*exhibit minimal crosstalk between signaling pathways*”, meaning there is practically no (indeed, if any) cross-talk, as the levels observed are similar to background in combinations where a corresponding agonist was not present. We have revised the relevant sentence in the figure legend to read as follows, in hopes of being clearer:

“Of note, the recorders did not exhibit any discernible crosstalk, suggesting that the underlying signaling pathways are truly orthogonal (*e.g.* stimulating with CHIR does not lead to appreciable recording by the NF-κB recorder).”

17. Recording the order of CREs would be more impressive.

Per the reviewer’s suggestion, we performed new experiments along these lines, combining ENGRAM and DNA Typewriter in order to decode complex patterns of biological signals as opposed to simple orders of serial transfections. Please see **Figure 4, Supplementary Figure 7** and corresponding additions to text, as well as our response to their Comment #1 above.

Minor comments:

18. “We speculated that this was due in part to the accumulation of background edits due to constitutive expression of Csy4, which may also have reduced growth of the monoclonal cell line due to cytotoxic effects” It would have been better if they tested other published means of expressing pegRNA from pol2 promoter such as through utilizing tRNA processing (Knapp et al 2019 Nature Communications). Or Ribozyme? Also, it would benefit the community how their system has an effect on cell viability and endogenous gene expression.

To address the first part of this comment (*i.e.* alternatives to Csy4), we chose to focus on tRNAs rather than ribozymes, because Knapp et al. *Nature Communications* (2019) showed that ribozymes were inefficient for this purpose (see Figure 5c of that paper). To evaluate tRNAs in this context, we replaced the Csy4 hairpins flanking the pegRNA with tRNA hairpins, in the context of a NF- κ B response element-driven 5' ENGRAM construct. However, the tRNA flanked recorders failed to show recording activity in the presence of 10ng/ml TNF α , as illustrated by new **Supplementary Figure 1i**, reproduced below.

To address the other aspect of the second part of this comment (the question about the effects of ENGRAM on endogenous gene expression), we performed bulk RNA-seq on HEK293T cells, either with no integrated modifications, with integrated PE2, or with integrated PE2 + the ENGRAM NF κ B recorder, harvesting both without or after TNF α stimulation. About half (52%; PC1) of the transcriptional variation is associated with the derivation of a PE2 line from a single cell (*vs.* the 293T parental strain), and then about 34% contributed by TNF α treatment. However, the transcriptional profiles of cells bearing PE2 and exposed to TNF α are robust to the presence/absence of the ENGRAM recorder (*i.e.* no significant changes, nor any changes of any gene over 50%). In other words, the presence of the ENGRAM recorder, even when activated by TNF α , does not appear to substantially impact the transcriptome above and beyond changes that are associated with the other variables (*i.e.* single cell derived PE2 line; TNF α exposure). These data are now included as **Supplementary Figure 1j-k**, also reproduced below.

Supplementary Figure 1. Optimization of ENGRAM architecture. (i) tRNA processing for pegRNA release is not compatible with ENGRAM. We replaced *csy4* hairpin with tRNA sequences as an alternative approach for pegRNA release. Pol-2 transcripts bearing pegRNAs flanked by either *csy4* hairpins or tRNA sequences, in both cases encoding degenerate 5N insertions, were driven by the NF κ B response element. Recorders were integrated to the genome via piggyBac, and their activities were measured in the absence or presence of 10ng/ml TNF α in triplicate. In contrast with *csy4* hairpin-flanked pegRNAs, tRNA-flanked pegRNAs failed to exhibit recording activity in the presence of TNF α . **(j-k)** Measuring the effects of ENGRAM components on cell functions using bulk RNA sequencing. We profiled bulk RNA expression in HEK293T cells, either with no integrated modifications (293T), with integrated PE2 (PE2, monoclonal line), or with integrated PE2 and ENGRAM NF κ B recorder (PE2.recorder, derived from PE2 line mentioned above), harvesting either without *vs.* after TNF α stimulation. **(j)** PCA plot of bulk RNA-seq data from various conditions. About half (52%) of the observed transcriptional variation was associated with derivation of the PE2 monoclonal line, and an additional 34% associated with TNF α treatment. **(k)** Volcano plot showing differentially expressed genes between the PE2+TNF α and PE.recorder+TNF α conditions. No significant changes in gene expression were detected (Wald-test with Benjamini-Hochberg correction $P < 0.05$ for a change $> 50\%$ (\log_2 -fold change > 0.58)).

Finally, another experiment that is relevant to the second part of this comment, and in particular the part about the impact of ENGRAM on cell viability, are the new gastruloid experiments, performed in response to Comment #5 from this reviewer. The fact that the presence of ENGRAM recorders at a high MOI ($n = 10-15$) did not interfere with gastruloid induction or morphology suggests minimal effects on cell viability (see **Supplementary Figure 8b**, reproduced below).

Supplementary Figure 8. Multiplex recording of CRE activity in embryonic stem cells and differentiating gastruloids. (b) Representative images of gastruloids induced from polyclonal ENGRAM mESCs, illustrating that the components of the ENGRAM recording system do not substantially impact the morphological development of gastruloids. Scale bar: 100 μm .

19. PE3 in the original PE paper or epegRNA efficiency is better than PE2. Is there any reason they used PE2?

We agree that using PE3 could potentially improve the recording efficiency. However, PE3 is based on an additional nicking gRNA that targets downstream of the primary target, which makes design and cloning substantially more challenging. Furthermore, PE3 is more prone to imprecise edits, e.g. PE3 in human ESCs has been reported to induce frequent deletion of sequences around two nick-sites³⁰, which is precisely one of the key problems with GESTALT and earlier generations of CRISPR-based molecular recorders that we were trying to avoid with the switch to prime editing. In a nutshell, this is why we went with PE2 rather than PE3.

However, inspired by this suggestion and literature/materials that became available after we had submitted the paper, we tested the newly optimized PEmax construct in the context of ENGRAM in both K562 cells and mESCs, and observed a reproducible 1.7-fold and 9.3-fold increase in editing efficiency with PEmax over PE2, respectively (see **Supplementary Figure 1h**, reproduced below). Although we did not repeat all experiments that had already been done with PEmax, we now explicitly recommend in the figure legend using PEmax going forward (and have done this ourselves). We anticipate that further improvements to prime editing by the community (e.g. the recent Doman *et al.* paper from the Liu group on evolved prime editors³¹) will further boost recording efficiencies achievable with ENGRAM.

Regarding epegRNAs, we performed a new experiment in which we compared the editing efficiency of pegRNAs vs. epegRNAs encoding a library of 5N insertions and driven by the PGK promoter. Although in other

experiments in our lab we observe higher efficiencies for epegRNAs as reported in the literature, here we observed, surprisingly, ~30% lower efficiency with epegRNAs in the context of ENGRAM. We speculate that this is possibly because the Csy4 hairpin plays a similar role to the tevoPreQ1 hairpin used in epegRNAs. We now include this data as **Supplementary Figure 1g**, also reproduced below. In this legend, we explicitly recommend using PEmax but not the epegRNA structure for ENGRAM-based recording.

Supplementary Figure 1. Optimization of ENGRAM architecture. (g) Comparison of recording efficiency between epegRNAs vs. pegRNAs. Libraries of epegRNAs or pegRNAs (constitutively active and programming a degenerate 5N insertion to the *HEK3* locus) were cloned into the 5' ENGRAM 2.0 architecture. These two libraries were transiently transfected, separately, into PE2+ HEK293T cells in triplicate. Genomic DNA was harvested 3 days post-transfection. Unexpectedly, pegRNAs showed 30% higher recording efficiency than epegRNAs. We speculate that in ENGRAM, the *csy4* hairpin might already serve to protect pegRNAs from degradation, and the double hairpin at the end of pegRNA might affect the RNA folding, reducing efficiency. Error bars correspond to standard deviations across 3 transfection replicates. P-values were obtained using the two-tailed Student's t-test. **(h)** Comparison of recording efficiency between prime editor variants. PE2 or PEmax, together with a library of constitutive ENGRAM recorders programming a degenerate 5N insertion to the *HEK3* locus, were co-transfected into K562 cells. In a separate experiment, an ENGRAM recorder encoding a 5-bp insertion was transiently transfected into mESCs bearing a synthetic DNA Tape (*HEK3*) and stably expressing PE2 or PEmax. Genomic DNA was harvested 3 days post-transfection and insertions at endogenous (K562) or synthetic (mESC) *HEK3* loci were quantified. We observed that PEmax drove 1.7-fold and 9.3-fold more efficient recording than PE2 in K562 cells and mESCs, respectively. Although we used PE2 for many of the experiments reported in this paper (as PEmax had yet to be described at the time that they were performed), we recommend using PEmax for all future recording assays. Error bars correspond to standard deviations across 3 transfection replicates. P-values were obtained using the two-tailed Student's t-test.

20. Figure 3e needs color legends

We thank the reviewer for this suggestion. Of note, in the original submission and Figure 3e in particular, we assessed the signal-to-noise of various ENGRAM recorders, but also tested the idea of using a non-targeting pegRNA to minimize recording from the plasmid before integration. We later identified conditions under which conventional transfection achieves similarly high signal-to-noise. Hence, we dropped our presentation of the blocking strategy and now just use conventional transfection with no blocking pegRNA. The relevant data is still presented in **Figure 3e**, reproduced below with labels added.

Figure 3 (original panel provided at left for reference) Multi-channel recording of the intensity and duration of signaling pathway activity. (e) Signal-to-noise. The Tet-On, NFκB and Wnt ENGRAM recorders respectively exhibit a 11.5-fold, 19.0-fold and 22.6-fold difference in editing levels with vs. without the maximum dose of the corresponding agonist. Error bars correspond to standard deviations across 3 replicates.

21. Figure 2g: “1 barcode each”? or did you mean “2 barcodes each”?

We apologize for any confusion. For the experiment in which we performed multiplex ENGRAM recording with ~300 CREs, we used only 1 barcode per CRE. Of note, we added a new table (**Supplementary Table 6**) to make it clearer how insertional barcodes were picked for each experiment in the paper.

22. Supplementary Figure 1a: Typo in the y-axis

We thank the reviewer for pointing this out. The old Supplementary Figure 1a (reproduced below, left panel) has been updated to **Supplementary Figure 1c** (reproduced below, right panel). The typo has been fixed.

Note that we no longer present the data for PGK-5N as we re-performed the 5N measurement with the ENGRAM 2.0 architecture in the context of a comparison of pegRNAs vs. epegRNAs (see **Supplementary Figure 1g**, reproduced above in response to Comment #19 from this reviewer).

Supplementary Figure 1c (original panel provided at left for reference). Optimization of ENGRAM architecture. (c) Across three transfection replicates, the ENGRAM 1.0 recorder driven by a constitutive Pol-2 PGK promoter (PGK-CTT) exhibited comparable efficiency for inserting CTT at the *HEK3* locus to a U6-driven CTT-pegRNA (U6-CTT). In the K562 cell line in which this experiment was performed, PE2 and Csy4 were constitutively expressed.

23. Supplementary Figure 1e: The x-axis needs to be properly labeled.

We thank the reviewer for this suggestion. Supplementary Figure 1e of the submitted manuscript is now **Supplementary Figure 2f**. The x-axis is now labeled as “Dinucleotide features”.

Supplementary Figure 2. ENGRAM installs insertional barcodes with reproducible, predictable efficiencies. (f) The rank-ordered coefficients of the linear lasso regression. Positional information of single nucleotides and dinucleotides and minimum free energy (MFE) of secondary structure were used as input features for training. In addition to MFE, which received the highest coefficient, the top 4 and bottom 4 coefficients for sequence features are annotated (e.g. 3-TC means TC dinucleotide starting at position 3).

24. The plasmid constructs should be deposited to the Addgene.

This was in fact done at the time of submission but it is possible that the relevant text was missed by the reviewer. The **Data availability** paragraph states:

“Raw sequencing data have been uploaded on Sequencing Read Archive (SRA) with associated BioProject ID PRJNA780310. Plasmids for ENGRAM 2.0 recorders (piggyBac-5'-ENGRAM and piggyBac-3'-FT-ENGRAM) have been deposited to Addgene (ID 179157 and ID 179158).”

25. The sequencing data should be deposited to the NCBI SRA and provided with accession number.

Please see response to previous comment.

Referee #2 (Remarks to the Author):

1. The submitted manuscript surely wins the 2021 award for creative toolbuilding in putting together new modalities of targeted genome engineering, new enzymes that target RNA, and next-generation sequencing to advance the rapidly growing field of using the cell's own genome to record a biological phenomenon – in this case, gene transcription in mammalian cells. As the authors point out, this is not a new idea, and many elegant studies have been done before Cas-driven editing was invented (in addition to the mouse literature the authors cite they may consider some beautiful examples from the yeast literature, eg <https://pubmed.ncbi.nlm.nih.gov/19543267/>).

We thank the reviewer for appreciating the integration of these tools to create ENGRAM. There is nothing new under the sun, and as with so much else in our field, many of the broad concepts trace back to yeast. However, we are not sure if the paper cited (a Rine Lab paper about the establishment of gene silencing) is the one that the reviewer intended, and we are wondering if the reviewer is instead referring to examples of using site-specific recombinases in yeast (or *Drosophila*) that preceded their application in mouse? Please let us know if we have it wrong, but regardless we agree that the pre-mouse work is terrific (and was critical) and should be cited. We have added references to these papers in the context of noting SSR-based recording:

<https://pubmed.ncbi.nlm.nih.gov/2509077/>

<https://www.ncbi.nlm.nih.gov/pmc/articles/PMC365329/>

If we got it wrong, please let us know!

2. The submitted manuscript describes in sometimes excessive – for a journal with a broad audience such as “Nature” – detail how this tool, thoughtfully named ENGRAM (see comment below), was optimized, and then put it to its key use: recording of multiple transcriptional outputs in multiplex. These data are shown in Fig 4 and lend a general sense of enthusiasm for the broad interest of this approach. In equally broad strokes, we remain woefully ignorant of how human gene regulation actually works (despite having enormous catalogs of component parts), and a tool such as ENGRAM could be a useful and powerful addition to our arsenal.

We agree that there is perhaps too much detail included on the optimization of ENGRAM for a broad audience. To address this, we have: (a) Greatly shortened the description of ENGRAM 1.0 in the main text, and moved all of the corresponding figure panels (as well as the schematic comparison of ENGRAM 1.0 vs. 2.0) to **Supplementary Figure 1**. (b) Consolidated other material from Figures 1 & 2 from the original paper into a single figure (new **Figure 1**). (c) Taken a pass at more generally tightening the Introduction and early sections of the Results. We welcome any further suggestions around further focusing or sharpening our presentation and avoiding unnecessary detail.

3. There are two gaps. The first – it's unclear how good of a ranking tool ENGRAM actually is? – can be addressed by reanalyzing the data in Figure 2H. The second is – in the opinion of this reviewer – key to demonstrating the utility of this approach to study biology of gene regulation at native loci, and requires additional – technically trivial – experimentation.

We thank the reviewer for bringing up these two key points. We have sought to address these as outlined in response to this reviewer's Comments #4 and #5 below.

Major comments

4. Fig 2h represents a key dataset. ENGRAM “records” transcription indirectly: enhancer activates promoter, Csy4 is translated off the mRNA driven by the enhancer-promoter combination, then cleaves the transcript, the transcript arms the prime editor to edit the target, the editing takes place (in a way that is biased by the sequence of the barcode), and then NGS measures the % reads with and without enhancer. Every single step in this process potentially has a nonlinear component. To show correlation with actual level of transcription using a log₂/log₂ scatterplot for 300 enhancer constructs is simply not the appropriate way forward here. The authors need to present a more detailed and systematic analysis. Specifically: what exactly can the assay detect in terms of relative enhancer activities? The simplest way to do this is to extract 8 random enhancers from each quartile of normalized expression (by RNA), rank them using RNA levels, and next to those 8 bars of data show the cognate ranking by ENGRAM. The authors have all those data, it is simply a matter of presenting them in way where the reader can for themselves ascertain whether they should be using ENGRAM to study relative enhancer activities.

This is a great suggestion. We have kept the log₂/log₂ scatterplot (**Figure 2c**, reproduced below), because we believe that it is important to show that overall relationship. However, we have supplemented it with precisely the analysis suggested (as we understand it), as shown in **Figure 2d-e** and **Supplementary Figure 4a** of the revised manuscript. In brief, we randomly sampled 8 enhancers from each of four quartiles of RNA-based activity measurement, and now display their activities directly above the cognate editing efficiencies driven by the same elements via ENGRAM recording.

In brief, the ENGRAM ranks match the MPRA ranks reasonably well at least at the level of comparisons between quartiles, e.g. all sampled enhancers in the top quartile of MPRA activity (*i.e.* Q4; most active) have higher ENGRAM activity than all sampled enhancers from the two bottom quartiles (*i.e.* Q1, Q2) (**Supplementary Figure 4a**, reproduced below).

Building on this analysis, to also quantify the extent to which ranks are preserved within quartiles, we also randomly sampled 20 enhancers from within each quartile and calculated their Spearman correlation (and then repeated this procedure 10 times). Even within quartiles, the rank orders of MPRA vs. ENGRAM were reasonably correlated (mean *rho* of 0.55; slightly higher rank-order correlations within the most active quartile) (**Figure 2e**, reproduced below).

Finally, to more globally summarize the preservation of rank (or extent of decay of that property), we also generated an additional plot, in which all 292 enhancers’ activities are shown in rank order (by MPRA activity) on the top, with ENGRAM activities for the same enhancers displayed immediately below (**Figure 2d**, reproduced below).

Supplementary Figure 4. Further benchmarking of ENGRAM. (a) Comparison of CRE ranks for ENGRAM vs. MPRA across quartiles. Eight CREs were randomly sampled from each of four quartiles based on the RNA-based activity measurement (*i.e.* MPRA). The relative activity based on reporters (MPRA) for each set of eight is shown at the top, and the activity for the same CREs based on recorders (ENGRAM) is shown at the bottom. Overall, ENGRAM reasonably preserved the rank of CREs when comparing the quartiles to one another.

Figure 2. Multiplex recording of natural or synthetic CRE activity with ENGRAM. (c) Values correspond to the proportion of each barcode read out from the synthetic DNA Tape (ENGRAM) or from the pegrNAs (MPRA), out of the total. The log-scaled proportions of ENGRAM events recorded to DNA were highly correlated with log-scaled proportions of barcodes measured directly from RNA. (d) ENGRAM reasonably preserves the overall rank order of CRE activity as measured by MPRA. CREs were first ranked by their activity measured by reporters (MPRA; RNA-seq, top); these activities are compared to activity measured by recorders (ENGRAM; DNA-seq, bottom), plotted in the same order. (e) Boxplot of Spearman correlations within each quartile of CRE activity. CREs were split into 4 quartiles based on their activities measured by MPRA. Within each quartile, 20 CREs were randomly sampled, and then their rank order, as measured by MPRA, was compared to that as measured by ENGRAM. Each point represents one of 10 iterations of sampling. The box represents 25th percentile, 50th percentile and 75th percentile and whiskers represent 1.5× of the inter-quartile range (IQR).

5. Every experiment in the manuscript is done on reporters – randomly integrated via PiggyBack, no less. This is, in plain terms, a cross between a necessity-driven compromise and an atavism. The genome regulatory literature is equal-parts replete with massively parallel reporter assays for regulatory element activity and data showing such reporters offer modest insight into what actual enhancers do in their native context. The advent of genome editing has rapidly relieved the field of the yoke of random integration and enhancer fragments removed from their native context. It is technically trivial for the authors to use plain-vanilla editing to integrate their Csy4-pegRNA tandem into stop codons of any number of native silent human genes (Manu Leonetti at the Biohub and Ru Garawande at the Allen have tagged near-everything at the native locus), and then activate them. A conceptually trivial set of targets for this would be genes regulated, for instance, by type 2 NHRs, where baseline level of activity is essentially zero. It would be, in the opinion of this reviewer, important that the authors add a Fig 5 to the manuscript in which they take 2-3 genes responsive to NF- κ B, or Wnt, or the thyroid hormone, integrate their reporter into the stop codon of those genes with a trivial 2A, and see how well the recording works. One does not need a single-cell-derived clone (which the authors refer to – in a puzzling lexical crossover with the field of immunology – monoclonal) line for this effort; – a pool with a sufficiently high level of the “gene->reporter” allele would suffice. This simple experiment – if successful – would make a bona fide impactful addition to the field of studying human gene control.

[REDACTED]

Therefore, we instead sought to address this comment through a different, arguably more ambitious experiment, which takes advantage of T7-based protocols that we’ve recently developed to efficiently map sites-of-integration³². As a large portion of the mammalian genome is transcribed, we reasoned that random integration of the “minimal” portion of the ENGRAM cassette containing the hairpin-flanked pegRNA might enable the global recording of transcriptional activity across the entire genome.

To test this idea, we devised a strategy to capture both the integration site of minimal ENGRAM cassettes (*csy4*-pegRNA-*csy4*, flanked by a T7 promoter as well as a unique barcode) and their transcriptional activities. In this setup, there is no enhancer or minimal promoter (**Reviewer Figure 1a**). Rather the idea is that many of these minimal ENGRAM cassettes would be integrated to introns or UTRs, their pegRNAs transcribed as part of hnRNA, released by Csy4, and then free to write specific insertions to the endogenous *HEK3* locus (DNA Tape) at a rate proportional to their abundance.

After bottlenecking a population of cells in which this construct had been randomly integrated, we used a T7 *in vitro* transcription assay to map the integration site of every pegRNA construct³², while also identifying which unique insertional barcode was associated with which unique integration site (**Reviewer Figure 1a**). Then, after expressing PE2 and Csy4 in these cells, we sequenced barcodes that had been written to the endogenous *HEK3* DNA Tape, and sought to ask whether recording levels correlated with endogenous transcription levels of the genes to which they were integrated. Overall, we mapped 109,626 integration sites across 10 pools of cells (~1000 cells per pool, median 10,730 integration sites per pool as we were at an MOI of ~10) and classified the sites based on their site of integration. Broadly they fell into two subgroups: genic (promoter, exon and intron) and intergenic (long/short non-coding RNA, ribosomal RNAs and unannotated

regions). Recording efficiencies of the endogenous *HEK3* site were modest (1-2%), but sufficient for reasonable reproducibility (**Reviewer Figure 1b-d**). Of note, the same population of cells expressing PE2 alone did not show detectable recording activities, suggesting the observed recording is truly driven by the minimal ENGRAM cassettes integrated into diverse sites across the genome (**Reviewer Figure 1d**).

However, the results were different from what we anticipated. Among the insertional barcodes written to the DNA Tape, ~77% were associated with the 0.33% of pegRNA cassettes whose integration sites fell within the 45S ribosomal DNA array (rDNA), a stunning 238-fold enrichment. The 45S rDNA array consists of repeats of a 45 kb rDNA unit, including an actively transcribed region (from 5' ETS to 3' ETS) and an intergenic spacer (**Reviewer Figure 1i**). Within each rDNA unit, almost all active ENGRAM cassettes (*i.e.* those for which the corresponding barcode was written to DNA Tape) mapped to the Crick strand of the active transcription unit (**Reviewer Figure 1i-j**). Thus, we can attribute the majority of recording via “minimal ENGRAM cassettes” that happened to integrate within rDNA to the highly active transcription of ribosomal genes.

In contrast we did not observe appreciable differences between other classes of integration sites, nor did we observe the expected correlation between recording levels from pegRNAs integrated within Pol-2-transcribed genes and their endogenous expression levels (**Reviewer Figure 1h,k**). This is not what we expected, and we still do not fully understand this result. However, we speculate that recording of endogenous transcription may be modulated by many more factors than is the case when ENGRAM recorders are placed within a more constant context. This includes the extent to which hnRNA is accessible to Csy4 processing, hnRNA pre-processing, etc. Such factors might allow us to capture a massive enrichment for rRNAs that happened to land within highly transcribed rRNAs, while still leaving us unable to quantitatively record from more lowly expressed Pol-2 transcripts in which the minimal ENGRAM cassette has landed in a diversity of locations/contexts.

For the time being, we have not included this experiment in the revised manuscript, as many other new experiments have been added such that we would risk bloating the manuscript by including it. However, given how interesting the rDNA results are in particular, we debated for some time whether to include it as a **Supplementary Note**. We welcome the reviewer's opinion on this point.

To summarize and coming back to the original point raised by this comment, we sought an alternative way of addressing the point, but with results that were more complex to interpret than we'd hoped. We would also like to raise some strong counterpoints to the assertion that randomly integrated (or more generally, “out-of-native context”) reporters cannot capture biological complexity. In particular, we would point to the fact that randomly integrated, kilobase-sized developmental enhancers in front of conventional reporters are clearly capable of recapitulating exquisitely specific and complex patterns of gene expression in *in vivo* mouse and fly transgenics³³⁻³⁵. Indeed, two papers appeared just this past week in which *de novo* designed synthetic enhancers, driving conventional reporters, were able to recapitulate complex patterns of gene expression during *Drosophila* embryogenesis^{36,37}.

In our view, the modest insights yielded by MPRAs to date have other causes. In particular, we would point to: (a) the strong tendency of MPRAs to be deployed in static, immortalized cancer cell lines rather than in dynamic, *in vivo* developmental contexts; and (b) the strong tendency of MPRAs to use relatively short fragments due to constraints on microarray-based oligonucleotide synthesis. Of note, in a recent preprint from our group implementing a new quantitative single-cell MPRA assay (“scQers”), we found that site-of-integration

effects on the specificity of autonomously active germ-layer specific developmental enhancers are quite modest, consistent with the transgenic mouse and fly experiments that we cite above (see Fig. S12 in <https://www.biorxiv.org/content/10.1101/2022.12.10.519236v1>).

[REDACTED]

Reviewer Figure 1. Recording endogenous transcription from randomly integrated ENGRAM mini-cassettes. (a) Schematic of endogenous transcription recording. The recorder library with T7 promoter, pegRNA encoding 8bp insertion barcode (insBC), and 16bp location barcode (locBC) was integrated into K562 cells using piggyBac. Cells were split into 10 pools of 1000 cells/pool. After expansion, half of the cells were used to map integration sites, and the other half was

used to record endogenous transcription activities. For mapping, genomic DNA is harvested and the genomic DNA is transcribed along with insBC and locBC using *in vitro* transcription. Integration sites were mapped based on a unique combination of insBCs and locBCs. For recording, PE2-Csy4 was transiently transfected in triplicates, and cells were harvested 3 days post-transfection. **(b)** Upset plot summarizing the integration site distribution, with 60.3% of the sites mapped to the genic region and 39.7% mapped to the intergenic region. **(c)** Endogenous gene expression recordings are reproducible across replicates. Two replicates of clone 5 were presented. Each dot represents a unique insBC. **(d)** Overall recording efficiency across 10 pools. Recording is only detectable when both PE2 and Csy4 are present. **(e)** 2.0% of the integrations are recorded (median, n=10). **(f)** The abundance of barcodes associated with rRNA transcripts. 77.4% of the recovered insBCs are associated with rRNA. **(g)** The proportion of integration sites mapped to ribosomal RNA clusters. 0.33% of the sites contribute to 77.4% of the recording, representing a ~238-fold enrichment. **(h)** Editing score by transcription category. Recording events were split into 6 categories based on their integration location. insBC associated with rRNA is much more abundant than other categories. P-values were obtained using the two-tailed Welch's t-test. **(i)** Integration of ENGRAM recorders into rDNA clusters. The vast majority of the active pegRNAs ("recorded") were oriented concordantly ("+") with the rDNA transcriptional unit (5'ETS to 3'ETS). **(j)** Boxplot showing the editing score (log2 ratio of barcode proportion) of pegRNAs integrated in a manner concordant ("+") vs. non-concordant ("-") with the orientation of the rDNA transcriptional unit. P-values were obtained using the two-tailed Welch's t-test. **(k)** Endogenous transcription recording is poorly correlated with gene expression in K562 cells, suggesting potential effects from factors such as gene functions and intron stability.

Minor

6. The acronym ENGRAM elegantly evokes the orthographically congruent term from neuroscience. The authors may wish to point that out explicitly to those readers who are unfamiliar with what that word means.

We thank the reviewer for appreciating the acronym, and agree it requires some context. We have added the following sentence in the revision:

"In neuroscience, an "engram" refers to the physical manifestation of a unit of memory."

7. The authors – appropriately so – devote a considerable length of time on pp 2-3 to comparing ENGRAM to other methods and then pointing out – using a traditional "However" clause – why ENGRAM is better. This is appropriate but makes for difficult reading. The authors may consider relegating all that to one paragraph in the Discussion, eg "The field has developed methods X, Y, and Z. In contrast, ENGRAM attains X', Y', and Z'."

We thank the reviewer for this suggestion. We see the point, but we would like to mention these predecessors upfront and contextualize ENGRAM so as not to risk any reader thinking that the broad concept of molecular recording is new. However, in response to this and related comments from the other reviewers, we have sought to make the introduction tighter and more readable.

We have also sought to achieve something like the suggested X, Y, Z framing via a new table (**Supplementary Table 1**) that compares features of traditional CRISPR cut-based recording, base editing-based recording, and prime editing-based recording. We certainly welcome feedback on the content and clarity of this new table:

Supplementary Table 1 Comparison of strategies for CRISPR-based molecular recording.				
		CRISPR cut	Base editing	Prime editing
Writing	Editing modality	Cas9 nuclease	Base editor	Prime editor
	Other writing components	gRNAs + targets OR homing gRNAs/stgRNAs	gRNAs + targets OR homing gRNAs/stgRNAs	pegRNAs + shared target
	Information content of edit sequence	pseudorandom	pseudorandom	signal specific
	Potential for recording temporal order of biological signals	Poor (current methods do not facilitate ordered recording)	Moderate (system complexity grows exponentially with number of biological signals, because guides/targets must be preprogrammed for each order)	High (system complexity grows linearly with number of biological signals, with ENGRAM + DNA Typewriter)
Storing	Signal location	Signals write to distinct sites or sets of sites	Signals write to distinct sites or sets of sites	Signals write to a shared site or sets of sites
Reading	Ease of recovery of target site(s)	Poor (target site(s) corresponding to each signal must be recovered separately; further challenge of DSB-mediated deletions deleting information and/or compromising primer sites)	Moderate (target site(s) corresponding to each signal must be recovered separately)	High (shared target site(s); compatible with sc-RNA-seq)
	References	(McKenna et al. 2016; Perli et al. 2017; Frieda et al. 2017; Chan et al. 2019; Bowling et al. 2020)	(Tang and Liu 2018; Farzadfard et al. 2019)	This paper

8. Fig 1a is confusing – the text refers to a “common DNA tape,” Fig 1a shows three such loci / tapes. Further, the idea to use Csy4 is elegant but Fig 1b is confusing. The authors should consider having a figure that explains, at a high level, both the system and the process of recording. I.e., here is the reporter construct (a given enhancer drives expression of a transcript that is processed, by Csy4, to yield a “liberated pegRNA”); the cell contains a prime editor that binds to that pegRNA and inscribes the cognate edit into a recording locus. Sequencing the locus determines whether the enhancer was active, and if there are multiple enhancer constructs in the same cell, or pool of cells, you can measure their relative activities. Essentially the figure needs to represent the first 4 sentences of the Discussion section. The current Fig 1a does not entirely do ENGRAM justice – certainly not for a broad audience.

We thank the reviewer for these excellent suggestions. We have substantially revised **Figure 1a** along these lines, with a “walk through” in the legend. We certainly welcome any suggestions for its further improvement.

Figure 1. ENhancer-driven Genomic Recording of transcriptional Activity in Multiplex (ENGRAM). (a) Schematic of ENGRAM. Naturally occurring or designed cis-regulatory elements (CREs) drive activity-dependent transcription of a prime editing guide RNA (pegRNA) encoding a CRE-specific insertion. The CRE could be a natural enhancer sequence (notably tested out of context, as in a conventional or massively parallel reporter assay) or a designed sequence (e.g. spacer-separated repeats of a transcription factor motif). The pegRNA is embedded within a Pol-2 transcript, flanked by two 17bp *csy4* hairpin such that it can be released by Csy4 ribonuclease. The liberated pegRNAs are designed to write a CRE-specific insertional barcode to a common DNA Tape site, which could be an endogenous genomic locus (e.g. the native *HEK3* site) or a synthetic DNA Tape (e.g. exogenously introduced via lentiviral or piggyBac integration). Sequencing of the DNA Tape reveals which CREs were active; if there are multiple ENGRAM reporters in the system, their relative activities can be measured across a pool of cells and/or multiple DNA Tape sites in each cell.

9. Fig 1d shows that hexamers are recorded at vastly different efficiencies – it is unclear at this point in the manuscript what implications this experiment has for subsequent work shown. A way to address this would be to say – “we rigorously controlled for this bias in subsequent experiments.”

This is a great point, and we apologize for not being clearer about how this issue was dealt with for subsequent experiments. In a nutshell, we addressed this by either minimizing the bias (by choosing similarly efficient barcodes) or controlling for it (e.g. by only making comparisons of a recorder/barcode pairing to itself in a different context). As the precise approach varied, we added a new table to make it transparent how we dealt with it in each experiment (**Supplementary Table 6**, reproduced below). We now state:

“How insertional barcodes were chosen for subsequent experiments described in the remainder of the paper is summarized in **Supplementary Table 6**.”

Supplementary Table 6 Overview of insertional barcode choice for various experiments			
Experiments described in..	Insertion Type	Balanced?	Comments
Figure 1	Degenerate 5N insertion	No	It is in these experiments that different insertion rates for various 5-mers is documented and characterized
Figure 2a-e, Supplementary Figure 3e-g	300 x 6 bp insertions	No	Ideally these would have been selected to optimize for balance, but in practice they were not. However, we did calculate ES scores by normalizing based on insertion proportions measured in a separate experiment in which a constitutive promoter drove pegRNAs encoding a degenerate hexamer. But because the resulting ES did not

			appreciably change the correlation with MPRA results (in fact, it reduced the Pearson correlation from 0.889 to 0.84), we plotted the unnormalized measurements.
Figure 2f-g, Figure 5b-h, Supplementary Figures 5, 8	98 x 5 bp insertions	Partially	Most (66) of the barcodes were balanced (within 2-fold) although some were more (n=9) or less (n=23) beyond that range. However, in all experiments where these ENGRAM recorders were being used, we were making comparisons with the relative levels of recording for the same recorder/barcode deployed in different contexts (e.g. different timepoints or 24-hr windows of gastruloid induction), such that balancing is not strictly necessary.
Supplementary Figure 3a	4 CREs x 2 barcodes each	Yes	Barcodes with similar levels of bias were used for this experiment.
Figure 3, Supplementary Figure 6	3 signal responsive CREs x 1 barcode each	Yes	Barcodes with similar levels of bias were used for this experiment.
Figure 4, Supplementary Figure 7	DNA Typewriter	Yes	Barcodes with similar levels of bias were used for this experiment.

10. Supp fig 1 refers to 4% editing as being of “reasonable efficiency.” Without wishing to quibble over semantics, in December 2021, 4% editing in transformed cells should not be described by the epithet “reasonable” – 40% would be. Vide infra for implications of this low efficiency.

We have replaced “reasonable efficiency and reproducibility” with “reproducible efficiency” in this figure legend (now **Supplementary Figure 2a-c** with improved data, reproduced below together with original corresponding panels; Spearman correlation for log-scaled insertion proportions between technical replicates is 0.99) to minimize the odds of misunderstanding. Of note, the improved correlation among three replicates is attributable to recovering more DNA Tapes from the genome by doing more independent PCR reactions.

Also, the reagents available for prime editing have improved considerably since our original submission. Specifically, the prime editor has been improved from previous Prime Editor-2 (“PE2”) to PEmax with optimized Cas9 nickase, reverse-transcriptase, and nuclear-localization domains. We tested this construct both in K562 cells and in mESCs and it reproducibly exhibited 1.7-fold (K562) or 9.3-fold (mESC) increases in editing efficiencies under the same recording conditions compared to PE2. (**Supplementary Figure 1h**; reproduced in left panel below). While we are still below 40%, we believe that further optimizations of prime editing by the community will be readily adaptable to improve ENGRAM’s recording efficiency even further.

Supplementary Figure 2 (with corresponding panels from original submission immediately above). The ENGRAM recorder installs barcodes with reproducible efficiency. (a-c) The relative proportions of 1023 5N barcodes installed by ENGRAM driven by the constitutive Pol-2 PGK promoter were measured in triplicate. Log-scaled insertion proportions (calculated as the proportion of edited *HEK3* sites with a given insertion) were strongly correlated between pairs of transfection replicates.

Supplementary Figure 1. Optimization of ENGRAM architecture. (h) Comparison of recording efficiency between prime editor variants. PE2 or PEmax, together with a library of constitutive ENGRAM recorders programming a degenerate 5N insertion to the *HEK3* locus, were co-transfected into K562 cells. In a separate experiment, an ENGRAM recorder encoding a 5-bp insertion was transiently transfected into mESCs bearing a synthetic DNA Tape (*HEK3*) and stably expressing PE2 or PEmax. Genomic DNA was harvested 3 days post-transfection and insertions at endogenous (K562) or synthetic (mESC) *HEK3* loci were quantified. We observed that PEmax drove 1.7-fold and 9.3-fold more efficient recording than PE2 in K562 cells and mESCs, respectively. Although we used PE2 for many of the experiments reported in this paper (as PEmax had yet to be described at the time that they were performed), we recommend using PEmax for all future recording assays. Error bars correspond to standard deviations across 3 transfection replicates. P-values were obtained using the two-tailed Student's t-test.

11. Fig 1f shows the dynamic range of recording in this 1.0 version of the ENGRAM system is a low one – as the authors correctly state. The explanation given – “due in part to the accumulation of background edits due to constitutive expression of Csy4, which may also have reduced growth of the monoclonal cell line due to cytotoxic effects” – is hard to understand and may be best omitted. The efficiency of editing is due to the amount of pegRNA (one presumes) – which, in turn, is due to transcriptional activity of the minimal promoter with and without enhancers. Surely it’s the latter that is the most parsimonious reason for the “considerably more modest S/N” – rather than cytotoxicity?

This is a good point. We have removed mention of cytotoxicity as a potential explanation, as we have no direct evidence for it. We agree that differences in efficiency is a more parsimonious explanation.

In fact, in new experiments focused on deploying ENGRAM in the context of gastruloids, a differentiating *in vitro* model of early mammalian development (see our response to Comment #5 from Reviewer #1), we found that the presence of ENGRAM recorders at a high MOI (n = 10-15) did not interfere with gastruloid induction or morphology, consistent with minimal effects on cell viability (**Supplementary Figure 8b**, reproduced below).

Supplementary Figure 8. Multiplex recording of CRE activity in embryonic stem cells and differentiating gastruloids. (b) Representative images of gastruloids induced from polyclonal ENGRAM mESCs, illustrating that the components of the ENGRAM recording system do not substantially impact the morphological development of gastruloids. Scale bar: 100 μ m.

12. The system optimization described in the first two paragraphs of page 7 is of interest only to a specialist audience and can be relegated to supplementary information.

We agree that this was too detailed in the submitted version of the manuscript, and Reviewer #3 makes a similar point. In the revision, we have greatly shortened the description of the development and optimization of

ENGRAM, moving most details about ENGRAM 1.0 into the supplement, and really focusing on ENGRAM 2.0. We moreover consolidated Figures 1 & 2 from the original paper into a single figure (**Figure 1** of the revised manuscript, reproduced below). Of note, we retained our comparison of three ENGRAM 2.0 architectures (including a newly added experiment in which we compare these side-by-side) in **Figure 1b-d**, but all other optimization details have been relegated to the corresponding supplementary figure legends, with only brief mention in the main text.

Figure 1. ENhancer-driven Genomic Recording of transcriptional Activity in Multiplex (ENGRAM). (a) Schematic of ENGRAM. Naturally occurring or designed cis-regulatory elements (CREs) drive activity-dependent transcription of a prime editing guide RNA (pegRNA) encoding a CRE-specific insertion. The CRE could be a natural enhancer sequence (notably tested out of context, as in a conventional or massively parallel reporter assay) or a designed sequence (e.g. spacer-separated repeats of a transcription factor motif). The pegRNA is embedded within a Pol-2 transcript, flanked by two 17bp *csy4* hairpin such that it can be released by Csy4 ribonuclease. The liberated pegRNAs are designed to write a CRE-specific insertional barcode to a common DNA Tape site, which could be an endogenous genomic locus (e.g. the native *HEK3* site) or a synthetic DNA Tape (e.g. exogenously introduced via lentiviral or piggyBac integration). Sequencing of the DNA Tape reveals which CREs were active; if there are multiple ENGRAM reporters in the system, their relative activities can be measured across a pool of cells and/or multiple DNA Tape sites in each cell. (b) Three distinct ENGRAM 2.0 architectures with the *csy4* hairpin-flanked pegRNA embedded in the 3' UTR (top) or 5' UTR (middle) of a transcript encoding Csy4, or 3' ENGRAM 2.0 with an additional *csy4* hairpin in the 5' UTR (bottom). The presumed cleavage of the *csy4* hairpin at the 5' UTR in both the 5' ENGRAM and 3'-FT ENGRAM architectures may impose an auto-regulatory negative feedback on Csy4 levels. (c) All three ENGRAM 2.0 recorders were integrated via

piggyBac into PE2-expressing cells in triplicate, each driving 1,024 potential barcodes (5N degenerate insertion) with minP. The background editing efficiency was periodically checked over 20 days. Error bars correspond to standard deviations across 3 transfection replicates. **(d)** An NF- κ B response element was cloned upstream of minP in each of the three ENGRAM 2.0 designs. These ENGRAM recorders were separately integrated into the genomes of PE2(+) HEK293T cells via piggyBac. We then measured recording at the endogenous *HEK3* locus in the absence vs. presence of 10ng/ml of TNF α in triplicate. Error bars correspond to standard deviations across 3 replicates. P-values were obtained using the two-tailed Student's t-test. **(e)** To assess impact of the sequence of the inserted barcode, a library of pegRNAs encoding a degenerate 5-mer insertion was cloned into a constitutively active 5' ENGRAM 2.0 architecture. **(f)** Log-scaled insertion proportions (calculated as the proportion of edited *HEK3* sites with a given insertion) were highly correlated between transfection replicates. **(g)** Range of editing scores (ES) for 5N insertions. ES are calculated as (genomic reads with specific insertion/total edited *HEK3* reads)/(plasmid reads with specific insertion/total plasmid reads), plotted here in rank order on a log₂-scale. 948 of 1024 potential 5-mers were recovered after removing highly underrepresented barcodes. For several of the most highly or lowly ranked insertional barcodes, we list the insertions observed in DNA Tape, which are the reverse complement of barcodes encoded by pegRNAs. **(h)** A linear lasso regression model trained on these data with one-hot encoded single and dinucleotide content of the 5-mer and MFE of secondary structure as features. Samples were split with 680 barcodes in a training set and 268 barcodes in a test set. The model was trained with 10-fold cross-validation on the training set and then used to predict the test set.

13. The purpose of the effort in Fig 2 was to address the problem of low dynamic range shown in Fig 1f. Why are the data in Fig 2f represented using a different metric - % insertions – than Fig 1f (editing with and without enhancer)? Are the data cognate in Fig 1f – in supp fig 2b? “To push the system further” is inappropriately colloquial and imprecise.

To the first point, we apologize and agree (this same point was raised by Reviewer #1). The original Figure 1f and Supplementary Figure 2b were the same data but presented in two ways. To improve transparency, we now only present absolute editing efficiency. The referenced main text figure (Figure 2f) has been moved to the supplement and is now **Supplementary Figure 3a-b**, reproduced below.

Supplementary Figure 3. Benchmarking of ENGRAM against reporter assays. **(a)** ENGRAM recorders with highly vs. lowly active CRE fragments (as previously measured via MPRA) upstream of a minP, together with minP-only and promoter-less constructs, were cloned, each driving expression of two distinct pegRNA-encoded barcodes. **(b)** Barcodes corresponding to the highly active CRE were 15.1-fold, 23.6-fold, and 41.3-fold more abundant than barcodes corresponding to lowly active CRE, minP-only or promoter-less controls, respectively. Error bars correspond to standard deviations across 3 transfection replicates. P-values were obtained using the two-tailed Student's t-test.

To the second point, we have revised the colloquial/imprecise text to read:

“To further evaluate multi-channel signal recording, we performed a separate experiment in which a subsamples of a population of cells bearing all three recorders were exposed to all possible combinations of low, medium or high concentrations of each stimulus (3 stimuli^(3 concentrations) x 3 replicates = 81 conditions).”

14. Could the authors briefly clarify what they mean by “ORI-driven transcription” on p 10? Plasmids lacking viral origins of replication (eg EBV) do not replicate in mammalian cells using defined origins (ORI) of replication unless a factor like EBNA is present. In other words, generic plasmids do not have an “ORI” that functions as such in mammalian cells. Do the plasmids being used have such ORIs? Or is ORI an acronym for something else?

We apologize for the confusion. The issue that we were referring to here is rather niche: the bacterial ORI in plasmids typically used for non-integrating MPRA assays exhibit substantial promoter activity in mammalian cells (which turned out to be a major confounder for the original version of the STARR-seq assay, cleverly addressed by simply using the bacterial ORI as the minimal promoter)^{38,39}. We reasoned that this might be a source of background activity during piggyBac transfection, *i.e.* with the bacterial ORI driving a bit of enhancer-independent transcription after transfection but prior to integration. To be clearer on this, we now say:

“We speculate that this background editing is due to bacterial origin-of-replication-driven transcription on plasmids^{28,29} prior to integration, rather than CRE/minP-driven transcription from integrated recorders.”

15. While the sigmoidal curves in Fig 3b are impressive, as the authors know, a difference between 1.5% and 3% editing is not what one wants for an editing-based readout (also *vide supra*). The actual dynamic range of editing by NGS is between 0.5% (amplicon resequencing without UMIs) and 95% indels (presumably for PE this is lower). It’s impressive the triplicates (unclear if biological – ie different experiments start to finish, or NGS in triplicate) are robust, but for the average user of this method, working with NGS “dynamic ranges” between 1.5% and 3% will be very hard. How many reads per amplicon did the authors have to use in order to get the numbers they got?

This is an important comment. In the revision, we have sought to address it via multiple updates:

1. We modified the text and figure legend to make it clear that the triplicates in this case are biological replicates, not technical replicates (*i.e.* independent measurements, from “start to finish”, *i.e.* ligand exposure to NGS library prep and sequencing, rather than solely NGS in triplicate). The revised text now reads as follows:

“The resulting recorders were separately integrated into PE2(+) HEK293T cells via piggyBac (for the doxycycline recorder, a constitutively expressed reverse tetracycline-controlled transactivator (rtTA) was integrated separately). All cells were cultured for at least a week before agonist exposure to dilute out unintegrated recorder plasmids. A 2-fold dilution series of doxycycline, TNF α or CHIR99021 was added to the medium of the corresponding recorder cells in triplicate, and genomic DNA was harvested 48 hours after the onset of exposure.”

2. We constructed a new recorder cell line with greatly reduced background recording levels (0.2% to 0.4%) and repeated the referenced experiment. For context, in the original submission, we tested the idea of using a non-targeting pegRNA to minimize recording from the plasmid before integration. However, during

revisions, we identified conditions under which conventional transfection achieves similarly high signal-to-noise. In particular, we found that background could be minimized by optimizing transfection conditions (*i.e.* the amount of total plasmid and the ratio of cargo and piggyBac transposase) This improved the dynamic range to about 10-fold for the TetON recorder, nearly 20-fold for the NFκB recorder, and greater than 20-fold for the Wnt recorder. The revised data is presented in **Figure 3b-e**, reproduced below:

Figure 3. Multi-channel recording of the intensity and duration of signaling pathway activity. (b-d) Upon 48 hours of stimulation with the corresponding stimulant, the Tet-On (b), NF-κB (c), and Wnt (d) recorders exhibited dose-dependent levels of recording. Signal recorder cells were constructed by integrating corresponding recorders into PE2(+) HEK293T cells via piggyBac. Recorder cells were cultured at least 7 days before recording to minimize the recording activity from unintegrated plasmids. Cells were exposed to a serial two-fold dilution series of doxycycline (b), TNFα (c) or CHIR99021 (d), with starting concentrations of 8 μg/ml, 64 ng/ml and 32 μM, respectively. For CHIR99021, additional concentrations were sampled between 1 to 4 μM. All recording experiments were performed in triplicate. The half-maximal effective concentration (EC₅₀) for doxycycline, TNFα, and CHIR99021 are 0.17μg/ml, 2.5ng/ml and 2.2uM, respectively. The concentrations tested here fall into the range in which these agonists are typically used^{20,21,23,40,41}. (e) Signal-to-noise. The Tet-On, NFκB and Wnt ENGRAM recorders respectively exhibit a 11.5-fold, 19.0-fold and 22.6-fold difference in editing levels with vs. without the maximum dose of the corresponding agonist. Error bars correspond to standard deviations across 3 replicates.

3. We agree that if there is only one recording site per cell and the recording efficiencies are very low, then large numbers of cells and/or reads may be needed to obtain reproducible estimates of relative recording efficiencies. To systematically explore whether we could reduce cellular input requirements by having

multiple DNA Tapes per cell, we performed a new experiment in which we integrated synthetic DNA Tape (bearing *HEK3* target) into K562 cells at a high MOI (~20) and repeated the 300-plex enhancer recording experiment (schematic in **Figure 2a**, reproduced below):

Figure 2. Multiplex recording of natural or synthetic CRE activity with ENGRAM. (a) Schematic of multiplex recording experiment. A library of ENGRAM reporters bearing 300 CRE fragments, each linked to a unique pegRNA-encoded insertion, was constructed and integrated into PE2(+) K562 cells via piggyBac. CRE activity was then recorded to both endogenous (genomic *HEK3*, n=2-3) or synthetic (*HEK3* locus integrated into the genome via piggyBac, n~20) DNA Tape. For the purposes of benchmarking, the relative activities of CREs were measured either via recording (ENGRAM, DNA-seq of *HEK3*) or reporting (MPRA, RNA-seq of pegRNAs).

“Average” editing rates at the ensemble of randomly integrated synthetic *HEK3* sites were about 2-fold lower than that of the endogenous *HEK3* site (**Supplementary Figure 3c**, reproduced below), likely due to chromatin-mediated effects, as we recently showed in a pre-print³². While recovering recorded information from a large number of cells (~300k), our estimates of the relative activity of the ~300 enhancers were highly reproducible between endogenous and synthetic *HEK3* recording sites (**Supplementary Figure 3d**, reproduced below). To assess robustness, we recorded CRE activity from various numbers of input cells. With synthetic DNA Tape (~20 copies per cell), we were able to record the relative activities of the CREs from as few as 12,000 cells (*i.e.* 40 cells per CRE) with reasonable recovery and reproducibility (291 of 300 detected; mean Pearson’s correlation of 0.87 between replicates; **Supplementary Figure 4b-c**, reproduced below). By downsampling the number of reads used in association with the 96,000 cell input condition, we found that 250,000 reads were sufficient (278 of 300 detected; mean Pearson’s correlation of 0.87 between replicates), although more reads were helpful (**Supplementary Figure 4d-e**, reproduced below). Looking forward, input and sequencing requirements could presumably be reduced by either increasing the number of DNA Tape units in each cell or by further increasing prime editing efficiency.

Supplementary Figure 3. Benchmarking of ENGRAM against reporter assays. (c) Insertion efficiency of various barcodes at synthetic (1.8%) vs. endogenous (3.1%) *HEK3* loci are highly correlated. Of note, for synthetic *HEK3* sites, the observed efficiencies reflect an average across many genomic contexts. (d) Log-scaled insertion proportions for 300 6-mer barcodes were highly correlated between DNA Tape sites located at synthetic vs. endogenous *HEK3* loci.

Supplementary Figure 4. Further benchmarking of ENGRAM. (b-c) Different cell numbers were sampled (6,000, 12,000, 24,000, 48,000, 96,000 cells) prior to measuring ENGRAM recorded activity of 300 CREs, either from endogenous and synthetic DNA Tape, and then recovery (b) and reproducibility (c) were assessed. (d-e) Sequencing data from synthetic DNA Tape and 96,000 cell input condition was downsampled, and then recovery (d) and reproducibility (e) were assessed.

16. The interesting experiment in Fig 4 is missing a key technical bit of editology: are the authors counting individual haplotypes (ie alleles in a given cell) for evidence of Boolean logic such as shown in Fig D? If yes, can the authors show details of the “shared DNA tape”? Where is the gRNA binding site, what are the sequences the pegRNA for each signal encode, and what happens when multiple pegRNAs compete, in the same cell, to write different alleles of the DNA tape? Conversely, are the individual recordings one and done, and in this case, the authors are reading this out on a pool level (ie counting stimulus-specific barcodes)? This is confusing even for this reviewer – who is familiar with the underlying technologies – and the authors should clarify.

We apologize that the experiment shown in Figure 4d of the original manuscript was not clearer on these points. However, the experiment has been removed, as the combination of ENGRAM and DNA Typewriter

represents a far more powerful path forward for DNA-based recording of the temporal order of biological signals. The combination of ENGRAM and DNA Typewriter is described in new **Figure 4, Supplementary Figure 7** and corresponding additions to text. For a more detailed summary within this point-by-point, please see our response to Comment #1 from Referee #1 above. In these experiments, we write biologically conditional symbols in an ordered fashion to a DNA Typewriter Tape which is indeed read out as a ‘haplotype’. However, our ultimate conclusions about the precise signal pattern are based on the ensemble of haplotypes, sequenced from multiple Tapes in each of many cells.

To the broader question about whether pegRNAs are competing for the same Tape and how we are reading these experiments out, this is the case for nearly all experiments in the original or revised manuscript, e.g. the 300-plex CRE experiments, the 98-plex synthetic CRE experiments in the context of cell lines or gastruloids, the quantitative 3-plex signal recording experiments from original Figure 4 (now **Figure 3** in the revision), and the ENGRAM + DNA Typewriter experiments. The results are then read out at a pool level, *i.e.* with counting of the relative proportions of stimulus-specific barcodes. Because this question was specifically about original Figure 4 (now **Figure 3** in the revision), we added a sentence to its legend that clarifies the experimental design (key sentences underlined):

Figure 3. **(a)** Signal-responsive regulatory elements were used to construct ENGRAM 2.0 recorders for activation by doxycycline (Tet-On; Tet Response Element), TNF α (a NF κ B response element) and CHIR99021 (a TCF-LEF response element, responsive to Wnt signaling). Each recorder encodes one or two unique barcodes and targets the endogenous *HEK3* locus. **(h)** Schematic of multiplex recording of signaling pathways. Three recorders shown in panel a were mixed at an equimolar ratio and integrated to a single population of PE2(+) HEK293T cells. These recorders target a shared DNA Tape, *i.e.* the endogenous *HEK3* locus.

Finally, to address the question about detailed information about binding sites, etc., we have added more detailed schematics of the underlying pegRNA constructs, both for ENGRAM (**Supplementary Figure 1a**, reproduced below) and the combination of ENGRAM and DNA Typewriter (**Supplementary Figure 7a**, reproduced below).

Supplementary Figure 1. Optimization of ENGRAM architecture. (a) Sequence of the pegRNA predicted to be liberated from a Pol-2 transcript by Csy4. The *csy4*-hairpin residuals on both ends are shown in lower case. The spacer and primer binding sequence (PBS) are highlighted in orange. The reverse transcription template (RTT) consists of a homology arm (blue) and barcode (red).

Supplementary Figure 7. Combining ENGRAM and DNA Typewriter to record the temporal dynamics of biological signals. (a) Sequence of the predicted pegRNA modified for compatibility with DNA Typewriter. This pegRNA is similar to the one shown in **Supplementary Figure 1a** except that the spacer and PBS are modified to target the DNA Typewriter Tape's active type guide and the encoded insert is modified to include both a symbol (3-bp in this case) and a key sequence (3-bp).

// Fyodor Urnov – Innovative Genomics Institute and UC Berkeley //

Referee #3 (Remarks to the Author):

This is a very creative and elegant new approach to record the activity of selected regulatory elements (and hence their upstream pathways). Enhancer activities are recorded and integrated over time and intensity, and a proof-of-principle experiment demonstrates that it is also possible to record the temporal order of signaling events. The method is highly multiplexable.

We thank the reviewer for these very positive comments, including characterizing the work as “creative”, “elegant” and “multiplexable”. We also want to make one quick new point to this (non-anonymous) reviewer specifically, because although it did not bring it up here, it came up in the context of a different paper that you reviewed and is relevant here as well. Specifically, we use minP here as we did with the scQer assay (such that we cannot rule out the transcripts being produced are eRNAs as opposed to promoter-initiated). We do not believe that this impacts any aspect of ENGRAM's demonstrated or forward-looking potential, but we have taken care to focus in the manuscript to describe the elements driving ENGRAM recorders as CREs rather than “enhancers”. The one place where we retain the use of the word enhancer is in the acronym ENGRAM, simply because the term is so fitting to the purpose of the technology. We hope the reviewer will give us some leniency there.

1. Strengths of the manuscript are its high conceptual novelty, the careful testing and iterative optimisation of the methodology, the overall high quality of the data, and the clarity of the text and figures. This new approach will inspire other scientists to think along similar lines and perhaps develop further variants of this technique.

We thank the reviewer for their further positive comments, including citation of the work's "high conceptual novelty", high data quality, clarity of the figures, and inspiration that the work will provide to others.

2. But for publication in Nature, I feel that something is missing. While the ENGRAM technology is very cleverly designed, the manuscript does not really demonstrate its practical utility for solving new biological problems. The manuscript remains very conceptual; no new biological insights are obtained. I took some time to think about this, but I cannot think of a biological question in cultured cells that cannot be addressed equally well (or even better) with conventional MPRA. But I would be happy to be convinced otherwise, by one or more clear examples.

This is a fair point, and similar to one raised by Reviewer #1. To paraphrase, despite the work being highly creative/elegant/novel (per this reviewer's earlier comments), we fail to showcase it in a way that shows it achieves something that could not have been achieved via conventional MPRA. To address this comment, we added several experiments that could not have been done with MPRA.

(1) The first set of new experiments involves the integration of ENGRAM and DNA Typewriter, which enables decoding of the order and/or timing of signals in the context of diverse temporal "programs". For details, see our response to Comment #1 from Reviewer #1, as well as new **Figure 4, Supplementary Figure 7** and corresponding additions to the main text. **Figure 4** is also reproduced below in response to this reviewer's Comment #4. In these experiments, we are obtaining results which could not have been obtained via an MPRA. In particular, we observe patterns with a DNA Typewriter Tape that document the recording of one signal followed by another signal within the same cell. In contrast, even new single cell MPRA methods measure only a cell's state at a single point in time, *i.e.* the moment of lysis and RNA purification. In sharp contrast with the ordering scheme in the original manuscript (as well as with CAMERA and DOMINO), all possible orders of many biological signals can be recorded to a shared media. The combination of ENGRAM and DNA Typewriter provides a clear path forward for the time-resolved, multi-channel recording of biological signals, potentially in combination with cell lineage.

(2) The second set of new experiments involves using ENGRAM recorders while differentiating mouse embryonic stem cells (mESCs) into gastruloids, a multi-germ-layer organoid model of early mammalian development. For details, please see our response to Comment #5 from Reviewer #1, or alternatively the entirety of new **Figure 5, Supplementary Figure 8**, and corresponding additions to the main text. The experiment highlights the ability of ENGRAM to capture the cumulative activity of a signaling pathway or a TF over an extended period of time (*e.g.* here, 5 x 24 hr windows that cumulatively "cover" the full 120 hrs), something that is not possible with an MPRA.

We recognize this reviewer may object because these experiments are still in cell culture, and such information could in principle be gathered by serial sampling of an MPRA experiment. However, such inferences about the order of signals would be indirect, requiring serial, static measurements from independent replicates of the system, as opposed to the explicit ordering along a single Tape (or ensemble of Tapes) provided by the combination of ENGRAM and DNA Typewriter. For gastruloids, one could in principle perform MPRA with an extremely large number of sampled timepoints to "cover" the full 120 hrs of differentiation, analogous to the infinitesimal sampling that eventually gives rise to the integral in calculus. However, this would be impractical, particularly for organoid models in which only a limited number of organoids can be reasonably cultured.

3. It seems to me that the real promise of ENGRAM lies in future applications in transgenic animals, to study signalling and gene regulation during morphogenesis and tissue development, in growing tumors, various disease models, etcetera. However, as pointed out by the authors in the Discussion, multiplexing would require integration of large sets of recorders into the genome ("a recorder locus"), and as yet this is still a major challenge.

We agree on where the potential lies and also that it will be challenging. However, it is increasingly possible, especially with technologies that have recently matured in other labs. For example, as recently shown by Jef Boeke's group, it is now possible to design and synthesize >100 Kb loci from scratch, as they did for a refactored version of the *HoxA* locus⁴². We recently initiated a collaboration with the Boeke group on building "one-shot" recorder loci that incorporate components of both DNA Typewriter and ENGRAM. This will take time, and probably multiple iterations, but in our view, the possibility of recording diverse biological signals in the context of an unfolding lineage has transformative potential for how we measure information in complex, dynamic biological systems. Disruptive technologies often enter at a starting point that is behind that of more mature paradigms (e.g. the initial demonstration of massively parallel or next-generation DNA sequencing was around one dozen 4-6 bp reads⁴³).

4. The authors also mention that ENGRAM may record "cellular histories". This sounds exciting, but again there still seems a big gap between the promise and the realisation. Related to this, I am a bit worried about the intrinsic property of ENGRAM that it integrates signals both over time and intensity (Figure 3j-k). This makes it difficult to reconstruct how a signal of medium strength arose: was the enhancer stably active at moderate levels, or was it very active but only part of the time? Both scenarios could give the same output. To deconvolve this, one would have to do time series experiments - but then the added value over standard reporter assays seems lost?

We apologize that our full vision on this point was not clear, and in any case it was not sufficiently demonstrated in the original submission. In new **Figure 4**, reproduced below, we explore a variety of considerably more complex temporal patterns than presented in the original manuscript, via the combination of ENGRAM and DNA Typewriter. In brief, in these experiments, Wnt and TetOn ENGRAM recorders are driving pegRNAs that each encode a signal-specific barcode to be written to DTT, but also the key sequence that shifts the type guide forward (**Figure 4a**). All relevant components (ENGRAM recorders, PEmax, DTT) were first integrated to the genomes of HEK293T cells. Activation of the Wnt and TetON ENGRAM recorders were controlled by the chemical ligands CHIR and doxycycline (Dox), respectively. We then executed a variety of programs in which these chemical ligands were added or removed in different temporal patterns.

In "serial" programs, the first and second signals are not overlapping, but their order and duration is varied (**Figure 4b**, left). In the "layered" programs, the second signal is added in, after some delay, to a constitutive first signal, with both the order and duration of delay varied (**Figure 4b**, right). In "pulse" programs, the TetON signal is added and then withdrawn, in the background of a constitutive Wnt signal, but the timing of the pulse is varied (**Figure 4d**).

Altogether, we tested 15 temporal programs, each in triplicate (45 experiments total). Within the serial and layered programs, we were readily able to determine which signal was introduced first, simply based on the ratio of DNA Tapes bearing the Wnt-specific barcode before vs. after the TetON-specific barcode (TetOn→Wnt

/ Wnt→TetOn in adjacent units of the DTT; **Figure 4b,c**). Moreover, the magnitude of this ratio was informative with respect to the length of time before the second signal was introduced, *i.e.* longer intervals yielding a greater signal-to-noise. Although we only tested intervals down to 24 hrs, the data suggests that we could likely distinguish orders for even shorter intervals, particularly in the case of serial signals (**Figure 4b**). Similarly, in the case of pulse programs, this same ratio was highly informative with respect to whether the second signal was pulsed early, midway or late in the experiment (**Figure 4d**).

On the other hand, this ratio in isolation was not sufficient to distinguish different pattern classes from one another, *e.g.* serial vs. layered patterns. To assess whether all 15 programs were distinguishable on the basis of the ensemble of patterns observed in DTT, we performed principal components analysis (PCA) on 26 variables observed in each of 45 treatments (15 programs * 3 replicates each). These included the proportions of the two symbols at each of the five DTT sites (2 symbols * 5 positions) and the proportions of each possible bigram (Tet-On→Tet-On, Tet-On→Wnt, Wnt→Tet-On, Wnt→Wnt) at pairs of adjacent sites (4 bigrams * 4 position-pairs). Upon plotting the top three PCs, which together explain 90% of the variance, we observe that replicates are tightly clustered (**Figure 4e**). This suggests that the 15 temporal programs of signal transduction give rise to reproducibly distinct ensembles of symbol patterns, as recorded to 5-unit DTT via the combination of ENGRAM and DNA Typewriter.

In summary, these experiments focus on how the combination of ENGRAM and DNA Typewriter enable a diversity of temporal patterns involving two distinct signal recorders to be distinguished via the symbols that are written to a common substrate, DNA Typewriter Tape (DTT). DTT is fundamentally different from other forms of multi-unit DNA-based recording media in that it is written to in an ordered fashion, with the activation of site $n+1$ depending only on site n having been written to but not on what symbol was written to it. And as shown in **Figure 4**, in the combination of ENGRAM and DNA Typewriter, there is more information about temporal dynamics of signals captured to an ensemble of DTTs than simply the signal order.

Of course, there are limitations to what we have achieved so far with this combination. We relied on an ensemble of cells, rather than a single cell, to extract each temporal pattern, and we have yet to develop algorithms or models that would allow one to directly infer signaling histories from a newly observed, complex patterns (as opposed to simply showing after the fact that the patterns are distinct, as we do in **Figure 4e**). We also acknowledge that these experiments do not directly hit on all aspects of the reviewer's comment, as we did not specifically look at moderate intensity (as opposed to on vs. off in various temporal patterns).

On the other hand, these experiments only partly showcase the full potential of this approach, as vastly more signals, orders, dynamics, etc. could be accommodated by this same framework. By introducing and recovering more DNA Tapes into each cell, we can also imagine how the number of cells required for pattern inference might be greatly reduced. We also envision that the combination of ENGRAM and DNA Typewriter might facilitate the concurrent recording of biological signals and cell lineage, *i.e.* with both stochastic and biologically conditional recorders all writing in an ordered fashion to a shared set of DNA Tape(s).

Altogether, we hope that the reviewer agrees that this combination of ENGRAM and DNA Typewriter unlocks the path to a bright future for DNA-based, time-resolved, multi-channel recording of biological signals. Nonetheless, we have sought to tone down our statements in this regard, and really only focus on the full set of possibilities that we believe this opens up in the closing paragraph of the revised discussion.

Figure 4. Combining ENGRAM and DNA Typewriter to record the temporal dynamics of biological signals. (a) Signal-responsive ENGRAM recorders can be modified to drive the expression of DNA Typewriter-compatible pegRNAs (DTT-pegRNA), each encoding signal-specific insertions to be written to the current type guide location of a multi-unit DNA Typewriter Tape (DTT). The insertions record the identity of the signal but also include a 'key' that shifts the location of the type guide by one position. Here we modified the pegRNAs encoded by Tet-On (orange) and Wnt (blue) ENGRAM recorders for DNA Typewriter compatibility. Temporal dynamics (in the simplest case, the order of two serially applied agonists) should be captured in the order in which the corresponding symbols appear in DTT. (b-c) These modified

ENGRAM recorders and 5-unit DTT were sequentially integrated to PEmax(+) HEK293T cells. We designed and tested serial (b) and layered (c) programs in which these cells were exposed to different patterns of 100 ng/ml doxycycline or 3 μ M CHIR-99021 (left columns), as described in the text, across a total of 36 cell populations (2 patterns * 3 intervals * 2 possible orders * 3 replicates). Harvesting, amplifying and sequencing the DTT region after 6 days, we calculated the log2 ratio of [Tet-On \rightarrow Wnt] vs. [Wnt \rightarrow Tet-On] bigrams at Sites 1-2, predicting and observing positive values when Tet-On activation preceded Wnt activation, and negative values when Wnt activation preceded Tet-On activation (right columns). Error bars correspond to standard deviations across 3 transfection replicates. (d) Same as panels b-c, but for pulse programs in which these cells were exposed to 500 ng/ml doxycycline for 24 hrs against the background of continuous stimulation with 3 μ M CHIR-99021. A total of 9 cell populations are represented (3 pulse timings * 3 replicates). (e) PCA based on proportions of unigrams and bigrams observed at each of 5 DTT positions and 4 DTT position-pairs, respectively, across 45 cell populations subjected to various patterns of exposure to doxycycline and CHIR-99021 (15 programs, executed in triplicate). Circled subsets correspond to serial and layered programs in either order, or to pulse programs. The top three PCs are plotted, which collectively explain 90% of the variance.

5. Thus, this is an elegant and interesting new technology, but what will it be used for in the foreseeable future, and how does it outperform existing technologies? I would not hesitate to recommend publication of this work in *Nature Methods* or *Nature Biotechnology*. Readers of these journals will be inspired by the technological innovation, even if the immediate advantages over existing tools are not obvious yet, and no new biological insights are included as evidence. But for publication in *Nature*, I feel that one or more concrete examples are needed to demonstrate that new biological problems can be solved that could not be solved by MPRAs or other reporter assays.

We thank the reviewer for these positive comments. To the point about concrete examples of problems that could not be solved with MPRAs or other reporter assays, please see our response to Comment #2 from this reviewer, as well as our responses to Comment #1 and Comment #5 from Reviewer #1. Alternatively, see new **Figures 4-5, Supplementary Figures 7-8** and corresponding additions to text. In brief, these responses summarize several new experiments that we added that could not have been done with MPRAs. These include the successful combination of ENGRAM and DNA Typewriter, as well as the multiplex “integration” of the activity of synthetically designed CREs over 24-hr intervals collectively spanning a differentiation time-course of mouse embryonic stem cells (mESCs) into gastruloids, a multi-germ-layer organoid model of early mammalian development. In the revised Discussion, we have sought to better emphasize key distinctions between ENGRAM and MPRAs, including that: (a) only ENGRAM permits integration of transient signals over time; and (b) only ENGRAM permits correlation of present state with past state(s) within the same system.

We also note that there are myriad examples of papers appearing in broad readership journals such as *Nature* or *Science* that are innovative technologies with important potential, in which a major new biological insight is not necessarily present. As one example, our initial demonstration of exome sequencing and the possibilities for deciphering Mendelian disorders was entirely a proof-of-concept⁴⁴, and foreshadowed the means of discovering thousands of new disease genes and population-scale cohorts of >1 million individuals. There are many other examples, including fluorescent Sanger sequencing⁴⁵, GFP⁴⁶, doxycycline activation in mammalian cells⁴⁷, single cell combinatorial indexing^{48,49}, prime editing⁵⁰, etc.

However, we recognize that where to draw the line for technological innovation having sufficient potential to warrant publication in a broad readership journal such as *Nature* is a challenging and subjective question, and one that we struggle with ourselves as peer reviewers. Here, we would argue that the multi-channel recording of the temporal dynamics of biological signals has the potential to drive a paradigm shift in how we make

longitudinal measurements in biological systems across the board, as it is fundamentally different from both microscopy/live-imaging as well as from destructive, time-series based genomics (which includes MPRA). This is a vision and a “bet”, for which this paper is an substantive entry point. (Of note, we are essentially betting the entire future of our lab on it, so it’s at least an honest bet!).

Other major points:

6. The text is easy to follow, and in principle it is very informative to learn about the sequential improvement steps that the authors went through. But I do not think that this chronological and rather technical account is suitable for the broad readership of Nature. If the Editor decides to move forward with this manuscript, I recommend a major rewriting/shortening of the text, focusing on the well-working versions of the method. The chronological account of the step-by-step improvements is only important for readers who want to implement the method themselves, and should therefore be provided as supplementary info.

We agree that there is perhaps too much detail included on the optimization of ENGRAM for a broad audience. To address this, we have: (a) Greatly shortened the description of ENGRAM 1.0 in the main text, and moved all of the corresponding figure panels (as well as the schematic comparison of ENGRAM 1.0 vs. 2.0) to **Supplementary Figure 1**. (b) Consolidated other material from Figures 1 & 2 from the original paper into a single figure (new **Figure 1**). (c) Taken a pass at more generally tightening the Introduction and early sections of the Results. We welcome any further suggestions around further focusing or sharpening our presentation and avoiding unnecessary detail.

7. The Materials and Methods section is far too minimal. Since this manuscript describes an entirely new method, a much more detailed protocol should be provided. I was unable to find the datasets. For example, a list of all tested enhancers and their precise sequences, and the data of Figs 2h and SuppFig 2.

We apologize that the Methods section was not sufficiently detailed, and that some data or information was hard to find. In response to this comment and similar comments from another reviewer, we have sought to substantially expand the Methods to provide greater clarity.

The other data/information was uploaded to github but not submitted with the manuscript, for which we apologize. We have reviewed the data presented in each figure and ensured that it is part of the data release to GEO; the mapping of presented analyses to datasets is now explicitly included in the manuscript.

For a detailed table of all sequences used, including the sequences of all CREs used in the revised manuscript, see **Supplementary Table 2-5**.

Minor points:

8. Figure 1a: The use of FISH as a readout seems too optimistic to me. To my knowledge, DNA FISH still does not work for such single, very short sequences.

The reviewer is correct that no FISH-based readout was demonstrated, and a similar point was raised by Reviewer #1. Rather, we were trying to highlight the potential (and note that probes could be designed targeting the broader target, centered on the 5-6 bp signal-specific insertion), but agree in retrospect that this is not appropriate as we did not actually demonstrate it. Thus, we have also deleted any reference to FISH from

the figure (which was further revised to improve clarity), although we still briefly allude to it in the Discussion as a future possibility.

Figure 1. ENhancer-driven Genomic Recording of transcriptional Activity in Multiplex (ENGRAM). (a) Schematic of ENGRAM. Naturally occurring or designed cis-regulatory elements (CREs) drive activity-dependent transcription of a prime editing guide RNA (pegRNA) encoding a CRE-specific insertion. The CRE could be a natural enhancer sequence (notably tested out of context, as in a conventional or massively parallel reporter assay) or a designed sequence (e.g. spacer-separated repeats of a transcription factor motif). The pegRNA is embedded within a Pol-2 transcript, flanked by two 17bp *csy4* hairpin such that it can be released by Csy4 ribonuclease. The liberated pegRNAs are designed to write a CRE-specific insertional barcode to a common DNA Tape site, which could be an endogenous genomic locus (e.g. the native *HEK3* site) or a synthetic DNA Tape (e.g. exogenously introduced via lentiviral or piggyBac integration). Sequencing of the DNA Tape reveals which CREs were active; if there are multiple ENGRAM reporters in the system, their relative activities can be measured across a pool of cells and/or multiple DNA Tape sites in each cell.

9. Please define term "recorder" as used throughout the text. Is it the vector that drives pegRNA expression, or its combination with the prime editor vector? Or does it also include the DNA tape? Please check that the term is used consistently.

We apologize for any lack of consistency. We have added the following definitions to the Introduction, and have checked that the terms are used consistently in the revision:

“Throughout this manuscript, we use “ENGRAM recorder” to refer to the pegRNA expression cassette, and “ENGRAM recorder system” to refer to the combination of prime editor, pegRNA expression cassette, and DNA Tape. A key attribute of the ENGRAM recorder system is that multiple ENGRAM recorders can concurrently operate, competing to write signal-specific insertions to a shared DNA Tape. ”

10. It was a good idea to reduce background by putting Csy4 expression under control of the same enhancer as the pegRNA. However, this background reduction may fail in multiplexed setting, as the Csy4 expression will be driven by many enhancers. What if one of these enhancers activates at an early time point -- will this mean higher background signals for the recording of the enhancers that activate later (or not at all)? Please comment on this potential problem.

This is an insightful point. Although this was initially a major worry, it has not really materialized as such in practice. In particular, we observe minimal background in the multiplex recorder experiment, in which all three

recorders are integrated at a high MOI (estimated ~15), such that the vast majority of cells (>98%) should have at least one copy of each recorder. If the activation of one ENGRAM recorder drove background from the other recorders, we should see that. However, this is not the case, as crosstalk is similar to background. This is shown in **Figure 3i**, reproduced below.

Figure 3. Multi-channel recording of the intensity and duration of signaling pathway activity. (i) These cells were exposed to all possible on/off combinations of three agonists for 48 hrs, followed by harvesting and sequencing-based quantification of the levels of signal-specific barcodes written to DNA Tape. Colored shapes as in panel a. Concentrations used were 500 ng/ml, 10 ng/ml and 3 μ M for doxycycline, TNF α and CHIR99021, respectively.

The values (%s) underlying this figure are shown in **Reviewer Table 1**. Recording levels for conditions/recorders for which a negative result is expected despite the activation of other ENGRAM recorders (tan colored cells) are not appreciably higher than cells in which the recorders were present but none of the agonists applied (background column; light green cells).

	Background	Dox	TNF α	CHIR	Dox+TNF α	Dox+CHIR	CHIR+TNF α	Dox+CHIR+TNF α
TetOn	0.063	0.766	0.117	0.137	0.601	0.861	0.114	0.814
NFkB	0.100	0.114	1.370	0.201	1.255	0.188	1.017	1.110
Wnt	0.068	0.092	0.188	1.785	0.112	1.570	1.508	1.561

Reviewer Table 1. The values underlying **Figure 3i**. Each number represents the percentage of reads with particular barcodes corresponding to different ligands, and is the average of three replicates.

11. In the multiplexed setting of Figure 2g/h, I understand that the editing of the DNA tape is competitive. In this setting, what would happen if enhancer A is active earlier than enhancer B? Would the editing driven by B erase some of the edits driven by A? Please comment on this.

The nature of the system is such that editing events are irreversible. Once one pegRNA succeeds in editing the DNA Tape, the insertion sequence disrupts the DNA Tape sequence and no further editing will occur (*i.e.* no “over-writing” of past recordings).

Of note, in the context of the combination of ENGRAM and DNA Typewriter, newly presented in the revision, this remains true, except that if enhancer A is active earlier than enhancer B, then it we would expect to observe A→B more often than B→A in adjacent units of multi-unit DNA Typewriter Tape (as the first site is still disrupted, but the type-head is shifted down by one unit). For more details, please see **Figure 4**, **Supplementary Figure 7** and corresponding revisions to text. Alternatively, see our response to your Comment #4 above, or Comment #1 from Reviewer #1.

12. Please clarify in the main text: for the multiplexing with 300 enhancers as in figure 2h, how many cells were needed? How does this compare to the number of cells needed for measuring the activity directly from barcoded RNA (*i.e.* a conventional MPRA)?

We agree that if there is only one recording site per cell and the recording efficiencies are very low, then large numbers of cells and/or reads may be needed to obtain reproducible estimates of relative recording efficiencies. To systematically explore whether we could reduce cellular input requirements by having multiple DNA Tapes per cell, we performed a new experiment in which we integrated synthetic DNA Tape (bearing *HEK3* target) into K562 cells at a high MOI (~20) and repeated the 300-plex enhancer recording experiment (schematic in **Figure 2a**, reproduced below):

Figure 2. (a) Schematic of multiplex recording experiment. A library of ENGRAM reporters bearing 300 CRE fragments, each linked to a unique pegRNA-encoded insertion, was constructed and integrated into PE2(+) K562 cells via piggyBac. CRE activity was then recorded to both endogenous (genomic *HEK3*, n=2-3) or synthetic (*HEK3* locus integrated into the genome via piggyBac, n~20) DNA Tape. For the purposes of benchmarking, the relative activities of CREs were measured either via recording (ENGRAM, DNA-seq of *HEK3*) or reporting (MPRA, RNA-seq of pegRNAs).

Despite it being a relatively stellar site for prime editing⁵⁰, editing rates across the ensemble of randomly integrated synthetic HEK3 sites were about 2-fold lower than that of the endogenous HEK3 site, likely due to chromatin-mediated effects, as we recently documented³² (**Supplementary Figure 3c**, reproduced below). While recovering recorded information from a large number of cells (~300k), our estimates of the relative activity of the ~300 enhancers were highly reproducible between endogenous and synthetic *HEK3* recording sites (**Supplementary Figure 3d**). To assess robustness, we recorded CRE activity from various numbers of input cells. With synthetic DNA Tape (~20 copies per cell), we were able to record the relative activities of the CREs from as few as 12,000 cells (*i.e.* 40 cells per CRE) with reasonable recovery and reproducibility (291 of 300 detected; mean Pearson's correlation of 0.87 between replicates; **Supplementary Figure 4b-c**, reproduced below). By downsampling the number of reads used in association with the 96,000 cell input condition, we found that 250,000 reads were sufficient (278 of 300 detected; mean Pearson's correlation of 0.87 between replicates), although more reads were helpful (**Supplementary Figure 4d-e**, reproduced below). Looking forward, input and sequencing requirements could presumably be reduced by either increasing the number of DNA Tape units in each cell or by further increasing prime editing efficiency.

Supplementary Figure 3. (c) Insertion efficiency of various barcodes at synthetic (1.8%) vs. endogenous (3.1%) *HEK3* loci are highly correlated. Of note, for synthetic *HEK3* sites, the observed efficiencies reflect an average across many genomic contexts. **(d)** Log-scaled insertion proportions for 300 6-mer barcodes were highly correlated between DNA Tape sites located at synthetic vs. endogenous *HEK3* loci.

Supplementary Figure 4. Further benchmarking of ENGRAM. (b-c) Different cell numbers were sampled (6,000, 12,000, 24,000, 48,000, 96,000 cells) prior to measuring ENGRAM recorded activity of 300 CREs, either from endogenous and synthetic DNA Tape, and then recovery **(b)** and reproducibility **(c)** were assessed. **(d-e)** Sequencing data from synthetic DNA Tape and 96,000 cell input condition was downsampled, and then recovery **(d)** and reproducibility **(e)** were assessed.

We would prefer not to make a strong statement in the paper about comparing these numbers to MPRA, especially as what numbers of cells or reads are used there may vary by group and experimental design. However, at least in our hands, the required input to achieve reproducibility with ENGRAM is considerably less than what is typical for us with MPRA, where we are often starting with ~500K-1M cells as input for silica-based column purification of both DNA and RNA. With ENGRAM, we need only purify DNA, which is much more straightforward and consistent, potentially contributing to the reproducibility observed with limited numbers of cells and/or reads as documented above. As prime editing reagents continue to improve, we anticipate that input and sequencing requirements will be reduced even further.

=====
Reviewed by Bas van Steensel. Assignment accepted: 26 Nov 2021; completed: 28 Nov 2021.
It is my standard policy to sign and date **all** of my manuscript reviews, irrespective of my comments and recommendations. All correspondence should go via the editor. PLEASE DO NOT REMOVE THIS NOTE.
=====

REFERENCES

1. Choi, J. *et al.* A time-resolved, multi-symbol molecular recorder via sequential genome editing. *Nature* (2022) doi:10.1038/s41586-022-04922-8.
2. Golic, K. G. & Lindquist, S. The FLP recombinase of yeast catalyzes site-specific recombination in the *Drosophila* genome. *Cell* **59**, (1989).
3. Sauer, B. Functional expression of the cre-lox site-specific recombination system in the yeast *Saccharomyces cerevisiae*. *Mol. Cell. Biol.* **7**, 2087 (1987).
4. Sheth, R. U. & Wang, H. H. DNA-based memory devices for recording cellular events. *Nat. Rev. Genet.* **19**, 718–732 (2018).
5. Kretzschmar, K. & Watt, F. M. Lineage Tracing. *Cell* vol. 148 33–45 Preprint at <https://doi.org/10.1016/j.cell.2012.01.002> (2012).
6. Livet, J. *et al.* Transgenic strategies for combinatorial expression of fluorescent proteins in the nervous system. *Nature* **450**, 56–62 (2007).
7. Lakso, M. *et al.* Targeted oncogene activation by site-specific recombination in transgenic mice. *Proc. Natl. Acad. Sci. U. S. A.* **89**, 6232–6236 (1992).
8. Shipman, S. L., Nivala, J., Macklis, J. D. & Church, G. M. Molecular recordings by directed CRISPR spacer acquisition. *Science* **353**, aaf1175 (2016).
9. Shipman, S. L., Nivala, J., Macklis, J. D. & Church, G. M. CRISPR-Cas encoding of a digital movie into the genomes of a population of living bacteria. *Nature* **547**, 345–349 (2017).
10. Sheth, R. U., Yim, S. S., Wu, F. L. & Wang, H. H. Multiplex recording of cellular events over time on CRISPR biological tape. *Science* **358**, 1457–1461 (2017).
11. Yim, S. S. *et al.* Robust direct digital-to-biological data storage in living cells. *Nat. Chem. Biol.* **17**, 246–253 (2021).
12. Bhattarai-Kline, S. *et al.* Recording gene expression order in DNA by CRISPR addition of retron barcodes. *Nature* **608**, 217–225 (2022).
13. Tang, W. & Liu, D. R. Rewritable multi-event analog recording in bacterial and mammalian cells. *Science*

- 360**, (2018).
14. Farzadfard, F. *et al.* Single-Nucleotide-Resolution Computing and Memory in Living Cells. *Mol. Cell* **75**, (2019).
 15. Knapp, D. J. H. F. *et al.* Decoupling tRNA promoter and processing activities enables specific Pol-II Cas9 guide RNA expression. *Nat. Commun.* **10**, 1490 (2019).
 16. Kim, H. K. *et al.* Predicting the efficiency of prime editing guide RNAs in human cells. *Nat. Biotechnol.* **39**, 198–206 (2021).
 17. Chen, W. *et al.* Massively parallel profiling and predictive modeling of the outcomes of CRISPR/Cas9-mediated double-strand break repair. *Nucleic Acids Res.* **47**, 7989–8003 (2019).
 18. Perli, S. D., Cui, C. H. & Lu, T. K. Continuous genetic recording with self-targeting CRISPR-Cas in human cells. *Science* **353**, (2016).
 19. Kalhor, R., Mali, P. & Church, G. M. Rapidly evolving homing CRISPR barcodes. *Nat. Methods* **14**, 195–200 (2017).
 20. van den Brink, S. C. *et al.* Single-cell and spatial transcriptomics reveal somitogenesis in gastruloids. *Nature* **582**, 405–409 (2020).
 21. Veenvliet, J. V. *et al.* Mouse embryonic stem cells self-organize into trunk-like structures with neural tube and somites. *Science* **370**, (2020).
 22. Hill, A. J. *et al.* On the design of CRISPR-based single-cell molecular screens. *Nat. Methods* **15**, 271–274 (2018).
 23. van den Brink, S. C. *et al.* Symmetry breaking, germ layer specification and axial organisation in aggregates of mouse embryonic stem cells. *Development* **141**, 4231–4242 (2014).
 24. GloResponse™ NF-κB-RE-luc2P HEK293 Cell Line.
<https://www.promega.com/products/luciferase-assays/genetic-reporter-vectors-and-cell-lines/gloresponse-nf-kb-re-luc2p-hek293-cell-line/?catNum=E8520>.
 25. Shukla, R. *et al.* Proinflammatory cytokine TNF-α increases the stability of hepatitis B virus X protein through NF-κB signaling. *Carcinogenesis* **32**, 978–985 (2011).

26. Vince, J. E. *et al.* IAP antagonists target cIAP1 to induce TNF α -dependent apoptosis. *Cell* **131**, 682–693 (2007).
27. Doumpas, N. *et al.* TCF/LEF dependent and independent transcriptional regulation of Wnt/ β -catenin target genes. *EMBO J.* **38**, (2019).
28. Shooshtarizadeh, P. *et al.* Gfi1b regulates the level of Wnt/ β -catenin signaling in hematopoietic stem cells and megakaryocytes. *Nat. Commun.* **10**, 1270 (2019).
29. Choi, J., Chen, W., Liao, H., Li, X. & Shendure, J. A molecular proximity sensor based on an engineered, dual-component guide RNA. *bioRxiv : the preprint server for biology* (2023) doi:10.1101/2023.08.14.553235.
30. Habib, O., Habib, G., Hwang, G.-H. & Bae, S. Comprehensive analysis of prime editing outcomes in human embryonic stem cells. *Nucleic Acids Res.* **50**, 1187–1197 (2022).
31. Doman, J. L. *et al.* Phage-assisted evolution and protein engineering yield compact, efficient prime editors. *Cell* **186**, (2023).
32. Li, X. *et al.* Chromatin context-dependent regulation and epigenetic manipulation of prime editing. *bioRxiv* 2023.04.12.536587 (2023) doi:10.1101/2023.04.12.536587.
33. Farley, E. K. *et al.* Suboptimization of developmental enhancers. *Science* **350**, 325–328 (2015).
34. Kvon, E. Z. *et al.* Genome-scale functional characterization of Drosophila developmental enhancers in vivo. *Nature* vol. 512 91–95 Preprint at <https://doi.org/10.1038/nature13395> (2014).
35. Pennacchio, L. A. *et al.* In vivo enhancer analysis of human conserved non-coding sequences. *Nature* **444**, 499–502 (2006).
36. de Almeida, B. P. *et al.* Targeted design of synthetic enhancers for selected tissues in the Drosophila embryo. *Nature* 1–2 (2023).
37. Taskiran, I. I. *et al.* Cell type directed design of synthetic enhancers. *Nature* 1–3 (2023).
38. Klein, J. C. *et al.* A systematic evaluation of the design and context dependencies of massively parallel reporter assays. *Nat. Methods* **17**, 1083–1091 (2020).
39. Muerdter, F. *et al.* Resolving systematic errors in widely used enhancer activity assays in human cells. *Nat.*

- Methods* **15**, 141–149 (2018).
40. pGL4.49[luc2P/TCF-LEF/Hygro] Vector Protocol.
<https://www.promega.com/resources/protocols/product-information-sheets/a/pgl4-49-vector-protocol/>.
 41. GloResponse™ NF-κB-RE-luc2P HEK293 Cell Line.
<https://www.promega.com/products/luciferase-assays/genetic-reporter-vectors-and-cell-lines/gloresponse-nf-kb-re-luc2p-hek293-cell-line/?catNum=E8520>.
 42. Pinglay, S. *et al.* Synthetic regulatory reconstitution reveals principles of mammalian cluster regulation. *Science* **377**, eabk2820 (2022).
 43. Mitra, R. D., Shendure, J., Olejnik, J., None, E.-K.-O. & Church, G. M. Fluorescent in situ sequencing on polymerase colonies. *Anal. Biochem.* **320**, (2003).
 44. Ng, S. B. *et al.* Targeted capture and massively parallel sequencing of 12 human exomes. *Nature* **461**, 272–276 (2009).
 45. Smith, L. M. *et al.* Fluorescence detection in automated DNA sequence analysis. *Nature* **321**, (1986).
 46. Chalfie, M., Tu, Y., Euskirchen, G., Ward, W. W. & Prasher, D. C. Green fluorescent protein as a marker for gene expression. *Science* **263**, (1994).
 47. Gossen, M. *et al.* Transcriptional Activation by Tetracyclines in Mammalian Cells. *Science* vol. 268 1766–1769 Preprint at <https://doi.org/10.1126/science.7792603> (1995).
 48. Cusanovich, D. A. *et al.* Multiplex single cell profiling of chromatin accessibility by combinatorial cellular indexing. *Science* **348**, (2015).
 49. Cao, J. *et al.* Comprehensive single-cell transcriptional profiling of a multicellular organism. *Science* **357**, (2017).
 50. Anzalone, A. V. *et al.* Search-and-replace genome editing without double-strand breaks or donor DNA. *Nature* **576**, 149–157 (2019).

Reviewer Reports on the First Revision:

Referees' comments:

Referee #1 (Remarks to the Author):

Chen and Choi et al. developed ENGRAM, a system that enables the recording of exogenous enhancer sequence activities into a target DNA using Prime Editing (PE). They first demonstrated the concept and optimized the circuit, followed by demonstrating the recording of genome-wide enhancer activities into DNA tapes of different cell populations. They then showed the recording of synthetic signals and their orders provided to different cell populations. Finally, they applied the system to the recording of selected synthetic enhancer activities in a mouse gastruloid model, which is impressive. As a long-term follower of the authors' works, although we have provided somewhat bitter comments in the initial round, I highly acknowledge the over one year of hard work done by the authors and enjoyed reading the manuscript. Because they added several new major data sets combining ENGRAM and their recent DNA Typewriter Tape (DTT) method, we have more comments, which I hope will help improve the manuscript. For the sections they improved from the initial manuscript, I will try to minimize my comments.

Major comments:

1. First, although I sincerely appreciate the work, I would suggest further toning down the claim that sounds like they have demonstrated ENGRAM (and DTT) for observations that cannot be achieved with the current method. For example, Figure 4 is still a toy model where the observers had all the signal input pattern information from the beginning. In Figure 5, one could potentially perform time-course RNA-seq to acquire similar information. Because the paper is already of high quality, I would suggest clarifying at the beginning of the paper that the demonstrations in this paper are initial proofs-of-concept at the outset so as not to overly excite the biology readers about the immediate application of the proposed method. The field is exciting, but it requires significant further investments.

2. "We speculate that this background editing is due to bacterial origin-of-replication-driven transcription on plasmids^{28,29} prior to integration, rather than CRE/minP-driven transcription from integrated recorders.": Sorry, I probably overlooked this statement in the initial submission, but this could be assessed by transfecting a linear DNA without the bacterial origin. However, this statement might be excessive for the paper as is and could be omitted or moved to supplementary material.

3. "However, a comparison of active CREs vs. a minimal promoter alone found only a modest difference (2.1-fold; Supplementary Figure 1d), possibly due to the accumulation of background edits during transfection of the constitutively Csy4-expressing cell line." Perhaps an independently expressed Csy4 could be introduced subsequent to providing pegRNAs?

4. "We speculate that the csy4 hairpin might already serve a similar role to tevoPreQ1 in protecting pegRNAs from degradation and/or that the presence of both the tevoPreQ1 and csy4 hairpins might disrupt RNA folding. " Were there any cases of the integration of the leftover Csy4 hairpin in PE-edited cells?"

5. I really like Figure 1h.

6. I'm not sure why the authors transiently transfected PE + ENGRAM recorders in K562, but then chose to integrate the prime editor and subsequently transiently transfect ENGRAM recorders in mESCs. Although they observe higher efficiencies with PEmax, which was likely the purpose of this experiment, it's not really a fair comparison between the cell lines. They should either uniformly transiently transfect the same components (ENGRAM recorders + PE) across both cell types, or consistently integrate PE and then transiently transfect ENGRAM recorders (as they did in mESCs).

7. "To assess whether ENGRAM can record CRE activity in a quantitative manner, we first evaluated a pair of 170-bp CRE fragments previously shown to have either high vs. low enhancing activity in K562 cells when positioned upstream of a minimal promoter²⁹, together with minimal-promoter-only and no-promoter controls, and with each construct linked to two distinct 5-bp insertions (Supplementary Figure 3a).": Could the authors provide any insights on the consistency between the two different tags assigned to each promoter construct?

8. "Furthermore, we observed a strong correlation between directly measured (MPRA, RNA) vs. recorded (ENGRAM, DNA) activities (Figure 2c; Supplementary Figure 3g).": After reviewing Figure 1h, I am curious whether there would be a better correlation with MPRA if the ENGRAM recording result were corrected using the MFE-based model with RNA secondary structure prediction.

9. Figure 2g is also beautiful!

10. Figure 4b-d: I previously provided similar comments in the first manuscript, but I would appreciate seeing the absolute fractions of reads supporting Tet-On>Wnt and Wnt>Tet-On signals, rather than their relative ratios. Also, the authors stated that the combination of ENGRAM and DTT could address the issue where ENGRAM, as a solo system, cannot record the duration of a signal. Have they now observed that the signal duration is captured by the number of signal sequences introduced to the DTT array? For example, does the maximum number of the same signal sequences introduced into a single sequencing read accurately reflect the signal duration?

11. "These included the proportions of the two symbols at each of the five DTT sites (2 symbols * 5 positions) and the proportions of each possible bigram (Tet-On→Tet-On, Tet-On→Wnt, Wnt→Tet-On, Wnt→Wnt) at pairs of adjacent sites (4 bigrams * 4 position-pairs).": Why did the authors choose to encode and compress their DTT sequencing reads in this manner? Is it not feasible to just use sequencing read proportions of raw DTT sequence patterns? Furthermore, can they predict input signal patterns from the sequencing results?

12. Supplementary Figure 7b-d: What explains the observation that the editing frequencies sometimes exceed 50%? In contrast, the editing efficiencies shown in Figure 3 were only up to a few percent. Why are the editing efficiencies in this instance so high? Could these efficiencies be relative to a specific reference or baseline?

13. Figure 5: This demonstration is impressive, and I can see a lot of effort has been invested here.

However, I wonder why the experiment wasn't designed such that the recording result can be read out by scRNA-seq. Could the authors at least perform scRNA-seq of different stages and compare their pseudobulk results to the recording results? Additionally, is it possible to project the recording results back onto the scRNA-seq data to identify which cell types were predominant in providing those recording results? I find these analyses to be extremely exciting and believe they would not be overly challenging for this team. However, I already appreciate the extensive amount of work undertaken, so I won't insist on this.

14. Rebuttal "We relied on an ensemble of cells, rather than a single cell, to extract each temporal signal pattern, and we have yet to develop algorithms or models that would allow one to directly infer signaling histories from a newly observed, complex patterns (as opposed to simply showing after the fact that the patterns are distinct, as we do in Figure 4e).": It's reasonable to demonstrate not only the ratio metric but also what fractions of the reads support the expected orders.

15. Rebuttal "For example, if dozens of homing gRNAs were used for different signals, each writing back to the genomic locations from which they were expressed, recovering recorded events from so many such sites by methods such as single-cell RNA-seq (sc-RNA-seq) would be very challenging.": I'm ok with this statement, but wouldn't the use of scRNA-seq be particularly advantageous, as evidenced in their previous scGESTALT demonstration? It's both aesthetically pleasing and highly scalable, to the extent that the minimal amount of information recorded per site becomes negligible.

16. Supplementary Table 1 "signal specific": If Base Editors (BE) are categorized as pseudorandom, Prime Editors (PE) should also be considered pseudorandom, given that they are not perfect. The authors might want to categorize them based on their degrees of pseudorandomness, where even qualitative metrics would suffice.

17. Supplementary Table 1 "Moderate (system complexity grows exponentially with number of biological signals, because guides/targets must be preprogrammed for each order)": I might challenge the categorization of CRISPR as "poor." Our group, which regularly tests BE-based CRISPR cell lineage tracing, has recently realized that a major drawback of the BE system is its tendency to generate identical mutation patterns. In contrast, the editing patterns produced by the wildtype Cas9 have a higher information content per site and are irreversible, a feature that can be advantageous in certain contexts.

Minor comments:

18. "(or combinations of fluorescent reporters, in the case of Brainbow)⁵⁻⁷": I would suggest adding the polylox lineage tracing (or clone labeling) method here.

19. "The enzymatic event is irreversible": This actually varies depending on the circuit construction. For instance, a single flipping unit enclosed by two loxP sites in reversible orientations can indeed be reversible.

20. "We tested PE2 vs. PEmax28 in the context of ENGRAM in both K562 cells and mouse embryonic stem cells (mESCs). In K562 cells, constitutively active ENGRAM recorders bearing 5N insertions, together with PE2 or PEmax, were transiently transfected in triplicate, targeting the endogenous

HEK3 locus. In mESCs, we first integrated a synthetic DNA Tape (bearing a copy of the HEK3 target site) together with either PE2 or PEmax via piggyBac. We then transiently transfected a ENGRAM recorder encoding a 5-bp insertion to be written to HEK3, again in triplicate. Both K562 and mESCs were harvested 3 days post-transfection, and the insertions at endogenous or synthetic DNA Tape quantified.”: I really appreciate these careful analyses in responding to the reviewers' comments, but I feel that they could be more appropriately moved to a figure legend or supplementary text. Also, it should be “an ENGRAM”?

21. “Finally, we tested whether tRNAs15 could be used as an alternative to csy4 to facilitate pegRNA release. However, upon replacing the csy4 hairpins with tRNAs, we observed a nearly complete ablation of recording activity (Supplementary Figure 1i). Overall, these experiments suggest that future ENGRAM experiments would benefit from switching to more active prime editors such as PEmax, but not from switching to epegRNAs nor tRNAs.”: This could also be moved to a supplementary text (I know I might be the one who provided all these annoying comments!).

Referee #2 (Remarks to the Author):

The revised manuscript incorporates 2+ years' worth of work by the Shendure lab and the team is to be commended for creativity and perseverance. While we are in 2024 vs 2021, it still remains the case that a lot more about human gene regulation is quite murky, and the structure-function of cis-regulatory elements remains an area that is murkiest. So at a high level, the field needs as many robust, scalable ways to address this issue as possible, and the work in the revised manuscript is relevant and interesting.

Two major concerns were raised in the initial review and the authors' response to them is discussed below.

The first concern related to the whether the new method has an operationally useful linearity and dynamic range (ie whether regulatory elements of low, medium, and high activity were consistently recorded as ranking in that order by the new assay). The original manuscript represented these data in a log₂/log₂ scale and thus made it challenging to assess this. The revised manuscript provides the reanalysis requested and the answer to the question appears to be a qualified “yes”: fig 2d demonstrates that the relationship between MPRA activity and ENGRAM activity of randomly sampled CREs is a decent one. There appear to be only 2 CREs that are inactive by ENGRAM and active by MPRA, and no CREs that exhibit the opposite behavior. Fig 2e is an interesting result: the highest-active CREs are the highest-correlated between the two assays, while the correlation becomes weaker (but still clearly there) as enhancer activity drops. Supplementary fig 4a shows more or less the same data but, importantly, adds CRE-by-CRE comparisons. Taken together there is good news – ENGRAM does rank CREs – and other news: we won't know the ground truth until we have genome-edited a whole bunch of CREs assayed in their native genomic location and looked at effects on target gene control. Overall the reanalysis fully addresses the concern raised.

The second concern related to whether the ENGRAM construct could be used to detect activity at endogenous genes. Such reporter tagging has been done at native human genes with fluorescent constructs since 2010 (21297641), and as mentioned in the initial review, Manu Leonetti at the Biohub has tagged 1000s of human genes this way (35271311) and all their gRNAs that worked for this massive effort could be repurposed wholesale for such an experiment. The reply to reviewer

comments the authors provide presents an extensive and thoughtful discussion of this issue, two salient points of which are addressed below.

[REDACTED]

The second issue is the authors – in another burst of trademark Shendure-lab creativity – do a related experiment to address this point. In brief, a promotorless reporter pegRNA is integrated using piggyBac, an ingenious method to map integration sites is then used to characterize the genome-wide insertion spectrum of the reporter constructs, and then the correlation between endogenous gene transcription and ENGRAM activity is determined. Two findings emerge. The first finding is the transcriptional activity recording is dominated by reporter constructs that landed into rDNA. Of note – and this is, in the eye of this reviewer a fascinating result – it is not the case that all these new “pol1 reporter constructs” are comparably active: in fact, as shown in reviewer figure 1h, these show a very wide range of “writing/recording activities.” This means that individual rDNA units are unequal in terms of transcriptional output, or specific insertion sites within the rDNA are “accommodating” the newly arrived reporter into rRNA with different efficiencies. One imagine that folks studying rDNA transcription control would find this of formidable interest (there are some data on this in Arabidopsis <https://www.nature.com/articles/s41467-020-20728-6>). The second finding is that the remaining signal from reporter constructs does not correlate with landing site transcriptional activity (reviewer figure 1h and 1k). These important data should be reviewed by insertion class, and here, the data on recording signal from intron-resident reporters are noteworthy. One can argue that the exon insertions are uncorrelated to native gene activity because the reporter surely brings in a premature stop codon in itself or downstream and native human genes are differentially sensitive to NMD. One can also argue that promoter insertions are hard to interpret because of the location of the native gene TSS (and coding region in exon 1) relative to where the cassette landed. Why would the intron-resident cassettes be so poorly correlated in their recording activity relative to what the native gene does transcription activity-wise? The simplest possibility is the cassette has a cryptic splice acceptor (or donor) that perturbs the native splicing pattern, but differently so dependent on location of insertion site in the intron and the local landscape of splicing enhancer and repressor signals (as in the SMN2 gene that is targeted by spinraza). The editing literature has a classic result from Wen Xue (28615073) that the cell ignores the editing-driven knockout of an exon by just skipping it. The above is a hypothesis that, frankly, does not deserve to be tested because the purpose here is to measure gene activity in native contexts, and as the authors’ careful analysis shows, the exact construct and approach they used to do that fails to meet that expectation. In the mind of this reviewer a brief reference to this result – as “unpublished observations” – would be warranted to prevent others from following this path. While the result is a negative one, the authors are to be commended for doing this creative experiment and thus refocusing future effort on minimal-engram-cassette integration into native genes or isogenic landing pads. The rest of the comments raised in initial review were minor and are adequately addressed in the revision.

// Fyodor Urnov, Innovative Genomics Institute, UC Berkeley //

Referee #3 (Remarks to the Author):

This revised manuscript includes two major additions. Figure 4 illustrates that ENGRAM can resolve temporal patterns of activity. This is a technically impressive and conceptually exciting addition. But the results also illustrate some practical limitations of the approach (inability to discriminate pulse, layered, serial), and the results remain at the level of proof-of-principle (with strong stimuli given to cells in a dish) rather than an illustration of real new biology that could not be discovered otherwise. In real biology, where signals are often weaker and more complex, it will still be a major challenge to apply this ENGRAM approach successfully.

A second major addition in the revised manuscript is the multiplexed probing of 98 different TF motifs. While I appreciate that this is meant to further illustrate that "biology" can be uncovered with ENGRAM, I am afraid that the authors have opened a can of worms. For several reasons, I think these results could be dominated by artefacts and noise (comment a-e) while there is a much simpler alternative (f):

a. It is far from trivial to design TF reporters. Full disclosure: my lab has worked on the design of such reporters over the past four years (manuscript in preparation). We have learned that every TF has its own requirements when it comes to the spacing of the motifs; the spacer sequence between the motifs; the distance of the motifs to the core promoter; and the identity of the core promoter. When these parameters are not carefully tuned for each TF motif individually, there will be lots of false-negatives and false-positives, and certainly one cannot trust quantitative differences between reporters within and between cell lines.

b. The TF ENGRAM results are not validated at all. The authors cherry-pick a few TF motifs that "make sense" (and, to their credit, also that don't make sense, e.g. HoxB9 in mESCs), but an objective criterium to judge the results is lacking. For the gastruloids, we are simply provided with a list of TFs that are detected as increased or decreased over time, but validation is completely lacking. At this stage, it cannot be ruled out that the majority of these TF recorder results are artefacts.

c. Figure S5d shows correlation with TF expression levels, but this shows only some the small number of TFs detected by ENGRAM and not the large number of TFs that ENGRAM did not detect. There are some odd discrepancies between Fig 2g and S5d: In S5d, ZNF449 is missing and FOSL1 and JUNB are included while they (AP1 recorders) are not in 2g.

d. For the gastruloids, it is mentioned that the overall recording efficiency decreases over time (Fig S8c). How does this affect the interpretation of the time course data in Fig 5e-h? Were these data somehow normalized for this decline? Is this overall efficiency perhaps dominated by a small subset of TFs? Wouldn't it be necessary to include some kind of internal normalization standard that is not dependent on any TF activity?

e. The experiments were done in polyclonal cell pools. It seems that the authors assume that each TF recorder is equally represented in these pools. That might seem reasonable, but in practice this is a risky assumption. Due to the (often mutagenic) integrations (the cells carry an estimated 25 insertions) and other clonal effects, there is a serious risk of clonal drift. Please see PMID 35173762, figure 4. Substantial drift can occur within a week! My lab has experienced this in multiple TRIP projects, with in some instances a few clones rapidly taking over the entire pool. It is thus critical that the relative representation of each recorder is determined in parallel, and used to normalize the data.

f. Moreover, it should be considered that there is a widely used alternative to estimate TF activities in complex tissues: single-cell ATAC-seq. The Shendure lab has published extensively with this technology, in particular to infer TF activities in various complex systems such as developing embryos. scATAC-seq also allows clustering of the data by cell type, while ENGRAM currently detects 'averaged' signals across a mix of cell types. Considering the above uncertainties about the quality of the TF recorder data, what then is the advantage of ENGRAM for monitoring TFs? scATAC-seq is commercially available and much simpler to set up than multiplexed ENGRAM. No doubt, scATAC-seq has its limitations, but from the current manuscript it remains entirely unclear whether ENGRAM will do a better job detecting TF activities - so far it seems to do worse.

In conclusion, I reiterate that it is an exceptionally cleverly and thoroughly designed technology that will inspire scientists in the field, and I fully appreciate how much additional work went into this revised manuscript. But I regret to say that the new TF recorder data are too preliminary and do not make a convincing case that ENGRAM will be a new workhorse in the field to probe TF activities. The ability to do temporal recording is demonstrated in a truly elegant proof-of-principle experiment, but it remains to be seen whether this will be more practical than (say) using fluorescent reporters or ATAC-seq.

Other points:

In each figure legend: please clarify whether the replicates were *independent* biological replicates (done on different days), or technical replicates (done in parallel dishes or tubes on the same day). In Fig 4bcd the reproducibility seems exceptionally high for independent biological replicates.

Raw sequencing reads were made available through SRA, but none of the processed data were made available as supplementary data. This makes it very difficult to thoroughly review the results, and readers should be given access to the processed data (in particular the TF recorder data). These data could have easily been provided in a few Excel sheets.

What is the source of the expression data in Supp5d? Please add to the legend.

Fig S5b seems a bit misleading: this correlation depends on TF reporter activity but also on barcode insertion bias (Figure 1f). The stronger the bias, the better the correlation will be... A correction for this bias seems necessary.

===== Reviewed by Bas van Steensel. Assignment accepted: 4 Jan 2024; completed: 8 Jan 2024. It is my standard policy to sign and date *all* of my manuscript reviews, irrespective of my comments and recommendations. All correspondence should go via the editor. PLEASE DO NOT REMOVE THIS NOTE. =====

Response to Reviewers

We thank the three reviewers for their constructive feedback on the revised manuscript. In this point-by-point response that accompanies our further revision, the new reviewer comments are replicated in full in **blue text**, while our responses are inline in **black text**.

Referee #1 (Remarks to the Author):

Chen and Choi et al. developed ENGRAM, a system that enables the recording of exogenous enhancer sequence activities into a target DNA using Prime Editing (PE). They first demonstrated the concept and optimized the circuit, followed by demonstrating the recording of genome-wide enhancer activities into DNA tapes of different cell populations. They then showed the recording of synthetic signals and their orders provided to different cell populations. Finally, they applied the system to the recording of selected synthetic enhancer activities in a mouse gastruloid model, which is impressive. As a long-term follower of the authors' works, although we have provided somewhat bitter comments in the initial round, I highly acknowledge the over one year of hard work done by the authors and enjoyed reading the manuscript. Because they added several new major data sets combining ENGRAM and their recent DNA Typewriter Tape (DTT) method, we have more comments, which I hope will help improve the manuscript. For the sections they improved from the initial manuscript, I will try to minimize my comments.

We thank the reviewer for their appreciation of the additions and improvements in the first revision.

Major comments:

1. First, although I sincerely appreciate the work, I would suggest further toning down the claim that sounds like they have demonstrated ENGRAM (and DTT) for observations that cannot be achieved with the current method. For example, Figure 4 is still a toy model where the observers had all the signal input pattern information from the beginning. In Figure 5, one could potentially perform time-course RNA-seq to acquire similar information. Because the paper is already of high quality, I would suggest clarifying at the beginning of the paper that the demonstrations in this paper are initial proofs-of-concept at the outset so as not to overly excite the biology readers about the immediate application of the proposed method. The field is exciting, but it requires significant further investments.

R1.1: Thank you for this comment, and we agree. In response, we added new sentences or clauses at the outset that explicitly note emphasize that these are proof-of-concept experiments and that more work is needed:

Abstract: "By coupling ENGRAM to sequential genome editing via DNA Typewriter¹, we show we can stably record information about the temporal dynamics of two orthogonal signal transduction pathways to genomic DNA. Finally, we apply ENGRAM to integratively record the transient activity of nearly 100 transcription factor (TF) consensus motifs across daily windows spanning the differentiation of mouse embryonic stem cells (mESCs) into gastruloids, an *in vitro* model of early mammalian development. **Although these are proof-of-concept experiments and much work remains to fully realize the potential of this paradigm**, we envision that multiplex genomic recording of biological signals has broad potential for application in functional genomics, developmental biology and neuroscience."

Figure 4: As a proof-of-concept evaluation of whether ENGRAM and DNA Typewriter could be combined to record the temporal dynamics of two orthogonal signal transduction pathways, we designed and cloned Tet-On and Wnt ENGRAM recorders encoding pegRNAs that targeted DTT (**Supplementary Figure 7a**).

Figure 5: “To evaluate this at the proof-of-concept level, we constructed a polyclonal mESC line bearing all components of the ENGRAM recorder system, including doxycycline-inducible PEmax, the library of ENGRAM recorders driven by the 98 synthetic CREs described above (each homotypically composed of 6 repeats of one of 98 TF motifs; **Figure 2f**; **Supplementary Table 3**), and synthetic DNA Tape (*HEK3*) (**Figure 5a**).”

2. “We speculate that this background editing is due to bacterial origin-of-replication-driven transcription on plasmids^{28,29} prior to integration, rather than CRE/minP-driven transcription from integrated recorders.”: Sorry, I probably overlooked this statement in the initial submission, but this could be assessed by transfecting a linear DNA without the bacterial origin. However, this statement might be excessive for the paper as is and could be omitted or moved to supplementary material.

R1.2: This is a fair point. We opted to not do the experiment, as hinted by the reviewer might be OK. Our rationale is that this was early in the project, and we now integrate ENGRAM recorders prior to integrating either the DNA Tape or the Prime Editor, such that the speculated mode of background accumulation is entirely circumvented. Especially given that, we agree the statement is distracting. We have moved it to new **Supplementary Note 1**, where it is characterized as “speculation”, still available for technical aficionados but without distracting the general reader (see also **R1.3**, **R1.20**, **R1.21**).

3. “However, a comparison of active CREs vs. a minimal promoter alone found only a modest difference (2.1-fold; **Supplementary Figure 1d**), possibly due to the accumulation of background edits during transfection of the constitutively *Csy4*-expressing cell line.” Perhaps an independently expressed *Csy4* could be introduced subsequent to providing pegRNAs?

R1.3: We agree that this would also likely solve the issue, and indeed the suggestion is a different form of the way that we did solve it (*i.e.* by changing the order in which components were introduced). As articulated in **R1.2** above, we opted not to do the experiment because this mode of background accumulation is not relevant to the updated state-of-the-art for ENGRAM. Also, we are now further downplaying ENGRAM 1.0 and these optimization experiments in the main text narrative, and they are solely described in new **Supplementary Note 1** (see also **R1.2**, **R1.20**, **R1.21**).

4. “We speculate that the *csy4* hairpin might already serve a similar role to *tevoPreQ1* in protecting pegRNAs from degradation and/or that the presence of both the *tevoPreQ1* and *csy4* hairpins might disrupt RNA folding. “ Were there any cases of the integration of the leftover *Csy4* hairpin in PE-edited cells?

R1.4: This is an interesting question. Given that a native 3' end of the nicked genomic DNA strand primes the RT reaction, and not the 3' end of the pegRNA (where *evoPreQ1* pseudoknot or *csy4* hairpins are appended), we believe that such a byproduct is unlikely. Although our original scripts would not have looked for this, we checked the raw sequencing data and did not observe any instances for the *csy4* hairpin being integrated at the target site. We note that the original Nelson *et al.* report did not mention detecting *tevoPreQ1* sequences at integration sites, although we are not sure if they checked.

5. I really like **Figure 1h**.

R1.5: Thank you!

6. I'm not sure why the authors transiently transfected PE + ENGRAM recorders in K562, but then chose to integrate the prime editor and subsequently transiently transfect ENGRAM recorders in mESCs. Although they observe higher efficiencies with PEmax, which was likely the purpose of this experiment, it's not really a fair comparison between the cell lines. They should either uniformly transiently transfect the same components (ENGRAM recorders + PE) across both cell types, or consistently integrate PE and then transiently transfect ENGRAM recorders (as they did in mESCs).

R1.6: Thank you for catching this, and you are correct that it is not an apples-to-apples comparison. The reason for the change is that these experiments were conceived and executed at quite different parts of the project. The K562 experiment was performed early on, primarily to get a rough handle on whether PEmax would be appreciably more efficient than PE, and thus the concurrent transient transfection. The mESC experiments were performed much later, and because of our intent to construct an mESC line that could be differentiated into gastruloids, we chose to integrate the PE or PEmax prior to ENGRAM recorder transient transfection. Because as the reviewer points out this is an apples-to-oranges comparison (not only transient vs. integrated, but also human vs. mouse cell type and endogenous vs. synthetic DNA Tape), we now separate our descriptions of the K562 and mESC experiments, and explicitly warn that the difference should not be overinterpreted. This is possible because these experiments are now described in **Supplementary Note 1** rather than the main text, such that we have more room. The relevant passages now read as follows:

“We initially tested PE2 vs. PEmax¹ in K562 cells. In this experiment, constitutively active ENGRAM recorders bearing 5N insertions, together with PE2 or PEmax constructs, were transiently transfected in triplicate. Cells were harvested 3 days post-transfection, and insertions at the endogenous *HEK3* locus quantified. We observed a 1.7-fold greater editing efficiency with PEmax compared to PE2 (**Supplementary Note Figure 1g**), consistent with PEmax's original description¹.

In separate experiments conducted later in the project, we tested PE2 vs. PEmax in mouse embryonic stem cells (mESCs). In contrast with the previous experiment, we first leveraged the piggyBac system to genomically integrate both a synthetic *HEK3* target site to serve as DNA Tape (because the endogenous human *HEK3* site is not present in the mouse genome) as well as constructs bearing constitutively expressed PE2 or PEmax. To the resulting polyclonal mESC cell lines, we transiently transfected constitutively active ENGRAM recorders bearing 5N insertions. Cells were harvested 3 days post-transfection, and insertions at synthetic *HEK3* target sites quantified. We observed a 9.3-fold increase in the editing efficiency with PEmax over PE2 in these mESCs, substantially higher than the 1.7-fold difference observed for PEmax vs. PE2 in K562s (**Supplementary Note Figure 1g**). However, because there are additional contrasts between these experiments (e.g. transient vs. integrated prime editor; endogenous vs. synthetic *HEK3* site; human vs. mouse translational machinery intersecting with non-codon-optimized [PE2] or human-codon-optimized [PEmax] RT domain), we cannot draw clear conclusions about which factor(s) are explanatory.”

7. “To assess whether ENGRAM can record CRE activity in a quantitative manner, we first evaluated a pair of 170-bp CRE fragments previously shown to have either high vs. low enhancing activity in K562 cells when positioned upstream of a minimal promoter²⁹, together with minimal-promoter-only and no-promoter controls, and with each construct linked to two distinct 5-bp insertions (Supplementary Figure 3a).”: Could the authors provide any insights on the consistency between the two different tags assigned to each promoter construct?

R1.7: Thank you for this suggestion. It was straightforward to implement simply by splitting out results derived from the two different tags into separate bars for each of the four conditions represented in **Supplementary Figure 3b**, as reproduced below. We continue to report the aggregate fold-differences and p-values.

Supplementary Figure 3. Benchmarking of ENGRAM against reporter assays. (a) ENGRAM recorders with highly vs. lowly active CRE fragments (as previously measured via MPRA) upstream of a minP, together with minP-only and promoter-less constructs, were cloned, each driving expression of two distinct pegRNA-encoded barcodes. (b) Barplot showing the editing efficiency of individual barcodes associated with each of the eight members of the CRE library (4 architectures x 2 barcodes each). Fold differences were calculated by first summing the counts for the pair of barcodes associated with each architecture, and then calculating the ratio between pairs of architectures. Barcodes corresponding to the highly active CRE were 41.3-fold, 23.6-fold, and 15.1-fold more abundant than barcodes corresponding to promoter-less, minP-only or lowly active CRE controls, respectively. Error bars correspond to standard deviations across 3 transfection replicates. P-values were obtained using the two-tailed Student's t-test.

8. “Furthermore, we observed a strong correlation between directly measured (MPRA, RNA) vs. recorded (ENGRAM, DNA) activities (Figure 2c; Supplementary Figure 3g).”: After reviewing Figure 1h, I am curious whether there would be a better correlation with MPRA if the ENGRAM recording result were corrected using the MFE-based model with RNA secondary structure prediction.

R1.8: This is a nice suggestion. We normalized the insertion proportion recorded by ENGRAM for this experiment (x-axis of Fig. 2c) by MFE. However, this did not appreciably change the correlation (Reviewer Fig. 1), probably because editing scores only vary over a much more modest range (2^4 or 16-fold dynamic range; see x-axis of Supplementary Fig. 2g) than the transcriptional activities of these 300 CRE fragments (2^{12} or 4096-fold dynamic range; see x-axis of Fig. 2c).

Reviewer Figure 1. Correlation between MPRA and ENGRAM. (Left) Values correspond to the proportion of each barcode read out from the synthetic DNA Tape (ENGRAM) or from the pegRNAs (MPRA), out of the total. The log-scaled proportions of ENGRAM events recorded to DNA were highly correlated with log-scaled proportions of barcodes measured directly from RNA. (Right) Same as (Left), except that the log-scaled proportions of ENGRAM events are normalized by MFE calculated by RNA secondary structure prediction.

Although adding a figure panel felt distracting given that the correlation did not appreciably change, we added a sentence to the **Fig. 2c** legend, which now reads:

“Values correspond to the proportion of each barcode read out from the synthetic DNA Tape (ENGRAM) or from the pegRNAs (MPRA), out of the total. The log-scaled proportions of ENGRAM events recorded to DNA were highly correlated with log-scaled proportions of barcodes measured directly from RNA. Correcting ENGRAM recording proportions by MFE of the corresponding pegRNAs did not appreciably change the correlation (Pearson’s correlation of 0.860 vs. 0.889 with vs. without MFE correction).”

9. Figure 2g is also beautiful!

R1.9: Thank you!

10. Figure 4b-d: I previously provided similar comments in the first manuscript, but I would appreciate seeing the absolute fractions of reads supporting Tet-On>Wnt and Wnt>Tet-On signals, rather than their relative ratios. Also, the authors stated that the combination of ENGRAM and DTT could address the issue where ENGRAM, as a solo system, cannot record the duration of a signal. Have they now observed that the signal duration is captured by the number of signal sequences introduced to the DTT array? For example, does the maximum number of the same signal sequences introduced into a single sequencing read accurately reflect the signal duration?

R1.10: These are great points, which we agree are of general interest and should be elevated.

To address the first question, we note that although these absolute fractions were available in the revision as processed files in github, they were not accessible to the general reader. We therefore added **Supplementary Figure 7f-h**, reproduced below, which shows the absolute fractions for these ENGRAM/DTT experiments.

To address the second question, we added **Supplementary Figure 7i**, which summarizes the absolute fractions of “homopolymeric” runs of symbols in DTT as a function of signal duration. Focusing on symbols corresponding to the first signal applied in Serial programs, we calculated the log-scaled proportion of reads containing 1-4 consecutive, identical symbols (“5-in-a-row” homopolymeric instance counts were near or at zero, thus excluded). As predicted, we observe clear relationship between signal duration and the length of homopolymeric runs in DTT, *e.g.* with the proportion of “4-in-a-row” homopolymeric instances rising 31.3-fold or 5.8-fold between 1-day and 3-day signal duration for Serial programs starting with Wnt or Tet-On, respectively.

Supplementary Figure 7. Combining ENGRAM and DNA Typewriter to record the temporal dynamics of biological signals. (f-h) Modified ENGRAM recorders and 5-unit DTT were sequentially integrated to PEmax(+) HEK293T cells. We designed and tested serial (f) and layered (g) programs in which these cells were exposed to different patterns of 100 ng/ml doxycycline or 3 μ M CHIR-99021 (left columns) across a total of 36 cell populations (2 patterns * 3 intervals * 2 possible orders * 3 integration replicates). Harvesting, amplifying and sequencing the DTT region after 6 days, we show here the absolute fractions of [Tet-On→Wnt] and [Wnt→Tet-On] bigrams at Sites 1-2 (right columns). Error bars correspond to standard deviations across 3 integration replicates. See also **Figure 4b-c**. (h) Same as panels f-g, but for pulse programs in which these cells were exposed to 500 ng/ml doxycycline for 24 hrs against the background of continuous stimulation with 3 μ M CHIR-99021. A total of 9 cell populations are represented (3 pulse timings * 3 integration replicates). See also **Figure 4d**. (i) Longer signal durations are associated with longer homopolymeric runs of the signal-specific symbol. Focusing on symbols corresponding to the first signal applied in Serial programs, we calculated the proportion of homopolymeric runs of various lengths, *i.e.* consecutive, identical symbols beginning at the first position in the DTT. Log-scaled proportions for homopolymeric runs of 1-4 are plotted for signal durations of 1, 2 or 3 days. We did not observe any “5-in-a-row” homopolymeric instances in the data. Error bars correspond to standard deviations across 3 integration replicates.

11. “These included the proportions of the two symbols at each of the five DTT sites (2 symbols * 5 positions) and the proportions of each possible bigram (Tet-On→Tet-On, Tet-On→Wnt, Wnt→Tet-On, Wnt→Wnt) at pairs of adjacent sites (4 bigrams * 4 position-pairs).”: Why did the authors choose to encode and compress their DTT sequencing reads in this manner? Is it not feasible to just use sequencing read proportions of raw DTT sequence patterns? Furthermore, can they predict input signal patterns from the sequencing results?

R1.11: These are great suggestions that we had yet to try. There are three possible states for each site in this experiment, (*unedited*, *Tet-on symbol*, *Wnt symbol*), such that there are 3^5 or 243 possible state sequences across the 5-unit DTT. To test the suggested “raw” approach, a matrix of the absolute frequencies of each of

these 243 state sequences observed in association with a given program was calculated. In parallel, we also tried an intermediate approach, in which we enforced a requirement that editing is fully ordered from left to right along the tape, such that there are $(2^0+2^1+2^2+2^3+2^4+2^5)$ or 63 possible state sequences (*i.e.* 2^0 or 1 possibility for fully blank tape; 2^1 or 2 possibilities for tapes in which only the first position is written; 2^2 or 4 possibilities for tapes in which the first and second positions are written, etc.).

We then performed PCA, quantified how much variance is explained by the first N principal components, and visualized the distribution of programs (and replicates) relying on the first three PCs. In newly added **Supplementary Figure 8a-c**, reproduced below, we show the results for the original approach (26 rows) vs. these two new approaches (63 or 243 rows). Overall, our original approach results in more variance explained with fewer PCs, but the alternative approaches are also viable. In particular, 96% of the variance is explained by the first 5 PCs with our original encoding, 89% with the intermediate approach, and 50% with the “raw” approach. The poorer performance observed with 63 or 243 variable encoding is likely attributable to the “curse of dimensionality”, *i.e.* the less compressed matrices are much sparser than the compressed matrix with a much higher proportion of low-value or 0-value entries.

To answer the question about whether we can predict signal patterns from sequencing results, we trained a random forest classifier on the task of assessing which of the 15 signal programs an unseen set of sequenced tapes derived from. We randomly split the tape ensembles from 45 samples (15 programs x 3 replicates) into 5 groups and conducted 5-fold cross-validation, *i.e.* using each group once as a test set, while training on all other groups. The model achieved a mean accuracy of 0.91 (**Supplementary Figure 8c**). Layered signal programs appeared particularly prone to misclassification, and in particular those that were closely located on the PCA plot (**Supplementary Figure 8b**), *e.g.* Tet-On (1d) → Wnt + Tet-On (5d) versus Tet-On (2d) → Wnt + Tet-On (4d).

Supplementary Figure 8. Combining ENGRAM and DNA Typewriter to record the temporal dynamics of biological signals. (a-c) Comparison of different encoding strategies. DTT recording data was encoded as either the proportions of various unigrams/bigrams at each position ($2^5 = 10$ unigrams; $4^4 = 16$ bigrams; 26 values) (**left**), the proportions of each possible state sequence that is consistent with ordered recording ($2^0+2^1+2^2+2^3+2^4+2^5=63$ values) (**middle**), or the

proportions of all possible sequences of three states (*unedited*, *Tet-on symbol*, *Wnt symbol*) across 5 positions ($3^5 = 243$ values) (**right**). (**a**) Scree plots showing the proportion of variance explained by the top 5 principal components with each of the three strategies), which explain 96%, 89% and 50% of the variance, respectively. (**b**) PCA based on patterns observed in DTT across 45 cell populations subjected to various patterns of exposure to doxycycline and CHIR-99021 (15 programs, executed in triplicate). Circled subsets correspond to serial and layered programs in either order, or to pulse programs. The top three PCs are plotted for 26 value (**left**), 62 value (**middle**) or 243 value (**right**) encoding. (**c**) Barplot showing the accuracy of applying a random forest classifier to assess which of the 15 signal programs an unseen set of sequenced tapes derived from. We randomly split the tape ensembles from 45 samples (15 programs x 3 integration replicates) into 5 groups and conducted 5-fold cross-validation, *i.e.* using each group once as a test set, while training on all other groups. The model achieved a mean accuracy of 0.91.

12. Supplementary Figure 7b-d: What explains the observation that the editing frequencies sometimes exceed 50%? In contrast, the editing efficiencies shown in Figure 3 were only up to a few percent. Why are the editing efficiencies in this instance so high? Could these efficiencies be relative to a specific reference or baseline?

R1.12: Great catch. These are indeed real fractions and a really big difference that is not due adjusting for some baseline. We think that the massive differences in editing efficiencies between the early and latest experiments are due to a few factors. Focusing on **Figure 3** vs. **Supplementary Figure 7**, there are several key experimental differences. First, both experiments are in monoclonal HEK293T cells, but the **Figure 3** experiment was with PE2, while the **Supplementary Figure 7** experiment was with PEmax. As we have confirmed, we expect higher activity with PEmax. Second, although neither line was explicitly selected for high levels of PE expression, the PEmax line was sorted for high mCherry expression, so we may simply have better expression of the prime editor, which would further augment any difference between PE2 and PEmax. Third in the **Figure 3** experiment, we are writing to the endogenous *HEK3* site, while in the **Supplementary Figure 7** experiment, we are writing to DNA Typewriter Tape (DTT) that is embedded in a constitutively expressed transcript (to facilitate its recovery in sc-RNA-seq experiments). In separate work, we have recently shown that high levels of transcription promote prime editing². We speculate that by positioning DTT within a highly expressed transcript, we may be facilitating higher levels of editing. The relevant section of the text now reads as follows:

“Most recording occurred at the first two sites of DTT, indicating that we had yet to saturate recording capacity (**Supplementary Figure 7b-c**). The higher levels of editing relative earlier experiments (e.g. exceeding 60% at the first DTT site in layered programs) may be due to several factors, including the switch to PEmax (**Supplementary Note 1**), sorting of this PEmax line for high levels of co-expressed mCherry, and the switch from writing to the endogenous *HEK3* site to DNA Typewriter Tape (DTT) embedded in a constitutively expressed transcript².”

13. Figure 5: This demonstration is impressive, and I can see a lot of effort has been invested here. However, I wonder why the experiment wasn't designed such that the recording result can be read out by scRNA-seq. Could the authors at least perform scRNA-seq of different stages and compare their pseudobulk results to the recording results? Additionally, is it possible to project the recording results back onto the scRNA-seq data to identify which cell types were predominant in providing those recording results? I find these analyses to be extremely exciting and believe they would not be overly challenging for this team. However, I already appreciate the extensive amount of work undertaken, so I won't insist on this.

R1.13: We are very excited about this direction as well. Indeed, in the DNA Typewriter paper³, we showcase the possibility of combining this style of recording with sc-RNA-seq (albeit in HEK293T cells, rather than gastruloids, and solely recording lineage). To provide further context around why we did not include such

experiments or results here, early in the course of the revisions, before we had built and tested the synthetic TF ENGRAM recorders, we sought to record Wnt signaling dynamics during mouse gastruloid induction. The results were promising (see **Reviewer Figure 2** below) but suffered from three overarching challenges. First, as we were using PE2 at the time, editing rates were low. Second, despite targeted enrichment in the course of leveraging the 10X Genomics platform for sc-RNA-seq, we were only able to capture DNA Tape from 11% of cells, limiting power to detect differences between cell types. Third, although the results are broadly consistent with expectation (e.g. neural cell types are expected to suppress Wnt signaling and exhibit lower levels of recording), the fact that a Chiron pulse is an early step in the gastruloid induction protocol meant that all cell types exhibited some degree of Wnt recording; because we had not yet combined ENGRAM and DTT, we could not distinguish between early vs. late Wnt signaling. Coupled with low power due to poor recovery of the DNA Tape on a per-cell-basis, the results are very difficult to interpret.

We later repeated the mouse gastruloid experiment with a PEmax line and sci-RNA-seq3 (hoping that a much larger number of cells obtained with combinatorial indexing would compensate for the low proportion of cells from which DNA Tape was recovered). In this experiment, we indeed observed a much higher editing rate, but the proportion of cells from which we recovered DNA Tape dropped to just 5%.

In light of all we have learned, to do the right experiment in gastruloids, rather than continuing to do sub-optimal experiments, we need to do the following:

1. Establish a high expression, inducible PEmax line in mESCs that does not silence upon differentiation, ideally at a safe harbor locus.
2. Establish a version of DTT in which tapes are circularized in RNA form by the Tornado system, which in separate work on single-cell MPRAs, we have shown massively boosts barcode recovery in scRNA-seq experiments by over 150-fold, resulting in near-deterministic recovery in concert with scRNA-seq profiles (see **Fig. S2** of Lalanne *et al.* bioRxiv 2023)⁴.
3. Introduce this DTT to the PEmax line at high MOI
4. Further modify this line to include a complement of synthetic TF ENGRAM recorders (likely it would not be all 98, but with a high MOI transfection, we could probably do reasonably well).
5. Construct a monoclonal version of this line, such that every cell within gastruloids made using this line will contain an identical complement of DTT and ENGRAM instances.
6. Induce gastruloids with this line, perform sc-RNA-seq, and recover transcriptomes together with Tornado-circularized DTT at high efficiency.

Work along these lines has been distributed across the lab and proceeding as one of our highest priorities. However, because of the amount of time that we anticipate that it will take to really do it well, together with the larger number of people involved, we would like to take the reviewer up on their offer to not insist on such an experiment being included in this manuscript. We are certainly open to including the results presented in **Reviewer Figure 2**, but their sub-optimal nature, especially relative to what we imagine will be possible with the above workflow, are such that our preference would be to not include it.

Reviewer Figure 2. Recording of Wnt signaling during gastruloid induction using ENGRAM. ENGRAM components were integrated into mESCs sequentially, with PE2 and the Wnt-ENGRAM-recorder integrated first followed by DNA TAPE bearing a constitutively transcribed, synthetic HEK3 target site. Roughly 300 modified mESCs were seeded and induced to form conventional mouse gastruloids. Here we show a 2D UMAP generated from scRNA-seq data collected on the 10X Genomics platform from gastruloids harvested at 120 hours, with targeted recovery of the synthetic HEK3 tape. A total of 7341 cells were recovered, of which 811 (11%) had successful DNA TAPE recovery. Clusters could be annotated as expected cell types, including endoderm, extraembryonic endoderm, allantois-like, pre-somatic mesoderm (PSM), notochord, somites, cardiac mesoderm, cardiomyocyte, endothelium, neuromesodermal progenitors (NMP), neural tube and differentiated neurons.

14. Rebuttal “We relied on an ensemble of cells, rather than a single cell, to extract each temporal signal pattern, and we have yet to develop algorithms or models that would allow one to directly infer signaling histories from a newly observed, complex patterns (as opposed to simply showing after the fact that the patterns are distinct, as we do in Figure 4e).”: It’s reasonable to demonstrate not only the ratio metric but also what fractions of the reads support the expected orders.

R1.14: Please see **R1.10** above, which responds to a similar comment, as well as the new panels of **Supplementary Fig. 7**, which display the absolute fractions obtained in these experiments.

15. Rebuttal “For example, if dozens of homing gRNAs were used for different signals, each writing back to the genomic locations from which they were expressed, recovering recorded events from so many such sites by methods such as single-cell RNA-seq (sc-RNA-seq) would be very challenging.”: I’m ok with this statement, but wouldn’t the use of scRNA-seq be particularly advantageous, as evidenced in their previous scGESTALT demonstration? It’s both aesthetically pleasing and highly scalable, to the extent that the minimal amount of information recorded per site becomes negligible.

R1.15: We agree to an extent but there are two key issues. First, the recovery of DNA Tape with scRNA-seq is challenging (see **R1.13** above). In scGESTALT, the fact that we were only looking at lineage made this tolerable, as the many copies of target arrays essentially provided a form of redundancy. In the homing gRNA scenario, each DNA Tape would be recording a different signal, and so the same kind of redundancy would not be there. Second, if different signals were recorded at different sites, inferring signal order would remain extremely difficult, even if one were perfectly recovering all DNA Tapes. In contrast, the combination of ENGRAM and DNA Typewriter writing various signals (and lineage information) to DNA Tapes in an ordered

manner, offers a straightforward means of inferring signal order, even when the number of encoded signals in a system is large (**Supplementary Table 1**).

16. Supplementary Table 1 “signal specific”: If Base Editors (BE) are categorized as pseudorandom, Prime Editors (PE) should also be considered pseudorandom, given that they are not perfect. The authors might want to categorize them based on their degrees of pseudorandomness, where even qualitative metrics would suffice.

R1.16: Thank you for pointing this out, which led us to realize that we were failing to be clear in our intent with this row. Here we were referencing the editing outcome of each genome editing modality. In the specific case of base editing, what we had in mind was when there are multiple editable-residues (A for ABEs and C for CBEs) within the editing window of the target. For example, if there are two consecutive A's (AA) in the target editing window, most base-editing outcomes by ABE will be G's (GG), but fractions of edits will also include a mix of G and A (GA and AG). In certain cases, this is taken advantage of to increase the information content of the lineage recording system (baseMEMOIR⁵). We have clarified this aspect of the **Supplementary Table 1** by avoiding the usage of the term “pseudorandom”. Also, the row title has been changed from “Information content of edit sequence” to “Diversity of editing outcome”, and the columns now read as follows:

Diversity of editing outcome

- CRISPR-Cut: High, determined by the NHEJ process after the cut.
- Base editing: Moderate, determined by the number of modifiable bases within the target editing window.
- Prime editing: Low, determined by the template sequence within pegRNA and possible errors during reverse-transcription/DNA-repair.

17. Supplementary Table 1 “Moderate (system complexity grows exponentially with number of biological signals, because guides/targets must be preprogrammed for each order”): I might challenge the categorization of CRISPR as “poor.” Our group, which regularly tests BE-based CRISPR cell lineage tracing, has recently realized that a major drawback of the BE system is its tendency to generate identical mutation patterns. In contrast, the editing patterns produced by the wildtype Cas9 have a higher information content per site and are irreversible, a feature that can be advantageous in certain contexts.

R1.17: We agree fully with this comment, and realized we worded the title of this row poorly as well. With this row, we wanted to convey the possibility of sequential editing to explicitly encode the temporal/ordering information, which has been demonstrated in the base-editing context by several groups (CAMERA⁶ and DOMINO⁷). This contrasts with the referenced property of BE, which is more relevant to inference of order. Please see **R1.16** above, as our revisions described there make it explicit that Cas9 > BE > PE with respect to the diversity of editing outcomes, which we believe is the point that the reviewer wants us to make clearer.

We have also revised the title of this row of **Supplementary Table 1** from “Potential for recording temporal order of biological signals” to “Potential for explicitly recording the temporal order of biological signals”

Minor comments:

18. “(or combinations of fluorescent reporters, in the case of Brainbow)^{5–7}.”: I would suggest adding the polylox lineage tracing (or clone labeling) method here.

R1.18: We thank the reviewer for the suggestion. We have substantially revised the relevant passage to now read as follows:

“As conventionally used by developmental biologists, the Cre SSR is expressed under the control of a cell type-specific enhancer. In cells or tissues where that enhancer is active, Cre-mediated recombination at a target locus excises a stop sequence to permit expression of a fluorescent reporter, such that that cell and its descendants continue to express the reporter regardless of whether the gating regulatory element remains active or not. Multiplex implementations of SSR-based recorders leverage excision and flipping to combinatorially diversify fluorescent reporter expression (e.g. Brainbow^{8,9}) or DNA barcode production (e.g. intMemoir¹⁰, PolyLox¹¹) for clone or lineage tracing.”

19. “The enzymatic event is irreversible”: This actually varies depending on the circuit construction. For instance, a single flipping unit enclosed by two loxP sites in reversible orientations can indeed be reversible.

R1.19: Thank you for pointing this out, and please see **R1.18**, where our revisions to the relevant passage make it clearer that the referenced irreversibility is specific to typical use-case of SSRs in developmental biology (i.e. *loxP-stop-loxP* cassettes).

20. “We tested PE2 vs. PEmax28 in the context of ENGRAM in both K562 cells and mouse embryonic stem cells (mESCs). In K562 cells, constitutively active ENGRAM recorders bearing 5N insertions, together with PE2 or PEmax, were transiently transfected in triplicate, targeting the endogenous HEK3 locus. In mESCs, we first integrated a synthetic DNA Tape (bearing a copy of the HEK3 target site) together with either PE2 or PEmax via piggyBac. We then transiently transfected a ENGRAM recorder encoding a 5-bp insertion to be written to HEK3, again in triplicate. Both K562 and mESCs were harvested 3 days post-transfection, and the insertions at endogenous or synthetic DNA Tape quantified.”: I really appreciate these careful analyses in responding to the reviewers' comments, but I feel that they could be more appropriately moved to a figure legend or supplementary text. Also, it should be “an ENGRAM”?

R1.20: We thank the reviewer for the suggestion. We agree, and this text and analyses have all been moved to **Supplementary Note 1** (see also **R1.2**, **R1.3**, **R1.21**). Please let us know if we can further improve the text.

21. “Finally, we tested whether tRNAs15 could be used as an alternative to *csy4* to facilitate pegRNA release. However, upon replacing the *csy4* hairpins with tRNAs, we observed a nearly complete ablation of recording activity (Supplementary Figure 1i). Overall, these experiments suggest that future ENGRAM experiments would benefit from switching to more active prime editors such as PEmax, but not from switching to epegRNAs nor tRNAs.”: This could also be moved to a supplementary text (I know I might be the one who provided all these annoying comments!).

R1.21: We thank the reviewer for the suggestion. We agree, and this text and analyses have all been moved to **Supplementary Note 1** (see also **R1.2**, **R1.3**, **R1.20**). Please let us know if we can further improve the text.

Referee #2 (Remarks to the Author):

The revised manuscript incorporates 2+ years' worth of work by the Shendure lab and the team is to be commended for creativity and perseverance. While we are in 2024 vs 2021, it still remains the case that a lot more about human gene regulation is quite murky, and the structure-function of cis-regulatory elements remains an area that is murkiest. So at a high level, the field needs as many robust, scalable ways to address this issue as possible, and the work in the revised manuscript is relevant and interesting.

R2.1: We thank the reviewer for these kind words.

Two major concerns were raised in the initial review and the authors' response to them is discussed below.

The first concern related to whether the new method has an operationally useful linearity and dynamic range (ie whether regulatory elements of low, medium, and high activity were consistently recorded as ranking in that order by the new assay). The original manuscript represented these data in a log₂/log₂ scale and thus made it challenging to assess this. The revised manuscript provides the reanalysis requested and the answer to the question appears to be a qualified “yes”: fig 2d demonstrates that the relationship between MPRA activity and ENGRAM activity of randomly sampled CREs is a decent one. There appear to be only 2 CREs that are inactive by ENGRAM and active by MPRA, and no CREs that exhibit the opposite behavior. Fig 2e is an interesting result: the highest-active CREs are the highest-correlated between the two assays, while the correlation becomes weaker (but still clearly there) as enhancer activity drops. Supplementary fig 4a shows more or less the same data but, importantly, adds CRE-by-CRE comparisons. Taken together there is good news – ENGRAM does rank CREs – and other news: we won't know the ground truth until we have genome-edited a whole bunch of CREs assayed in their native genomic location and looked at effects on target gene control. Overall the reanalysis fully addresses the concern raised.

R2.2: We are glad to hear that this concern has been fully addressed by the reanalysis.

The second concern related to whether the ENGRAM construct could be used to detect activity at endogenous genes. Such reporter tagging has been done at native human genes with fluorescent constructs since 2010 (21297641), and as mentioned in the initial review, Manu Leonetti at the Biohub has tagged 1000s of human genes this way (35271311) and all their gRNAs that worked for this massive effort could be repurposed wholesale for such an experiment. The reply to reviewer comments the authors provide presents an extensive and thoughtful discussion of this issue, two salient points of which are addressed below.

[REDACTED]

R2.3: Thank you for understanding our concerns on this point.

The second issue is the authors – in another burst of trademark Shendure-lab creativity – do a related experiment to address this point. In brief, a promotorless reporter pegRNA is integrated using piggyBac, an ingenious method to map integration sites is then used to characterize the genome-wide insertion spectrum of the reporter constructs, and then the correlation between endogenous gene transcription and ENGRAM activity is determined. Two findings emerge.

The first finding is the transcriptional activity recording is dominated by reporter constructs that landed into rDNA. Of note – and this is, in the eye of this reviewer a fascinating result – it is not the case that all these new “pol1 reporter constructs” are comparably active: in fact, as shown in reviewer figure 1h, these show a very wide range of “writing/recording activities.” This means that individual rDNA units are unequal in terms of

transcriptional output, or specific insertion sites within the rDNA are “accommodating” the newly arrived reporter into rRNA with different efficiencies. One imagine that folks studying rDNA transcription control would find this of formidable interest (there are some data on this in Arabidopsis <https://www.nature.com/articles/s41467-020-20728-6>).

R2.4: Agree. The reviewer may be interested to know that this finding is motivating new exploratory projects in the lab, including attempting to engineer both guides and targets into a synthetic nucleolus¹².

The second finding is that the remaining signal from reporter constructs does not correlate with landing site transcriptional activity (reviewer figure 1h and 1k). These important data should be reviewed by insertion class, and here, the data on recording signal from intron-resident reporters are noteworthy. One can argue that the exon insertions are uncorrelated to native gene activity because the reporter surely brings in a premature stop codon in itself or downstream and native human genes are differentially sensitive to NMD. One can also argue that promoter insertions are hard to interpret because of the location of the native gene TSS (and coding region in exon 1) relative to where the cassette landed. Why would the intron-resident cassettes be so poorly correlated in their recording activity relative to what the native gene does transcription activity-wise? The simplest possibility is the cassette has a cryptic splice acceptor (or donor) that perturbs the native splicing pattern, but differently so dependent on location of insertion site in the intron and the local landscape of splicing enhancer and repressor signals (as in the SMN2 gene that is targeted by spinraza). The editing literature has a classic result from Wen Xue (28615073) that the cell ignores the editing-driven knockout of an exon by just skipping it.

R2.5: This possibility had not occurred to us. We thought that it perhaps might relate to the location and timing of Csy4-mediated excision of the pegRNA (*i.e.* perhaps this can only happen from the mature mRNA?) but the suggestion above seems entirely plausible as well.

The above is a hypothesis that, frankly, does not deserve to be tested because the purpose here is to measure gene activity in native contexts, and as the authors’ careful analysis shows, the exact construct and approach they used to do that fails to meet that expectation. In the mind of this reviewer a brief reference to this result – as “unpublished observations” – would be warranted to prevent others from following this path. While the result is a negative one, the authors are to be commended for doing this creative experiment and thus refocusing future effort on minimal-engram-cassette integration into native genes or isogenic landing pads.

R2.6: We agree. We are not 100% sure about *Nature’s* policy on stating “unpublished observations” (some journals do not allow this), but under the assumption that it is permissible, we have added the following to the Discussion:

“One can imagine variants of the ENGRAM strategy, *e.g.* integrating a minimal *csy4*-pegRNA-*csy4* cassette to endogenous gene bodies, with the goal of recording endogenous gene expression levels to DNA Tape. Our initial attempts at exactly this leveraged random integration of a T7-mappable² version of a minimal ENGRAM cassette, but failed in that the barcode proportions observed in DNA Tape did not correlate with the expression levels of genes in which pegRNAs encoding those barcodes resided (unpublished observations). Possible explanations for this negative result include the dominance of barcodes derived from pegRNAs that happened to integrate within highly transcribed ribosomal gDNA regions (confounding dynamic range), the short life and nuclear location of the pre-mRNA (confounding intronic integrations) and/or cryptic splicing sites within the minimal ENGRAM cassette (confounding exonic integrations). Further work is required to adapt ENGRAM to quantitatively record endogenous gene expression levels.”

The rest of the comments raised in initial review were minor and are adequately addressed in the revision.

R2.7: We thank the reviewer for their constructive comments on the original and revised versions of the manuscript.

// Fyodor Urnov, Innovative Genomics Institute, UC Berkeley //

Referee #3 (Remarks to the Author):

This revised manuscript includes two major additions. Figure 4 illustrates that ENGRAM can resolve temporal patterns of activity. This is a technically impressive and conceptually exciting addition. But the results also illustrate some practical limitations of the approach (inability to discriminate pulse, layered, serial), and the results remain at the level of proof-of-principle (with strong stimuli given to cells in a dish) rather than an illustration of real new biology that could not be discovered otherwise. In real biology, where signals are often weaker and more complex, it will still be a major challenge to apply this ENGRAM approach successfully.

R3.1: We appreciate the reviewer for noting that the newly added **Figure 4** is “technically impressive” and “conceptually exciting”. We agree that the results remain at the level of proof-of-principle of a technology with major potential rather than new biology. In further revising the manuscript, we have sought to make changes at appropriate places in order to generally tone it down to a level more aligned with current performance. We also added new sentences or clauses, in the abstract, introduction and discussion, that explicitly note this is a proof-of-concept study, and that more work is needed. Please see **R1.1** above for further details.

We also note that while addressing a related comment (see **R1.11**), we performed new analyses to ask whether we can predict signal patterns from sequencing results. Specifically, we trained a random forest classifier on the task of assessing which of the 15 signal programs (**Figure 4**) an unseen set of sequenced tapes derived from. We randomly split the tape ensembles from 45 samples (15 programs x 3 replicates) into 5 groups and conducted 5-fold cross-validation, *i.e.* using each group once as a test set, while training on all other groups. The model achieved a mean accuracy of 0.91. Layered signal programs appeared particularly prone to misclassification, and in particular those that were closely located on the PCA plot, *e.g.* Tet-On (1d) → Wnt + Tet-On (5d) versus Tet-On (2d) → Wnt + Tet-On (4d). The inclusion of this new result highlights both the possibilities and remaining challenges of leveraging the combination of ENGRAM and DNA Typewriter to record and decode complex signal patterns.

We respectfully disagree with the statement that ENGRAM is not sufficiently sensitive to record biologically relevant signals, even in its current form. For Wnt signaling, 3 μ M of CHIR99021 is used to induce gastruloid formation¹³, which falls into the 1-4 μ M range tested by us. For NF κ B signaling, we used TNF α as an agonist. Although the sensitivity of this pathway to this agonist varies across cell types, it has been shown in reporter assays that the widely used NF κ B response element has an EC₅₀ of 3.34 ng/mL¹⁴ for TNF α in HEK293T cells, which falls within the 0.1-16 ng/ml range over which we show ENGRAM is sensitive. These points are discussed in the **Figure 3** legend.

A second major addition in the revised manuscript is the multiplexed probing of 98 different TF motifs. While I appreciate that this is meant to further illustrate that “biology” can be uncovered with ENGRAM, I am afraid that the authors have opened a can of worms. For several reasons, I think these results could be dominated by artefacts and noise (comment a-e) while there is a much simpler alternative (f):

a. It is far from trivial to design TF reporters. Full disclosure: my lab has worked on the design of such reporters over the past four years (manuscript in preparation). We have learned that every TF has its own requirements when it comes to the spacing of the motifs; the spacer sequence between the motifs; the distance of the motifs to the core promoter; and the identity of the core promoter. When these parameters are not carefully tuned for each TF motif individually, there will be lots of false-negatives and false-positives, and certainly one cannot trust quantitative differences between reporters within and between cell lines

R3.2: This concern may have arisen because of the manner in which we introduced this topic: (“To assess whether ENGRAM can record the differential activities of TFs between mammalian cell types...”), which in retrospect was clearly imprecise language. We have revised this to now read as follows:

“We sought to design synthetic CREs that capture the differential propensity of individual TF motifs to enhance transcriptional activation in mammalian cell types.”

We agree that designing a TF reporter that is solely and optimally responsive to a single TF is very challenging. Rather, we aimed to design synthetic CREs with a reasonable degree of responsiveness, specificity and reproducibility for one or more members of a TF family known to bind that motif. We have experience on this topic as well¹⁵, and a homotypic array of TF motifs will perform suboptimally but sufficiently for many purposes. Apart from how we introduced the section, we were careful to describe these as “synthetic CREs” and not “TF reporters” or “TF recorders” for this reason (in one instance, we referred to them as “TF motif recorders”, which we would argue is sufficiently qualified). Also, please note that the TF labels shown in some figures are intended to be representative as stated in the legends (e.g. **Figure 2g**), and the full list of TFs that potentially bind each motif based on current knowledge is provided in **Supplementary Table 3**.

We would like to respectfully disagree with the assertion that we cannot make quantitative assessments of recording differences between cell lines. We provide one line of evidence in the revision, and add a second with this further response and additional revision. First, as shown in **Supplementary Figures 5b** (HEK293T), **5c** (K562), and **8a** (mESCs) (reproduced below), transfection replicates exhibited a high level of reproducibility. Second, as shown in newly added **Supplementary Figure 9c-e**, reproduced below, if we perform pairwise comparisons between individual replicates of different cell lines (e.g. replicate 1 of cell type A vs. replicate 1 of cell type B), the ratios that we obtain are also highly reproducible ($\text{rep1-A} / \text{rep1-B} \approx \text{rep2-A} / \text{rep2-B}$). Together, these results show that we can reproducibly record the relative activity of a synthetic CRE in cell type A vs. cell type B. As stated in the figure legends, the lower reproducibility for comparisons involving K562 cells is likely secondary to lower transfection/editing efficiency.

Supplementary Figure 5. Multiplex recording of cell-type-specific activities of synthetic CREs with ENGRAM. (b-c) Log-scaled insertion proportions for 5-mer barcodes linked to the 98 synthetic CREs were highly reproducible across transfection replicates for both HEK293T (**b**) and K562 (**c**) cells. Each value corresponds to the proportion of barcodes read out at the DNA level from the endogenous *HEK3* locus. As the same number of cells were sampled for recording, the lower reproducibility in K562 cells is likely secondary to lower transfection/editing efficiency.

Supplementary Figure 9. Multiplex recording of CRE activity in embryonic stem cells and differentiating gastruloids. (a) Log-scaled insertion proportions for 5-mer barcodes linked to the 98 synthetic CRE-driven ENGRAM recorders were highly reproducible across integration replicates for cultured mESC, as read out by amplification and sequencing of synthetic DNA Tape. **(b)** Log-scaled proportions for 5-mer barcodes linked to the 98 synthetic CRE-driven ENGRAM recorders are well correlated between the original plasmid pool and genomic integrations in the polyclonal mESC line. **(c-e)** Log-scaled barcode proportion ratios, as calculated from one pair of replicates vs. as calculated from another pair of replicates, for mESC vs. HEK293T cells (**c**), mESC vs. K562 cells (**d**), or K562 vs. HEK293T cells (**e**). Note that we corrected the mESC data for differences in relative abundances of recorders in the polyclonal mESC line vs. the plasmid pool used to transiently transfect K562 and HEK293T cells (as shown in **Supplementary Figure 9b**), prior to performing these comparisons. As the same number of cells were sampled for recording, the lower reproducibility for comparisons involving K562 cells is likely secondary to lower transfection/editing efficiency, as shown in **Supplementary Figure 5a**.

On the other hand, we acknowledge that the evidence that we presented regarding which TF explains the differential behavior of each significantly differential synthetic CRE, was overly anecdotal. In particular, we asserted that most of the significant, substantial differences in recording proportions observed in HEK293T vs. K562 cells “were concordant with the differential expression of at least one TF that putatively binds the corresponding TF motif”, and further noted examples of coincident differential expression and differential recording. In light of this aspect of the reviewer’s point, we substantially revised **Supplementary Figure 5d**, now **Supplementary Figure 5d-e** (reproduced below), to show the relative expression levels of all TFs that are putatively associated with a given TF motif recorder by the JASPAR database, rather than only a selected TF. We have also simplified the text calling out these panels to avoid making claims around specific TF::TF-motif relationships. It now reads:

“Most of these differential recording activities are directionally concordant with the summed expression of TFs that JASPAR assigns to the TF motif of a given synthetic CRE recorder (13/17; $p = 0.02$; binomial test; **Supplementary Figure 5d-e**). However, these coincidences should be interpreted with caution, as we have not validated that these TFs are binding to the corresponding synthetic CREs in these cell lines.”

Furthermore, the anecdotal highlights about coincidences between TF expression and synthetic CRE recorder activity have been cut. In our view, this adds sufficient caution for the reader, but we would be happy to entirely strike **Supplementary Figure 5d-e** and the text about it if the reviewer feels the above changes are insufficient. As per the first part of our response to this comment, our claim that the TF motif recorders exhibit highly reproducible differential recording between the two cell lines is not dependent on **Supplementary Figure 5d-e**.

Further down, we amended the paragraph about mESCs vs. other cell line to add a similar qualification and minimize the odds of over-interpretation by the reader:

“The mESC-recorded activities were highly reproducible (**Supplementary Figure 9a**). Of the 98 synthetic CRE ENGRAM recorders, 22 (mESC vs. K562) and 16 (mESC vs. HEK293T) exhibited significant and substantial differences (Wald-test with Benjamini-Hochberg correction $P < 0.001$ for a fold-difference > 2 ; **Figure 5b-c**; **Supplementary Figure 9b**). Among these, there were 5 recorders that were consistently more active in mESCs, including those bearing versions of the SOX, PRDM4, retinoic acid response element (RARE), and ZBED1 motifs. In the other direction, the 2 consistently less active mESCs included recorders for TBX and NFkB motifs. The cell type-specific differences were consistent across independent pairs of integration/transfection replicates (**Supplementary Figure 9c-e**), and recording events were contributed by nearly all ENGRAM recorders in all samples tested (**Supplementary Figure 9f**). However, we reiterate that although the motifs homotypically embedded in the synthetic CRE recorders are associated with these TFs or TF families in the literature, it remains uncertain which specific TF(s) are driving their recording activity.”

Supplementary Figure 5. Multiplex recording of cell-type-specific activities of synthetic CREs with ENGRAM. (d-e) Differential expression of TFs in HEK293T vs. K562 cells. As many TFs share similar binding motifs, here we show expression ratios between the cell lines for all expressed TFs (normalized transcripts per million > 0.5 in one or both cell lines) assigned to the corresponding motif by JASPAR, for each of the 17 differentially active synthetic CRE recorders (**Figure 2g**). In the bottom row of each plot, we show an expression ratio based on summing the read counts of all the motif-associated TFs in bulk RNA-seq data¹⁵ from these cell lines, except for GCM1, as its JASPAR-associated TFs (GCM1, GCM2) are not detected as expressed in either cell line. Pink, higher expression in K562 cells; Blue, higher expression in HEK293T cells. Most K562-specific (**d**) and HEK293T-specific (**e**) recording activities are directionally concordant with the summed expression of the TFs that JASPAR associates with the motif embedded in the synthetic CRE (all but the HOXB9, POU2F1, ZNF449, and GCM1-named synthetic CRE recorders; 13/17; $p = 0.02$; binomial test).

b. The TF ENGRAM results are not validated at all. The authors cherry-pick a few TF motifs that "make sense" (and, to their credit, also that don't make sense, e.g. HoxB9 in mESCs), but an objective criterium to judge the results is lacking. For the gastruloids, we are simply provided with a list of TFs that are detected as increased or decreased over time, but validation is completely lacking. At this stage, it cannot be ruled out that the majority of these TF recorder results are artefacts.

R3.3: Much of our response here mirrors **R3.2** above. To summarize our results from testing the synthetic CRE recorders in three cell lines, we show that for all pairwise comparisons (including K562 vs. HEK293T, K562 vs. mESC and HEK293T vs. K562), the differential activities of synthetic CRE recorders are highly reproducible across independent replicates, and therefore very unlikely to be artifactual. In this second revision, we added a new **Supplementary Figure 9c-e**, reproduced above, which makes this point more clearly. We also remind the reviewer that labels/names given to synthetic CRE recorders are exemplary, and that there are multiple TFs that are candidates for driving the observed differential activity. To make the paper clearer on this point, we now heavily qualify our interpretation of the correlation between the expression of these TFs and the activities of recorders bearing the corresponding motifs, and document the relative expression of all TFs associated with each motif, according to JASPER, in revised **Supplementary Figure 5d-e** (and finally, we offer to remove this last result, as it is not essential to our conclusion that the panel of synthetic CRE recorders are reproducibly and quantitatively recording differences between the cell lines). All of these points are detailed in **R3.2** above.

Turning to the gastruloid experiments, our response is analogous. In particular, as shown in **Supplementary Figure 10c** (previously **Supplementary Figure 8d**), reproduced below, the relative activities of synthetic CRE recorders are highly reproducible between integration replicates (*i.e.* independent gastruloid inductions).

Supplementary Figure 10. Multiplex recording of CRE activity in embryonic stem cells and differentiating gastruloids. (c) Log-scaled insertion proportions for 5-mer barcodes linked to the 98 synthetic CRE-driven ENGRAM recorders were highly reproducible across integration replicates for differentiating gastruloids, as read out by amplification and sequencing of synthetic DNA Tape, for integration replicates in which doxycycline was used to induce PEmax during particular 24 hr windows.

The dynamics of synthetic CRE recorders over the time course of gastruloid induction are generally reproducible as well between the integration replicates, as illustrated in what was previously **Figure 5f-h** (now simplified to **Figure 5f-g**; see below).

Figure 5. Multiplex recording of CRE activity in embryonic stem cells and differentiating gastruloids. (f-g) Dynamics of selected ENGRAM recorders during induction of mouse gastruloids, including those bearing arrays of motifs corresponding to major signaling pathways (f) or TF families (g). **Note that these labels are representative of TF(s) thought to bind each motif, and it remains uncertain which TF(s) are driving the activity of each synthetic CRE recorder. See Supplementary Table 3 for the corresponding consensus motifs, and the full list of TFs associated with each motif by the JASPAR database.** Plotted on the y-axis are the log₂-scaled barcode proportion ratios for gastruloids with windowed recording in a particular 24-hr interval (x-axis) vs. cultured mESCs. Integration replicates are plotted separately and the line shadow represents the 95% confidence interval.

Similar to our response to **R3.2**, these results show that ENGRAM can reproducibly record the dynamic activities of CREs over a time course of gastruloid differentiation. However, we also concede that our ability to definitely link these synthetic CREs to specific TFs (or specific signaling pathways) is limited. To address this, we substantially revised and shortened the paragraph that layers on interpretations in order to be more analytically spare:

“Of note, the dynamics of four major developmental signaling pathways—Wnt, Nodal, FGF and retinoic acid—are **potentially** probed by subsets of the 98 recorders bearing motifs for their effector TFs (**Supplementary Table 3**). Synthetic CRE recorders bearing arrays of TCF/LEF (Wnt signaling), SMAD/FOXH1 (Nodal signaling), ETS [one of three versions] (FGF signaling) and RARE [one of five versions] (retinoic acid signaling) consensus motifs were significantly dynamic, rising monotonically with each successive 24 hr interval, but with activity of the TCF/LEF synthetic CRE recorder dropping, and the SMAD/FOXH1 synthetic CRE recorder plateauing, in the final 24 hrs (**Figure 5f**). Other synthetic CRE recorders, **including some bearing homotypic arrays of motifs associated in the literature with core developmental TFs or TF families, exhibited dynamic behavior as well (Figure 5g; Supplementary Table 3). Once again, we reiterate that it remains ambiguous which specific TF(s) are driving the dynamic activity of these recorders in differentiating gastruloids.**”

To summarize, we concede the reviewer’s point that we cannot be conclusive here in linking specific TFs to specific motifs or synthetic CREs, and we should have been more conservative in our language and analytical

speculation (and now are). We further agree that the optimization of individual synthetic CREs to specifically report the activity of specific TFs is a major undertaking, but we argue that it is beyond the scope of this manuscript. We anticipate that as efforts to devise highly specific synthetic CREs for specific TFs (or cell types) unfold, the optimized versions of these elements can be used with ENGRAM to record their dynamic activities in *in vitro* and *in vivo* models of development, just as we have done here with “first pass” versions of a diversity of synthetic CRE recorders.

c. Figure S5d shows correlation with TF expression levels, but this shows only some the small number of TFs detected by ENGRAM and not the large number of TFs that ENGRAM did not detect. There are some odd discrepancies between Fig 2g and S5d: In S5d, ZNF449 is missing and FOSL1 and JUNB are included while they (AP1 recorders) are not in 2g.

R3.4: Thank you for catching these discrepancies. Please see **R3.2** above, which we believe partly covers this concern. In particular, we have substantially revised this figure panel (now **Supplementary Figure 5d-e**, reproduced above) to show the expression ratios of all expressed TFs associated with each of the 17 synthetic CRE recorders exhibiting substantial, differential activity (as per JASPAR; see **Supplementary Table 3**). The reason that there are 15 panels in **Supplementary Figure 5d-e**, rather than 17, is because SOX_v1 and SOX_v2 are shown on the same plot, as their associated TFs are entirely overlapping; and then the TFs associated by JASPAR with the GCM1 synthetic CRE recorder (GCM1, GCM2) are not detected as expressed in either cell line.

Another point indirectly touched on by this comment is that it is challenging to interpret the negative results, *i.e.* the synthetic CRE recorders that were not differentially active in comparisons of cell lines or gastruloid timepoints. To be clear, the synthetic CRE recorders that we are highlighting are those exhibiting significant and substantial differences (Wald-test with Benjamini-Hochberg correction $P < 0.001$ for a fold-difference > 2). For the pairwise cell line comparisons, there were 17 (K562 vs. HEK293T), 22 (mESC vs. K562) and 16 (mESC vs. HEK293T) synthetic CRE recorders that met this threshold, while for the gastruloid experiments, there were 8 synthetic CRE recorders that met this threshold for at least one pairwise comparison of a 24-hr gastruloid time window to mESCs (**Figure 5e**). Altogether, 46 of the 98 synthetic CRE recorders exhibited differential activity in at least one of these comparisons, while the remaining recorders are not necessarily inactive but simply do not exhibit differences in these comparisons.

To make these points clearer to the reader, as well as to reiterate some of the other points covered in **R3.1-R3.4**, we added the following to the Discussion:

“Of note, our 98 synthetic CREs were unoptimized designs consisting of homotypic arrays of representative motifs. Although 46 of these 98 exhibited reproducible, substantial and significantly differential recording activity in at least one comparison (cell line vs. cell line or mESCs vs. gastruloid time-window), we cannot definitively assign these differences to specific TFs. Looking forward, we anticipate that as efforts to devise and optimize synthetic CREs for specific TFs, signaling pathways and cell types advance^{17,18}, such elements can be used with ENGRAM to record their dynamic activities in *in vitro* and *in vivo* models of development.”

d. For the gastruloids, it is mentioned that the overall recording efficiency decreases over time (Fig S8c). How does this affect the interpretation of the time course data in Fig 5e-h? Were these data somehow normalized for this decline? Is this overall efficiency perhaps dominated by a small subset of TFs? Wouldn't it be necessary to include some kind of internal normalization standard that is not dependent on any TF activity?

R3.5: This is a good point we should have been clearer on. In both the cell line and gastruloid comparisons, we are looking at changes in proportions of recorded events, not the absolute number of recording events. As a consequence, we do not need to normalize for differences or changes in overall recording efficiency (whether between cell lines or over the course of gastruloid differentiation), as the level of prime editor expression will uniformly impact all pegRNAs. To be clearer on this point, we added the following sentences in relevant places:

“A further caution is that these differences between cell lines in recorded activity are quantified relative to the other synthetic CRE recorders, rather than in terms of absolute activity.”

“A further caution is that the observed dynamics are relative to all other recorders in this panel, rather than in terms of absolute activity.”

Another great question raised here is whether the data are dominated by a small subset of recorders. Although brought up in the context of gastruloids, the point is relevant to all experiments where we are deploying the panel of 98 synthetic CRE recorders. As it turns out, this is not the case, as illustrated in the stacked barplots shown below, which show the distribution of recorded proportions for each of the 98 synthetic CRE recorders in each of the eight samples. This representation of the data is newly added as **Supplementary Figure 9f** and referenced in the following new sentence:

“The cell type-specific differences were consistent across independent pairs of integration/transfection replicates (**Supplementary Figure 9c-e**), and **recording events were contributed by nearly all ENGRAM recorders in all samples tested** (**Supplementary Figure 9f**).”

Supplementary Figure 9. Multiplex recording of CRE activity in embryonic stem cells and differentiating gastruloids. (f). Stacked bar plot showing the proportion of 5-mer barcodes associated with each of the 98 synthetic CRE-driven ENGRAM recorders in cell lines and gastruloids. Recorder activities are presented in the order of their maximal proportion across all samples. Error bars correspond to standard deviations across 3 transfection replicates (in K562 and HEK293T cells), or 3 integration replicates in mESCs and 2 integration replicates for gastruloid time-windows. Note that these labels are representative of TF(s) thought to bind each motif, and it remains uncertain which TF(s) are driving the activity of each synthetic CRE recorder. See **Supplementary Table 3** for the corresponding consensus motifs, and the full list of TFs associated with each motif by the JASPAR database.

e. The experiments were done in polyclonal cell pools. It seems that the authors assume that each TF recorder is equally represented in these pools. That might seem reasonable, but in practice this is a risky assumption. Due to the (often mutagenic) integrations (the cells carry an estimated 25 insertions) and other clonal effects, there is a serious risk of clonal drift. Please see PMID 35173762, figure 4. Substantial drift can occur within a week! My lab has experienced this in multiple TRIP projects, with in some instances a few clones rapidly taking over the entire pool. It is thus critical that the relative representation of each recorder is determined in parallel, and used to normalize the data.

R3.6: Thank you for raising this point. We have observed this phenomenon as well in our lab; it has cost us many headaches and lost time in other projects, so we are empathetic. In the context of this project, we took some care to design our experiments in a way that we felt we could be comfortable that it was not confounding the results, but we are glad you raised it and we have made some changes as a consequence. However, we note that the concern is more relevant for some comparisons than others:

1. For HEK293T and K562 cells, the panel of synthetic CRE recorders were transiently transfected into cells and recordings measured 2 days post-transfection. Because the same plasmid library was independently (and transiently) transfected into each replicate of each cell line, the reproducibility within cell lines, as well as the reproducible differences between the cell lines, should not be impacted by drift, as there is no integration of recorders. As noted above, the poorer reproducibility of our results in K562 cells is likely due to low transfection efficiency.
2. For the mESCs, we constructed a single polyclonal line to which the synthetic CRE recorders were integrated by piggyBac. This line was maintained as a large population of cells, *i.e.* no bottlenecks. For mESC replicate comparisons, it was split into three subpools, followed by doxycycline induction of PEmax (occurring separately in each cell line), and then harvesting of genomic DNA and sequencing. The high level of reproducibility in comparisons of these mESC replicates to one another indicates that drift subsequent to splitting was not an issue.
3. For the gastruloid induction experiments, additional cells from the same mESC polyclonal line were split to individual wells, subjected to the gastruloid induction protocol, and then 20-50 gastruloids from each well were taken as each integration replicate. Here, comparisons were between gastruloid timepoints vs. the mESCs referenced above. Therefore, although we have the risk that drift impacts the composition of recorders in each gastruloid replicate, the fact that the measurements and differences are highly reproducible between the gastruloid replicates indicates that drift was not an issue. Note that although each gastruloid is seeded by only ~300 cells, the fact that we were using ~20-50 gastruloids per replicate well, together with the high MOI at which these recorders are present per cell, is probably why the experiment did not suffer from drift (as in other experiments, we do observe drift in individual gastruloids, consistent with the reviewer's experience).
4. The context where the concern is most relevant is comparisons between mESCs vs. K562s or mESCs vs. HEK293T cells. Although the same plasmid library that was used for transient transfection (of K562s or HEK293T cells) was also used for piggyBac mediated integration (in mESCs), it is possible that drift occurred during the establishment of the polyclonal mESC population bearing integrated recorders, and we had not looked into this previously. To address this, we sequenced the 5-bp barcodes of recorders integrated to the mESC cell population, and compared their relative abundances to those of the plasmid library that was used for both transient transfections and stable integrations. This has been added to the paper as **Supplementary Figure 9b** and is reproduced below. Although there is a strong correlation, it's

not perfect. Therefore, we normalized the mESC counts to account for this, and updated our comparisons of mESCs vs. K562s or mESCs vs. HEK293T cells. The original and updated versions of **Figure 5b-c** are reproduced below as well. The results are generally very similar, and there are now 22 (mESC vs. K562; originally 28) and 16 (mESC vs. HEK293T; originally 19) synthetic CRE recorders that meet the threshold of significant and substantial differences (Wald-test with Benjamini-Hochberg correction $P < 0.001$ for a fold-difference > 2). Other numbers and figures involving comparisons of mESCs vs. K562s or mESCs vs. HEK293T cells have also been updated to reflect this normalization prior to performing the comparison.

Supplementary Figure 9b. Multiplex recording of CRE activity in embryonic stem cells and differentiating gastruloids. Log-scaled proportions for 5-mer barcodes linked to the 98 synthetic CRE-driven ENGRAM recorders are well correlated between the original plasmid pool and genomic integrations in the polyclonal mESC line.

Old volcano plot. mESC counts are not normalized by abundance

Figure 5. Multiplex recording of CRE activity in embryonic stem cells and differentiating gastruloids. (b-c) Volcano plot of differential recorded activity of ENGRAM recorders in cultured mESCs vs. K562 (b) or cultured mESC vs. HEK293T (c) cells. Each plotted point represents a distinct ENGRAM recorder driven by a synthetic CRE composed of a homotypic array of the consensus DNA binding motif for a TF or TF family (**Supplementary Table 3**). Red, labeled points are ENGRAM recorders exhibiting significant and substantial increasing activity (Wald-test with Benjamini-Hochberg correction $P < 0.001$ for a fold-difference > 2). **Top:** updated Figure 5 (b-c). mESC recording counts were normalized to account for slight differences in their abundance in the plasmid library vs. the polyclonal mESC line. **Bottom:** previous Figure 5 (b-c), where the mESC counts were not normalized for abundance shifts.

f. Moreover, it should be considered that there is a widely used alternative to estimate TF activities in complex tissues: single-cell ATAC-seq. The Shendure lab has published extensively with this technology, in particular to infer TF activities in various complex systems such as developing embryos. scATAC-seq also allows clustering of the data by cell type, while ENGRAM currently detects 'averaged' signals across a mix of cell types. Considering the above uncertainties about the quality of the TF recorder data, what then is the advantage of ENGRAM for monitoring TFs? scATAC-seq is commercially available and much simpler to set up than multiplexed ENGRAM. No doubt, scATAC-seq has its limitations, but from the current manuscript it remains entirely unclear whether ENGRAM will do a better job detecting TF activities - so far it seems to do worse.

R3.7: We do indeed have extensive experience with sc-ATAC-seq. We note that the inference of TF activities with sc-ATAC-seq suffers from the same limitations as highlighted here, *i.e.* in general, one is inferring activity of the TF motif, and it is only with co-assays or accompanying sc-RNA-seq data that one is able to pin it to a specific TF.

Putting that aside and focusing on TF motif activity inference, there are both immediate and longer term advantages to ENGRAM over sc-ATAC-seq. The immediate advantages of ENGRAM over sc-ATAC-seq include: (1) the ability to integrate over a window of time, rather than a snapshot of a single moment; (2) the ability to ascertain activity based on a single measurement (*i.e.* one reporter per TF motif) rather than genome-wide data (*i.e.* motif enrichment across a genome-wide dataset); (3) ENGRAM is a functional assay, whereas sc-ATAC-seq is a biochemical assay; and (4) like MPRA, ENGRAM is an immediately broadly accessible technique without requiring an expensive commercial kit, while the technical complexity of sc-ATAC-seq has resulted in most groups relying on an expensive commercial kit (even though that is not strictly necessary).

The longer term advantages of ENGRAM over sc-ATAC-seq include: (1) we envision that it will be possible in the near future to deploy sc-ENGRAM (see **R1.13** above, where we describe experiments not included in the paper that show that we can do this, even as we would like to optimize it further before reporting on it); (2) the combination of ENGRAM and DNA Typewriter will make it possible to order synthetic CRE activities over time, *i.e.* enabling longitudinal measurements from a single sample, something that is definitely not possible with sc-ATAC-seq.

In conclusion, I reiterate that it is an exceptionally cleverly and thoroughly designed technology that will inspire scientists in the field, and I fully appreciate how much additional work went into this revised manuscript. But I regret to say that the new TF recorder data are too preliminary and do not make a convincing case that ENGRAM will be a new workhorse in the field to probe TF activities. The ability to do temporal recording is demonstrated in a truly elegant proof-of-principle experiment, but it remains to be seen whether this will be more practical than (say) using fluorescent reporters or ATAC-seq.

R3.8: We thank the reviewer for describing ENGRAM as a “exceptionally cleverly and thoroughly designed technology that will inspire scientists in the field”, for acknowledging the work involved with the new experiments, and for characterizing the temporal recording experiment as a “truly elegant proof-of-principle”. We note that our goal here is not a short term replacement of fluorescent reporter or ATAC-seq. Rather, we believe that multi-channel biological recording to genomic DNA, as made possible by ENGRAM and its intersection with DNA Typewriter, has the potential to fundamentally transform how we make longitudinal biological measurements *in vitro* and *in vivo*. There is still a lot of work between where the technology is at now and realizing that vision, but the fundamental potential is there and the path is clear. Our lab has entirely reoriented to this goal (*i.e.* this is not a “one-off”), and we hope to inspire others to do the same.

Other points:

In each figure legend: please clarify whether the replicates were *independent* biological replicates (done on different days), or technical replicates (done in parallel dishes or tubes on the same day). In Fig 4bcd the reproducibility seems exceptionally high for independent biological replicates.

R3.9: Thank you for pointing this out. These terms are used quite heterogeneously in the literature, and what we might call a biological replicate, it sounds like you might call a technical replicate. To address this question and provide maximum transparency and clarity to the reader, we have added **Supplementary Table 7**, in which we detail what is meant by “replicate” in each main and supplementary figure panel in which the term is used. We don’t use the terms “technical replicate” or “biological replicate” any longer, but rather have broken things down into “transfection replicates” and “integration replicates”, with fuller details provided in this table.

For example, to the specific question of **Fig. 4b-d**, a single polyclonal population of cells bearing integrated copies (via piggyBac) of both the Tet-On and Wnt recorders at high MOI was generated. For each program, cells were split from this population into three wells and cultured and treated to the specified agonist regimes separately. So, the replicates in question were done on the same day, but derive from different cells.

Experiments described in:	Transfection or integration	# of replicates	Term	Comments
Fig. 1c	Integration	3	Integration replicates	For each ENGRAM variant, recorders were integrated into a single, polyclonal population of cells via piggyBac. This polyclonal population was split to 3 wells which were cultured independently for 20 days.
Fig. 1d	Integration	3	Integration replicates	For each ENGRAM variant, recorders were integrated into a single, polyclonal population of cells via piggyBac. This polyclonal population was split to 6 wells which were cultured independently, 3 with and 3 without 10ng/ml TNFa.
Fig. 1e-g; Supp. Fig. 2a-c	Transfection	3	Transfection replicates	A single plasmid library encoding 5N recorders was separately transfected into 3 wells of cells; these were harvested 2 days later.
Fig. 2a-c; Supp. Fig. 3c,e-g; Supp Fig. 4b-e	Integration	3	Integration replicates	A single library encoding 300 CRE-driven recorders was separately transfected and integrated to three different populations of cells via piggyBac. Cells were cultured independently for 5 days before harvesting.
Fig. 2f-g; Supp. Fig. 5a-c;	Transfection	3	Transfection replicates	A single plasmid library encoding 98 synthetic CRE-driven recorders was separately transfected into 3 wells of cells; these were harvested 2 days later.
Supp. Fig. 3a-b	Transfection	3	Transfection replicates	A single plasmid library encoding high, low, minP or promoterless recorders was separately transfected into 3 wells of cells; these were harvested 2 days later.
Fig. 3; Supp. Fig. 6	Integration	3	Integration replicates	For each signal or combination of signals, recorders were integrated into a single, polyclonal population of cells via piggyBac. This polyclonal population was split to 3 wells which were cultured independently with the indicated concentration and/or combination of agonists.

Fig. 4; Supp. Fig. 7	Integration	3	Integration replicates	A single polyclonal population of cells bearing integrated copies (via piggyBac) of both the Tet-On and Wnt recorders at high MOI was generated. For each program, cells were split from this population into three wells and cultured and treated to the specified agonist regimes separately.
Supp. Fig. 9a	Integration	3	Integration replicates	A single polyclonal population of mESCs bearing integrated copies (via piggyBac) bearing 98 synthetic CRE-driven ENGRAM recorders at high MOI was generated. Cells were split into 3 wells and cultured for 2 days in 100 ng/ml doxycycline.
Fig. 5f-g; Supp. Fig. 10b-c	Integration	2	Integration replicates	A single polyclonal population of mESCs bearing integrated copies (via piggyBac) bearing 98 synthetic CRE-driven ENGRAM recorders at high MOI was generated. For each of many gastruloids, ~300 cells from this population were seeded to a round-bottom well. For each time-window of the gastruloid time-course, 2 replicates of 20-50 independently seeded aggregates were harvested.

Raw sequencing reads were made available through SRA, but none of the processed data were made available as supplementary data. This makes it very difficult to thoroughly review the results, and readers should be given access to the processed data (in particular the TF recorder data). These data could have easily been provided in a few Excel sheets.

R3.10: We apologize that this was not clearer. The processed data and related scripts have been uploaded to Github. Using the Jupyter notebook and processed data, one can fully reproduce each figure in this manuscript. We now have added the sentence below to make it more clear:

“Processed data is available at Github: <https://github.com/shendurelab/ENGRAM>. With the provided Jupyter notebook, the results and figures in the manuscript are fully reproducible.”

What is the source of the expression data in Supp5d? Please add to the legend.

R3.11: The expression data in Supplementary Figure 5d is from the Protein Atlas project¹⁶. This paper was cited in the original main text but is now also cited in the figure legend.

Fig S5b seems a bit misleading: this correlation depends on TF reporter activity but also on barcode insertion bias (Figure 1f). The stronger the bias, the better the correlation will be... A correction for this bias seems necessary.

R3.12: We agree that insertion bias should be corrected for in cases where two enhancers/signal intensities are compared to one another. However, **Supplementary Figure 5b** shows the correlation of insertion proportions of two replicates, *i.e.* we are comparing a synthetic CRE recorder (via its 5-bp barcode) to itself in another sample (via the same 5-bp barcode). As the barcode is the same, any associated insertion bias is the same. For such comparisons, correction of barcode insertion bias is not necessary.

===== Reviewed by Bas van Steensel. Assignment accepted: 4 Jan 2024; completed: 8 Jan 2024. It is my standard policy to sign and date **all** of my manuscript reviews, irrespective of my comments and

REFERENCES

1. Chen, P. J. *et al.* Enhanced prime editing systems by manipulating cellular determinants of editing outcomes. *Cell* **184**, 5635–5652.e29 (2021).
2. Li, X. *et al.* Chromatin context-dependent regulation and epigenetic manipulation of prime editing. *bioRxiv* 2023.04.12.536587 (2023) doi:10.1101/2023.04.12.536587.
3. Choi, J. *et al.* A time-resolved, multi-symbol molecular recorder via sequential genome editing. *Nature* (2022) doi:10.1038/s41586-022-04922-8.
4. Lalanne, J.-B. *et al.* Multiplex profiling of developmental enhancers with quantitative, single-cell expression reporters. *bioRxiv* 2022.12.10.519236 (2022) doi:10.1101/2022.12.10.519236.
5. Chadly, D. M. *et al.* Reconstructing cell histories in space with image-readable base editor recording. *bioRxiv* 2024.01.03.573434 (2024) doi:10.1101/2024.01.03.573434.
6. Tang, W. & Liu, D. R. Rewritable multi-event analog recording in bacterial and mammalian cells. *Science* **360**, (2018).
7. Farzadfard, F. *et al.* Single-Nucleotide-Resolution Computing and Memory in Living Cells. *Mol. Cell* **75**, (2019).
8. Kretzschmar, K. & Watt, F. M. Lineage Tracing. *Cell* **148**, 33–45 (2012).
9. Livet, J. *et al.* Transgenic strategies for combinatorial expression of fluorescent proteins in the nervous system. *Nature* **450**, 56–62 (2007).
10. Chow, K.-H. K. *et al.* Imaging cell lineage with a synthetic digital recording system. *Science* **372**, (2021).
11. Pei, W. *et al.* Polylox barcoding reveals haematopoietic stem cell fates realized in vivo. *Nature* **548**, 456–460 (2017).
12. Grob, A. & McStay, B. Construction of synthetic nucleoli and what it tells us about propagation of sub-nuclear domains through cell division. *Cell Cycle* **13**, 2501–2508 (2014).
13. van den Brink, S. C. *et al.* Symmetry breaking, germ layer specification and axial organisation in

- aggregates of mouse embryonic stem cells. *Development* **141**, 4231–4242 (2014).
14. GloResponse™ NF-κB-RE-luc2P HEK293 Cell Line.
<https://www.promega.com/products/luciferase-assays/genetic-reporter-vectors-and-cell-lines/gloresponse-nf-kb-re-luc2p-hek293-cell-line/?catNum=E8520>.
 15. Smith, R. P. *et al.* Massively parallel decoding of mammalian regulatory sequences supports a flexible organizational model. *Nat. Genet.* **45**, 1021–1028 (2013).
 16. Uhlén, M. *et al.* Tissue-based map of the human proteome. *Science* (2015) doi:10.1126/science.1260419.
 17. Taskiran, I. I. *et al.* Cell-type-directed design of synthetic enhancers. *Nature* **626**, 212–220 (2024).
 18. de Almeida, B. P. *et al.* Targeted design of synthetic enhancers for selected tissues in the *Drosophila* embryo. *Nature* **626**, 207–211 (2024).

Reviewer Reports on the Second Revision:

Referees' comments:

Referee #1 (Remarks to the Author):

The authors answered all of our comments, and congratulations on the great work. We really look forward to the development of the field.

Referee #3 (Remarks to the Author):

As in the previous revision, the authors have gone very much out of their way to address my concerns as much as possible. The solutions that they have implemented are very reasonable. In principle I now support publication, albeit with two cautionary notes:

1. I thank the authors for providing precise definitions of "replicate experiments" (new Supp Table 7). From this table I gather that many of the replicates were not truly independent, but semi-independent. Culturing three wells of cells in parallel, split from a single transfection (Fig 1d, 3, S6, 4, S7, S9a - if I understand the descriptions correctly), may not capture the full biological variation. We all know that it can be very difficult to exactly reproduce culture conditions from week to week, and that transfections can have quite a variable impact on the cells. The conclusions about reproducibility of the assays throughout the manuscript are thus perhaps optimistic. I agree that standards in the field vary, and no doubt Nature has published numerous papers that are based on such semi-independent replicates. In my opinion this is not where we want to be as genomics / systems biology community. But I won't dig my heels in the sand in this instance, as Table S7 provides full transparency (although I hope this does not encourage others to conclude that semi-independent replicates are totally fine; something for the Nature Editorial team to ponder).

2. I imagine that the Editor would like the main text and figure legends to be condensed quite a bit. If this is the case, please make sure that the added nuanced interpretations and cautionary notes (e.g. about the synthetic TF motif reporters) are not lost!

===== Reviewed by Bas van Steensel. Assignment accepted: 10 April 2024; completed: 16 April 2024. It is my standard policy to sign and date *all* of my manuscript reviews, irrespective of my comments and recommendations. All correspondence should go via the editor. PLEASE DO NOT REMOVE THIS NOTE. =====

Referee #3 (Remarks on code availability):

I have not tested any of the code, but it appears to be complete, very structured and nicely organized into Jupyter notebook format. Processed data also appear to be available in the github folders.